# TOWARDS A STATISTICAL THEORY OF DATA SELECTION UNDER WEAK SUPERVISION

**Germain Kolossov, Andrea Montanari, Pulkit Tandon**
Granica Computing, Inc.
{germain.kolossov,andrea.montanari,pulkit.tandon}@granica.ai

## ABSTRACT

Given a sample of size $N$, it is often useful to select a subsample of smaller size $n < N$ to be used for statistical estimation or learning. Such a data selection step is useful to reduce the requirements of data labeling and the computational complexity of learning. We assume to be given $N$ unlabeled samples $\{\boldsymbol{x}_i\}_{i \leq N}$, and to be given access to a 'surrogate model' that can predict labels $y_i$ better than random guessing. Our goal is to select a subset of the samples, to be denoted by $\{\boldsymbol{x}_i\}_{i \in G}$, of size $|G| = n < N$. We then acquire labels for this set and we use them to train a model via regularized empirical risk minimization. By using a mixture of numerical experiments on real and synthetic data, and mathematical derivations under low- and high- dimensional asymptotics, we show that: $(i)$ Data selection can be very effective, in particular beating training on the full sample in some cases; $(ii)$ Certain popular choices in data selection methods (e.g. unbiased reweighted subsampling, or influence function-based subsampling) can be substantially suboptimal.

## 1 INTRODUCTION

Labeling is a notoriously laborious task in machine learning. A possible approach towards reducing this burden is to select a small subset of training samples that are most valuable for training.

Data selection-based learning consists of two steps – $(i)$ *Data Selection*: given feature vectors $\boldsymbol{X} := (\boldsymbol{x}_i)_{i \leq N}$, select a subset $G \subseteq [N]$ of size $n$ (or close to $n$); $(ii)$ *Training*: having acquired labels for the selected subset $\{y_i\}_{i \in G}$, train a model $f(\,\cdot\,; \boldsymbol{\theta}) : \mathbb{R}^p \to \mathbb{R}$ (with parameters $\boldsymbol{\theta}$) on the labeled data $\{(\boldsymbol{x}_i, y_i)\}_{i \in G}$. Throughout this paper we will focus on methods with the following structure. In the first step, set $G$ is generated by selecting each datapoint $i$ independently with probability $\pi_i = \pi(\boldsymbol{x}_i)$. Namely

$$\mathbb{P}(i \in G | \boldsymbol{y}, \boldsymbol{X}) = \pi_i, \quad \text{independently for } i \leq N. \tag{1.1}$$

The second step (training) is carried out by (weighted) empirical risk minimization (ERM) over the selected set $G$. Namely, given loss function $\ell : \mathbb{R} \times \mathbb{R} \to \mathbb{R}$ and regularizer $\Omega : \mathbb{R}^p \to \mathbb{R}$, we solve

$$\hat{\boldsymbol{\theta}} = \arg\min_{\boldsymbol{\theta}} \hat{R}_N(\boldsymbol{\theta}), \qquad \hat{R}_N(\boldsymbol{\theta}) := \frac{1}{N} \sum_{i \in G} w_i \, \ell(y_i, f(\boldsymbol{x}_i; \boldsymbol{\theta})) + \lambda \, \Omega(\boldsymbol{\theta}) \,. \tag{1.2}$$

The weights $w_i$ can depend on the feature vectors, and are designed as to reduce the error. A popular choice is $w_i = \text{cst}/\pi_i$ because the resulting $\hat{R}_N(\boldsymbol{\theta})$ is an unbiased version of the full empirical risk.

Throughout this paper, we will assume data $(\boldsymbol{x}_i, y_i)$ to be i.i.d. samples from a common distribution $\mathbb{P} \in \mathscr{P}(\mathbb{R}^d \times \mathbb{R})$ and will evaluate a data selection procedure via the test error:

$$R_{\text{test}}(\boldsymbol{\theta}) = \mathbb{E} \, \ell_{\text{test}}\big(y_{\text{new}}; f(\boldsymbol{x}_{\text{new}}; \boldsymbol{\theta})\big) \,, \tag{1.3}$$

where expectation is taken with respect to the test sample $(\boldsymbol{x}_{\text{new}}, y_{\text{new}}) \sim \mathbb{P}$. We will denote the (training) population risk by $R(\boldsymbol{\theta}) = \mathbb{E} \, \ell\big(y; f(\boldsymbol{x}; \boldsymbol{\theta})\big)$.

Data selection has been extensively studied in a variety of settings, including experimental design, active learning, learning algorithms and data optimization. While some heuristics (e.g., focus on

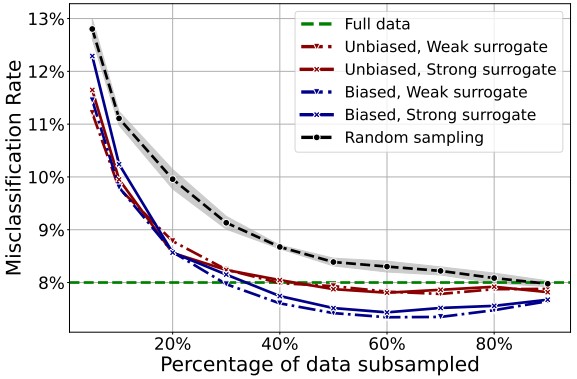
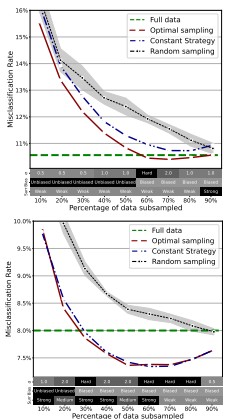

Figure 1: Misclassification error in an image classification problem after data selection (logistic regression on SwAV embeddings). Left: $N = 34345$ samples, $p = 2048$ dimensions. We use surrogate models trained on a small separated fraction of the data ('strong': $N_{\text{su}} = 14720$, 'weak': $N_{\text{su}} = 1472$). Right plots: subsampling schemes optimized over ridge regularization $\lambda$; exponent $\alpha$ in the data selection scheme (larger $\alpha$ select 'harder' samples); biased vs unbiased selection; and strength of the surrogate. Constant strategy: biased selection with $\lambda = 0.01$, $\alpha = 0.5$, and weak surrogate. Top: $N = 3434$; bottom $N = 34345$. See Section 7.

samples that are most difficult to predict) have resurfaced from different viewpoints, their implementation and effectiveness depends on the problem formulation. Unlike the most common active learning scenario, we crucially assume to be given a fixed data sample $\{\boldsymbol{x}_i\}_{i \leq N}$ (without labels) and carry out a single data selection step, often referred as 'Pool-based active learning' (Settles, 2012).

Before summarizing our contributions, it is useful to mention some of the existing approaches.

**Bayesian methods.** Within a Bayesian setting, to select the subsample $G$ is to minimize the conditional entropy of $\boldsymbol{\theta}$ given the data Lindley (1956); Seung et al. (1992). This is a first example of "uncertainty sampling." We refer to Houlsby et al. (2011); Gal et al. (2017) for recent pointers, with a focus on online active learning. See Cui et al. (2021) for a recent exception.

**Heuristic approaches.** Several groups developed measures of the impact of each sample on the model: Lewis & Gale (1994) use probabilities predicted by a single current model; Vodrahalli et al. (2018) use the norm of $\nabla_{\boldsymbol{\theta}} \ell(y_i, f(\boldsymbol{x}_i, \boldsymbol{\theta}))$; Jiang et al. (2019) use the loss itself $\ell(y_i, f(\boldsymbol{x}_i, \boldsymbol{\theta}))$.

**Leverage scores** have been used for a long time in statistics as a measure of importance of a data point $i \in \{1, \ldots, N\}$ in linear regression (Chatterjee & Hadi, 1986). Recall that the leverage of sample $\boldsymbol{x}_i$ is defined as $H_{ii} := \boldsymbol{x}_i^{\top}(\boldsymbol{X}^{\top}\boldsymbol{X})^{-1}\boldsymbol{x}_i$. Their study was reinvigorated by a numerical linear algebra perspective (Drineas & Mahoney, 2006; Drineas et al., 2011). Statistical analyses of leverage score-based data selection was developed in Ma et al. (2014); Raskutti & Mahoney (2016); Ma et al. (2022). These works all assume $w_i = 1/\pi_i$ for unbiasedness.

**Influence functions.** Let $\boldsymbol{\theta}_* = \arg\min_{\boldsymbol{\theta} \in \mathbb{R}^p} R(\boldsymbol{\theta})$ denote the population minimizer. For M-estimators (1.2) (under regularity conditions) approximate linearity holds as $n \to \infty$ for $p$ fixed (van der Vaart, 2000)

$$\hat{\boldsymbol{\theta}} - \boldsymbol{\theta}_* = \frac{1}{n} \sum_{i \in G} w_i \psi(\boldsymbol{x}_i, y_i) + o_P(1/\sqrt{n}),$$

where $\psi : \mathbb{R}^p \times \mathbb{R} \to \mathbb{R}$ is the so-called influence function. This approximation can be used to select the probabilities $\pi_i$ and weights $w_i$. Several authors have developed this approach in the context of generalized linear models, while assuming unbiasedness $w_i = 1/\pi_i$, resulting in the choice $\pi_i \propto \|\psi(\boldsymbol{x}_i, y_i)\|_2$ (Ting & Brochu, 2018; Wang et al., 2018; Ai et al., 2021). Wang et al. (2020) shows potential for improvement by using biased schemes. These works assume a specific asymptotics in which $1 \ll n \ll N$. In many applications, we would like the generalization error achieved from the $n$-subsample to be comparable to the one with full sample and hence focus on the case $n = \gamma N$, $\gamma = \Theta(1)$.

**Margin-based selection.** In a recent paper Sorscher et al. (2022) show that, in the case of the binary perceptron, under a noiseless teacher-student distribution, selecting samples far from the margin is beneficial in a high-dimensional or data-poor regime. This is in stark contrast with influence-function and related approaches that upsample data points whose labels are more difficult to predict. While our results confirm these findings, measuring uncertainty in terms of distance from the margin is somewhat specific to binary classification under certain distributional assumptions[1].

## 2 DEFINITIONS

Since our objective is to alleviate the need to label data, we will focus on data selection schemes that do not use the labels $(y_i)_{i \leq N}$. On the other hand, we will assume to have access to a surrogate model $\mathsf{P}_{su}(dy|\boldsymbol{x})$ which is a probability kernel from $\mathbb{R}^d$ to $\mathbb{R}$. We will consider two settings – $(i)$ Ideal surrogate: In this case $\mathsf{P}_{su}(dy|\boldsymbol{x}) = \mathbb{P}(dy|\boldsymbol{x})$ is the actual conditional distribution of the labels. Of course this is the very object we want to learn. Since learning scheme is constrained, the problem is nevertheless non-trivial and informative. $(ii)$ Imperfect surrogates: In this case $\mathsf{P}_{su}(dy|\boldsymbol{x})$ is predictive of the actual value of the labels, but does not coincide with the Bayes predictor.

The selection probabilities $\pi_i$ and weights $w_i$ can depend on $\boldsymbol{x}_i$ and $\mathsf{P}_{su}(\cdot|\boldsymbol{x}_i)$ (the conditional distribution of $y_i$ given $\boldsymbol{x}_i$ under the surrogate). We will omit the dependence of the surrogate predictions unless needed, and write $\pi_i = \pi(\boldsymbol{x}_i)$, $w_i = w(\boldsymbol{x}_i)$.

It is convenient to encode a general data selection process into random variables $S_i(\boldsymbol{x}_i) \geq 0$ which depend on $\boldsymbol{x}_i, \mathsf{P}_{su}(\cdot|\boldsymbol{x}_i)$, and also on additional randomness independent across different samples[2]. We recover the original formulation by setting $S_i(\boldsymbol{x}_i) = w(\boldsymbol{x}_i)\,\mathbf{1}_{i \in G}$, $\mathbb{P}(i \in G|\boldsymbol{X}, \boldsymbol{y}) = \pi(\boldsymbol{x}_i)$. We can thus rewrite the empirical risk (1.2) as:

$$\hat{R}_N(\boldsymbol{\theta}) = \frac{1}{N} \sum_{i=1}^N S_i(\boldsymbol{x}_i)\, L(\boldsymbol{\theta}; y_i, \boldsymbol{x}_i) + \lambda\,\Omega(\boldsymbol{\theta})\,, \qquad L(\boldsymbol{\theta}; y, \boldsymbol{x}) := \ell(y, f(\boldsymbol{x}; \boldsymbol{\theta}))\,. \tag{2.1}$$

We will enforce that the target sample size $n$ is achieved in expectation, namely

$$n = \sum_{i=1}^N \mathbb{E}\,\pi(\boldsymbol{x}_i)\,, \qquad \pi(\boldsymbol{x}_i) := \mathbb{P}\big(S_i(\boldsymbol{x}_i) > 0 \big| \boldsymbol{X}, \boldsymbol{y}\big)\,. \tag{2.2}$$

**Definition 2.1.** *A data selection scheme is* unbiased *if* $\mathbb{E}\{S(\boldsymbol{x})|\boldsymbol{X}, \boldsymbol{y}\} = 1$. *We denote the set of unbiased data selection schemes by* $\mathscr{U}$.

Throughout, we consider $n, N \to \infty$, with converging ratio $n/N \to \gamma \in (0, 1)$. We refer to $\gamma$ as to the 'subsampling fraction.' We are interested in $\gamma$ bounded away from 0 because we want to keep the accuracy close to the full sample accuracy. Standard notations are summarized in Appendix A.

## 3 SUMMARY OF RESULTS

Our theory is based on two types of asymptotics, that capture complementary regimes. In Section 4 we will study the low-dimensional asymptotics, whereby $p$ is fixed as $n, N \to \infty$. This analysis applies to fairly general data distributions and models $f(\cdot; \boldsymbol{\theta})$. In Section 5, we will instead assume $p \to \infty$ as $n, N \to \infty$, with $N/p \to \delta_0 \in (0, \infty)$. In this case we study convex M-estimators, for models that are linear in the features $\boldsymbol{x}_i$. Namely, $L(\boldsymbol{\theta}; y_i, \boldsymbol{x}_i) = L_*(\langle\boldsymbol{\theta}, \boldsymbol{x}_i\rangle; y_i)$, with $L_*$ convex in its first argument, and $\boldsymbol{x}_i$ standard Gaussian. We next summarize some of the insights from our work and illustrate them in Figure 1 (experiments with real data, described in Section 7).

**Unbiased subsampling can be suboptimal.** As mentioned in the introduction, the majority of theoretical studies assumes unbiased subsampling: $w_i \propto 1/\pi_i$. We show both theoretically and empirically that this can lead to potentially unbounded loss in accuracy with respect to biased schemes.

---

[1]Sorscher et al. (2022) proposes also schemes that are empirically effective more generally, but the connection with their mathematical analysis is indirect.

[2]Formally, $S_i(\boldsymbol{x}_i) = s(U_i, \boldsymbol{x}_i)$, where $s$ is a deterministic function and $(U_i)_{i \leq N} \sim_{i.i.d.} \mathsf{Unif}([0, 1])$.

Low-dimensional asymptotics provide a compelling explanation: estimation error is inversely proportional to the curvature of the expected risk. Unbiased methods do not change the curvature, while biased methods can increase the curvature. In fact, we will prove that unbiased methods are suboptimal under a broad set of conditions. This result is illustrated in Figure 1. In particular, in Figure 1 we compare a biased and an unbiased scheme (with selection probabilities which approximate the influence-function scheme).

**Optimal selection depends on the subsampling fraction.** Popular data selection methods are independent of subsampling fraction in the sense that the selection probabilities $\pi_i$ can be computed without knowing the target sampling fraction $n/N$, except from the overall normalization that enforces the constraint (2.2). In contrast, we will see that optimal schemes depend in a non-trivial way on $n/N$. This is very natural: the utility of sample $(\boldsymbol{x}_i, y_i)$ depends on the other selected samples.

**Better surrogate models do not always lead to better selection.** How to proceed in absence of the labels $y_i$? A natural idea would be to start from selection probabilities that use the labels, $\pi_0(y_i, \boldsymbol{x}_i)$, and then take a conditional expectation with respect to $y_i$, under the surrogate model: $\pi(\boldsymbol{x}_i) = \mathsf{E}_{\mathrm{su}}\{\pi_0(y_i, \boldsymbol{x}_i)|\boldsymbol{x}_i\}$. We show that this is not always optimal, and in particular better surrogate models do not always lead to better selection (if used in a plugin fashion as described).

**Uncertainty-based subsampling is effective.** The simplest rule of thumb emerging from our work confirms earlier research: subsampling schemes based on how uncertain surrogate predictions are, often perform well. On the other hand, influence-based subsampling is not generally optimal. First: in low-dimension, stronger bias towards hard examples is beneficial, cf. Section 4.2. Second: in certain high-dimensional cases, this bias must be reversed, and 'easier' examples should be selected, see Section 5 (in agreement with Sorscher et al. (2022)). Third, adding nuance to the previous point, selecting the 'hardest' or 'easiest' even in high dimension, is not always optimal, see e.g. Section 6.

**Data subsampling can beat full-sample training.** Our objective is to select a fraction $\gamma \in (0, 1)$ of the data such that, training on the selected subset yields test error close to training on the whole dataset. To our surprise, see Figure 1, we discover that good data selection can actually reduce the test error below the one obtained from the full sample for $\gamma$ as small as $0.4$. A naive explanation would be that information is being passed from the surrogate model to the trained model via data selection. While this is of course true, the effect is more subtle. Indeed, we achieve a reduction in test error even when the surrogate model is trained on significantly less than $(1 - \gamma)N$ random samples (so that overall, we are using significantly less than $N$ labels). Recent empirical studies (Nakkiran et al., 2021; Guo et al., 2022; Yang et al., 2022; Sorscher et al., 2022; Gadre et al., 2024) have observed similar phenomenon for deep models with real data but we show that this is true even for simpler settings.

## 4 LOW-DIMENSIONAL ASYMPTOTICS

In this section we study the classical asymptotics in which $p$ is fixed as $n, N \to \infty$. The asymptotics of $\hat{\boldsymbol{\theta}}$ depends on the expectation of the empirical risk (2.1):

$$R_S(\boldsymbol{\theta}) := \mathbb{E}\{S(\boldsymbol{x}) L(\boldsymbol{\theta}; y, \boldsymbol{x})\}. \tag{4.1}$$

(In this notation, the argument $S$ indicates the dependence on the *function* that defines the subsampling procedure.) The conditional gradient covariance, and conditional Hessian will play an important role, and are defined below (recall that $\boldsymbol{\theta}_* = \arg\min_{\boldsymbol{\theta}} R(\boldsymbol{\theta})$):

$$\boldsymbol{G}(\boldsymbol{x}) := \mathbb{E}\{\nabla_{\boldsymbol{\theta}} L(\boldsymbol{\theta}_*; y, \boldsymbol{x})\nabla_{\boldsymbol{\theta}} L(\boldsymbol{\theta}_*; y, \boldsymbol{x})^\mathsf{T}|\boldsymbol{x}\}, \qquad \boldsymbol{H}(\boldsymbol{x}) := \mathbb{E}\{\nabla_{\boldsymbol{\theta}}^2 L(\boldsymbol{\theta}_*; y, \boldsymbol{x})|\boldsymbol{x}\}. \tag{4.2}$$

We consider weighted quadratic error $\|\hat{\boldsymbol{\theta}} - \boldsymbol{\theta}_*\|_{\boldsymbol{Q}}^2 := \langle \hat{\boldsymbol{\theta}} - \boldsymbol{\theta}_*, \boldsymbol{Q}(\hat{\boldsymbol{\theta}} - \boldsymbol{\theta}_*)\rangle$, where $\boldsymbol{Q} \in \mathbb{R}^{p \times p}$. This covers both the standard $\ell_2$ estimation error (by setting $\boldsymbol{Q} = \boldsymbol{I}$) and, under smoothness conditions, the test error $R_{\mathrm{test}}(\hat{\boldsymbol{\theta}})$ (setting $\boldsymbol{Q} = \nabla^2 R_{\mathrm{test}}(\boldsymbol{\theta}_*)$). We define the asymptotic error coefficient $\rho$ via

$$\mathbb{E}\|\hat{\boldsymbol{\theta}} - \boldsymbol{\theta}_*\|_{\boldsymbol{Q}}^2 = \rho(S; \boldsymbol{Q})/N + o_P(N^{-1}). \tag{4.3}$$

In other words, $\rho(S; \boldsymbol{Q}) := \lim_{N \to \infty} N\mathbb{E}\{\|\hat{\boldsymbol{\theta}} - \boldsymbol{\theta}_*\|_{\boldsymbol{Q}}^2\}$. A general formula for $\rho(S; \boldsymbol{Q})$ follows from standard arguments (see Appendix B). In the next two sections, we discuss optimal choices of the subsampling procedure under this asymptotics.

## 4.1 Unbiased data selection

We begin by considering unbiased schemes, cf. Definition 2.1. This case is covered by earlier work Ting & Brochu (2018); Wang et al. (2018); Ai et al. (2021), but provides useful context.

The key simplification is that, in the unbiased case, the matrix $\boldsymbol{H}_S$ (see Appendix B) does not depend on $S$, namely $\boldsymbol{H}_S = \boldsymbol{H}$, where

$$\boldsymbol{H} = \nabla^2 R(\boldsymbol{\theta}_*), \qquad R(\boldsymbol{\theta}) := \mathbb{E} L(\boldsymbol{\theta}; y, \boldsymbol{x}). \tag{4.4}$$

We thus recover the standard subsampling scheme based on influence functions. (See Appendix C.)

**Proposition 4.1.** *Under the assumptions of Proposition B.1, further assume $\boldsymbol{Q} \succeq \boldsymbol{0}$, $\boldsymbol{H} \succ \boldsymbol{0}$. Then $\rho(S; \boldsymbol{Q})$ is minimized among unbiased data selection schemes by a scheme of the form: $S_{\mathrm{unb}}(\boldsymbol{x}) = \pi_{\mathrm{unb}}(\boldsymbol{x})^{-1}$ with probability $\pi_{\mathrm{unb}}(\boldsymbol{x})$ and $S_{\mathrm{unb}}(\boldsymbol{x}) = 0$ otherwise, where:*

$$\pi_{\mathrm{unb}}(\boldsymbol{x}) = \min\left(1; c(\gamma)\, Z(\boldsymbol{x})^{1/2}\right),$$

$$Z(\boldsymbol{x}) := \mathsf{Tr}\big(\boldsymbol{G}(\boldsymbol{x}) \boldsymbol{H}^{-1} \boldsymbol{Q} \boldsymbol{H}^{-1}\big) = \mathbb{E}\big\{\big\|\nabla_{\boldsymbol{\theta}} L(\boldsymbol{\theta}; y, \boldsymbol{x})\big\|_{\boldsymbol{H}^{-1}\boldsymbol{Q}\boldsymbol{H}^{-1}}^2 \big| \boldsymbol{x}\big\}.$$

*Here the constant $c(\gamma)$ is defined so that $\mathbb{E}\pi_{\mathrm{unb}}(\boldsymbol{x}) = \gamma$.*

*Finally the the optimal coefficient $\rho_{\mathrm{unb}}(\boldsymbol{Q}) := \inf_{S \in \mathscr{U}} \rho(S; \boldsymbol{Q}) = \rho(S_{\mathrm{unb}}; \boldsymbol{Q})$ is given by*

$$\rho_{\mathrm{unb}}(\boldsymbol{Q}) = \inf_{\mathbb{E}\pi(\boldsymbol{x})=\gamma} \mathbb{E}\Big\{Z(\boldsymbol{x})/\pi(\boldsymbol{x})\Big\} = \mathbb{E}\max\left(Z(\boldsymbol{x})^{1/2}/c(\gamma); Z(\boldsymbol{x})\right). \tag{4.5}$$

Denote by $S_{\mathrm{rand}}$ random subsampling, i.e. $S_{\mathrm{rand}}(\boldsymbol{x}) = 1/\gamma$ with probability $\gamma$, and $S_{\mathrm{rand}}(\boldsymbol{x}) = 0$ else ($S_{\mathrm{rand}}$ is unbiased). We next establish some basic properties of $\rho_{\mathrm{unb}}(\boldsymbol{Q}; \gamma)$, cf. Appendix D.

**Proposition 4.2.** *Write $\rho_{\mathrm{unb}}(\boldsymbol{Q}; \gamma)$ for the optimal unbiased coefficient when the subsampling fraction equals $\gamma$. Then: (1) $\rho_{\mathrm{unb}}(\boldsymbol{Q}) \le \rho_{\mathrm{rand}}(\boldsymbol{Q})$ with the inequality being strict unless $Z(\boldsymbol{x})$ is almost surely constant or $\gamma = 1$; (2) $\gamma \mapsto \rho_{\mathrm{unb}}(\boldsymbol{Q}; \gamma)$ is monotone non-increasing (for $\boldsymbol{Q} \succeq \boldsymbol{0}$).*

Proof of this proposition is provided in Appendix D. While monotonicity may appear intuitively obvious, we will see in the Section 4.3 that it does not hold for biased subsampling.

**Remark 4.1** (Connection with influence functions)**.** Under the assumptions of Proposition 4.1 the influence function is $\psi(\boldsymbol{x}, y) = -\boldsymbol{H}^{-1}\nabla_{\boldsymbol{\theta}} L(\boldsymbol{\theta}_*; y, \boldsymbol{x})$ and therefore we can rewrite (omitting the capping at 1) $\pi(\boldsymbol{x}_i) \propto \mathbb{E}\big\{\big\|\psi(\boldsymbol{x}_i, y_i)\big\|_{\boldsymbol{Q}}^2 \big| \boldsymbol{x}_i\big\}^{1/2}$. We recovered influence function-based subsampling Ting & Brochu (2018); Wang et al. (2018); Ai et al. (2021). Interestingly, the optimal way to deal with unknown $y_i$ is to take conditional expectation of the *square* of the sampling probability: we compute $\mathbb{E}\big\{\big\|\psi(\boldsymbol{x}_i, y_i)\big\|_{\boldsymbol{Q}}^2 \big| \boldsymbol{x}_i\big\}^{1/2}$ instead of $\mathbb{E}\big\{\|\psi(\boldsymbol{x}_i, y_i)\|_{\boldsymbol{Q}} \big| \boldsymbol{x}_i\big\}$. See also Appendix C.

**Example 4.2** (Generalized linear models)**.** We assume $\boldsymbol{x}_i$ to be a centered vector and

$$\mathbb{P}(\mathrm{d}y_i | \boldsymbol{x}_i) = \exp\big\{y_i \langle \boldsymbol{\theta}_*, \boldsymbol{x}_i\rangle - \phi(\langle\boldsymbol{\theta}_*, \boldsymbol{x}_i\rangle)\big\} \nu_0(\mathrm{d}y_i). \tag{4.6}$$

Here $\nu_0$ is the reference measure, e.g. $\nu_0 = \delta_{+1} + \delta_{-1}$ for logistic regression. We fit a model of the same type and train via the loss $L(\boldsymbol{\theta}; y, \boldsymbol{x}) = -y\langle\boldsymbol{\theta}, \boldsymbol{x}\rangle + \phi(\langle\boldsymbol{\theta}, \boldsymbol{x}\rangle)$. Hence $\nabla_{\boldsymbol{\theta}} L(\boldsymbol{\theta}; y, \boldsymbol{x}) = -(y - \phi'(\langle\boldsymbol{\theta}, \boldsymbol{x}\rangle))\boldsymbol{x}^{\mathsf{T}}$ and $\nabla^2 R(\boldsymbol{\theta}_*) = \mathbb{E}\{\phi''(\langle\boldsymbol{\theta}, \boldsymbol{x}\rangle)\boldsymbol{x}\boldsymbol{x}^{\mathsf{T}}\} := \boldsymbol{H}$. A simple calculation yields:

$$Z(\boldsymbol{x}) = \phi''(\langle\boldsymbol{\theta}_*, \boldsymbol{x}\rangle) \cdot \langle\boldsymbol{x}, \boldsymbol{H}^{-1}\boldsymbol{Q}\boldsymbol{H}^{-1}\boldsymbol{x}\rangle. \tag{4.7}$$

In what follows, we will consider also a generalization of this model whereby (here, for each $z$, $\mathsf{P}(\,\cdot\,|z)$ is a probability distribution over $\mathbb{R}$):

$$\mathbb{P}(y_i \in A | \boldsymbol{x}_i) = \mathsf{P}(A | \langle\boldsymbol{\theta}_0, \boldsymbol{x}_i\rangle). \tag{4.8}$$

**Example 4.3** (Linear regression)**.** Setting $\nu_0(\mathrm{d}y) = \exp(-y_i^2/2)/\sqrt{2\pi}$, $\phi(t) = t^2/2$ above, we obtain linear regression: $L(\boldsymbol{\theta}; y, \boldsymbol{x}) = L_{\mathrm{test}}(\boldsymbol{\theta}; y, \boldsymbol{x}) = (y - \langle\boldsymbol{\theta}, \boldsymbol{x}\rangle)^2/2$. Equation (4.7) yields $Z(\boldsymbol{x}) \propto \langle\boldsymbol{x}, \boldsymbol{\Sigma}^{-1}\boldsymbol{Q}\boldsymbol{\Sigma}^{-1}\boldsymbol{x}\rangle$, and, for prediction error ($\boldsymbol{Q} = \boldsymbol{\Sigma}$), $Z(\boldsymbol{x}) \propto \langle\boldsymbol{x}, \boldsymbol{\Sigma}^{-1}\boldsymbol{x}\rangle$: we recover a population version of the leverage score.

### 4.2 BIASED DATA SELECTION

We now consider biased subsampling schemes $S$. It is straightforward to see that the weights can always be chosen to be deterministic (see Appendix E). In other words, $S(\boldsymbol{x}) = w(\boldsymbol{x})$ with probability $\pi(\boldsymbol{x})$, and $S(\boldsymbol{x}) = 0$ otherwise, for two deterministic functions $w, \pi$. We can identify these data selection scheme with the pair $(\pi, w)$, hence we will also write $\rho(\pi, w; \boldsymbol{Q})$ for $\rho(S; \boldsymbol{Q})$.

Among these, we focus on 'non-reweighting schemes' that are defined by the additional condition $w = 1$ and denote them by $\mathscr{N}$. Non-reweighting schemes are practically convenient since they do not require to modify the training procedure. Optimal non-reweighting schemes have a simple structure, as stated below. (See Appendix F.)

**Proposition 4.3.** *Under the assumptions of Proposition B.1, further assume $\boldsymbol{Q} \succeq \boldsymbol{0}$, $\boldsymbol{H} \succ \boldsymbol{0}$, $\mathbb{E}\{\nabla_{\boldsymbol{\theta}} L(\boldsymbol{\theta}_*; y, \boldsymbol{x}) | \boldsymbol{x}\} = \boldsymbol{0}$, and $\boldsymbol{x} \mapsto \boldsymbol{G}(\boldsymbol{x})$, $\boldsymbol{H}(\boldsymbol{x})$ to be continuous. Define $\boldsymbol{G}_\pi := \mathbb{E}_\pi \boldsymbol{G}(\boldsymbol{x})$, $\boldsymbol{H}_\pi := \mathbb{E}_\pi \boldsymbol{H}(\boldsymbol{x})$, where $\mathbb{E}_\pi f(\boldsymbol{x}) := \mathbb{E}\{f(\boldsymbol{x}) \pi(\boldsymbol{x})\} / \mathbb{E}\{\pi(\boldsymbol{x})\}$. Finally define the function*

$$Z(\boldsymbol{x}; \pi) := -\mathsf{Tr}\{\boldsymbol{G}(\boldsymbol{x}) \boldsymbol{H}_\pi^{-1} \boldsymbol{Q} \boldsymbol{H}_\pi^{-1}\} + 2\mathsf{Tr}\{\boldsymbol{H}(\boldsymbol{x}) \boldsymbol{H}_\pi^{-1} \boldsymbol{Q} \boldsymbol{H}_\pi^{-1} \boldsymbol{G}_\pi \boldsymbol{H}_\pi^{-1}\}. \qquad (4.9)$$

*Then there exists $\pi_{\mathrm{nr}}$ achieving the minimum of the asymptotic error over non-reweighting schemes $\rho(\pi_{\mathrm{nr}}, 1; \boldsymbol{Q}) = \inf_{S \in \mathscr{N}} \rho(S; \boldsymbol{Q})$. Further, if $\boldsymbol{H}_{\pi_{\mathrm{nr}}} \succ \boldsymbol{0}$ strictly, then $\pi_{\mathrm{nr}}$ takes the form*

$$\pi_{\mathrm{nr}}(\boldsymbol{x}) = \begin{cases} 1 & \text{if } Z(\boldsymbol{x}; \pi_{\mathrm{nr}}) > \lambda, \\ 0 & \text{if } Z(\boldsymbol{x}; \pi_{\mathrm{nr}}) < \lambda, \\ b(\boldsymbol{x}) \in [0, 1] & \text{if } Z(\boldsymbol{x}; \pi_{\mathrm{nr}}) = \lambda. \end{cases} \qquad (4.10)$$

*where $\lambda$ and $b(\boldsymbol{x})$ are chosen so that $\mathbb{E}\pi_{\mathrm{nr}}(\boldsymbol{x}) = \gamma$. The resulting optimal asymptotic error is $\rho_{\mathrm{nr}}(\boldsymbol{Q}) = \gamma^{-1} \mathsf{Tr}(\boldsymbol{G}_{\pi_{\mathrm{nr}}} \boldsymbol{H}_{\pi_{\mathrm{nr}}}^{-1} \boldsymbol{Q} \boldsymbol{H}_{\pi_{\mathrm{nr}}}^{-1}) = \gamma^{-1} \inf_{\pi: \mathbb{E}\pi = \gamma} \mathsf{Tr}(\boldsymbol{G}_\pi \boldsymbol{H}_\pi^{-1} \boldsymbol{Q} \boldsymbol{H}_\pi^{-1})$.*

In many cases of interest, the set $\{\boldsymbol{x} : Z(\boldsymbol{x}; \pi_*) = \lambda\}$ has zero measure (e.g. if $\boldsymbol{x}$ has a density). Roughly speaking, the above implies that, under the non-reweighing scheme (and in the low-dimensional asymptotics) we should select the $n$ examples with largest score ('hardest' examples).

Another important difference compared to the unbiased case is that the score is now computed with respect to the selected data, namely $\boldsymbol{H}$ is replaced by $\boldsymbol{H}_\pi$. As anticipated in Section 3, the selected set depends on $\gamma$ in a nontrivial way.

We compare $\rho_{\mathrm{nr}}$ and $\rho_{\mathrm{unb}}$, proving that unbiased sampling can be arbitrarily worse than non-reweighting. (See Appendix G.)

**Theorem 1.** *Consider least-squares under the model $y_i = \langle \boldsymbol{\theta}_*, \boldsymbol{x}_i \rangle + \varepsilon_i$, with $\mathbb{E}(\varepsilon_i^2 | \boldsymbol{x}_i) = \tau^2$. For any $M$, there is a distribution of $\boldsymbol{x}_i$ and a $\gamma \in (0, 1)$ such that $\rho_{\mathrm{unb}}(\boldsymbol{\Sigma}; \gamma) / \rho_{\mathrm{nr}}(\boldsymbol{\Sigma}; \gamma) > M$.*

### 4.3 DATA SELECTION CAN BEAT FULL-SAMPLE ESTIMATION

One striking phenomenon illustrated in Figure 1 is that ERM with respect to a selected subset of $n < N$ samples out of $N$ total produces a better model than ERM on the full dataset. This is the case even when the surrogate model is trained on $n_{\mathrm{su}}$ samples, with $n_{\mathrm{su}} + n$ is significantly below $N$.

Is the non-monotonic behavior compatible with theoretical expectations? The answer depends on the data selection scheme (see Appendix H for details and an illustration): $(i)$ Unbiased data selection is always monotone, cf. Section 4.1; $(ii)$ The optimal data selection scheme with reweighting is also monotone by a simple reduction argument. However, for $n = N$, this scheme does not reduce to unweighted ERM, but to optimally weighted ERM. $(iii)$ Finally, non-monotonicity is possible in a neighborhood of $\gamma = 1$ for non-reweighting schemes, as established below (see Appendix I).

**Theorem 2.** *Assume $\boldsymbol{x}_i \sim \mathsf{N}(0, \boldsymbol{I}_d)$, and $y_i$ distributed according to Eq. (4.8). Let $\boldsymbol{Q} = \boldsymbol{H} := \nabla^2 R(\boldsymbol{\theta}_*)$. Then both for linear and logistic regression there exists $\mathsf{P}(A|t)$ such that $\gamma \mapsto \rho_{\mathrm{nr}}(\boldsymbol{Q}; \gamma)$ is monotone increasing in a non-empty interval $(\gamma_0, 1)$.*

### 4.4 IS UNBIASED SUBSAMPLING EVER OPTIMAL?

We will next prove that, in a natural setting, unbiased subsampling is always suboptimal below a certain subsampling ratio. (The last condition is likely to be a proof artifact, cf. Appendix J.) We show in Appendix J that the conditions below hold generically for generalized linear models.

**Theorem 3.** *Consider the setting of Proposition B.1, and consider the prediction error under test loss equal to training loss (i.e. $\boldsymbol{Q} = \boldsymbol{H} := \nabla^2 R(\boldsymbol{\theta}_*)$). Further assume:* **A1.** *For $\boldsymbol{G}(\boldsymbol{x})$ and $\boldsymbol{H}(\boldsymbol{x})$ defined as in Eqs (4.2), we have $\boldsymbol{G}(\boldsymbol{x}) = \boldsymbol{H}(\boldsymbol{x})$. (This is for instance the case in maximum likelihood);* **A2.** *$\boldsymbol{G}(\boldsymbol{x})$ is almost surely non-zero has an almost sure upper bound;* **A3.** *The distribution of $\boldsymbol{G}(\boldsymbol{x})/\mathsf{Tr}(\boldsymbol{G}(\boldsymbol{x})\boldsymbol{H}^{-1})^{1/2}$ is not not supported on any strict affine subspace of the set of $d \times d$ symmetric matrices.* **A4.** *$\mathbb{E}\{\nabla_{\boldsymbol{\theta}} L(\boldsymbol{\theta}_*; y, \boldsymbol{x}) | \boldsymbol{x}\} = \boldsymbol{0}$.*

*Then there exists $\gamma_0 > 0$ such that, for any $\gamma \in (0, \gamma_0)$ there is a biased subsampling scheme asymptotically outperforming the best unbiased scheme.*

### 4.5 IMPERFECT SURROGATES

In a more realistic setting, the surrogate model $\mathsf{P}_{\text{su}}(\mathrm{d}y|\boldsymbol{x})$ does not coincide with the actual conditional distribution of $y_i$ given $\boldsymbol{x}_i$. We model this situation using a minimax point of view. Namely, we assume that the actual conditional distribution is in a neighborhood of the surrogate model, and study the worst case error in this neighborhood. The minimax theorem then implies that we should use data selection schemes that are optimal for the worst data distribution in this neighborhood. We will not pursue the construction of minimax optimal data selection schemes in this paper. However, we point out that the broad conclusion is consistent with some of our empirical findings in Section 7. Namely, in certain cases using a less accurate surrogate model yields better data selection. Theorems and examples are presented in Appendix K.

## 5 HIGH-DIMENSIONAL ASYMPTOTICS: GENERALIZED LINEAR MODELS

We next study a high dimensional setting whereby the number of samples $N$ and the dimension $p$ diverge simultaneously. More precisely, we assume $n, N, p \to \infty$ with $n/N \to \gamma$, $N/p \to \delta_0$ for some $\gamma, \delta_0 \in (0, \infty)$. We will restrict ourselves to the case of (potentially misspecified) generalized linear models already introduced in Example 4.2. As we will see in this section and the next, the high-dimensional setting allow us to unveil a few interesting phenomena, namely: $(i)$ Biasing data selection towards hard samples (those that are uncertain under the surrogate model) can be suboptimal; $(ii)$ Even when biasing towards hard samples is effective, selecting the top hardest one can lead to poor behavior at small $\gamma$; $(iii)$ A one parameter family of selection probabilities introduced in the next section is broadly effective.

### 5.1 SETTING

Since our focus is on the surrogate model, we consider a data distribution in which the covariates carry little or no information about the value of a sample. We assume $\boldsymbol{x}_i \sim \mathsf{N}(0, \boldsymbol{I}_p)$, and (potentially misspecified) responses $y_i$ that depend on a one-dimensional projection of the data, as per Eq. (4.8). This includes the general misspecified binary response model: $y_i \in \{+1, -1\}$ and $\mathbb{P}(y_i = +1 | \boldsymbol{x}_i) = f(\langle \boldsymbol{\theta}_0, \boldsymbol{x}_i \rangle) = 1 - \mathbb{P}(y_i = -1 | \boldsymbol{x}_i)$.

We specialize the ERM procedure of Eq. (2.1) to

$$\hat{R}_N(\boldsymbol{\theta}) := \frac{1}{N} \sum_{i=1}^{N} S_i(\langle \hat{\boldsymbol{\theta}}^{\text{su}}, \boldsymbol{x}_i \rangle) \, L(\langle \boldsymbol{\theta}, \boldsymbol{x}_i \rangle, y_i) + \frac{\lambda}{2} \|\boldsymbol{\theta}\|_2^2. \tag{5.1}$$

where $L : \mathbb{R} \times \mathbb{R} \to \mathbb{R}$ is a loss function (we abuse notations here). This is a special case of Eq. (2.1) in two ways: *first,* the loss depends on $\boldsymbol{\theta}$, $\boldsymbol{x}_i$ only through $\langle \boldsymbol{\theta}, \boldsymbol{x}_i \rangle$ and, *second,* we focus on a ridge regularizer. Our characterization will require $L$ to be convex in its first argument. As before, it is understood that $S_i(\langle \hat{\boldsymbol{\theta}}^{\text{su}}, \boldsymbol{x}_i \rangle)$ depends on some additional i.i.d. randomness. Denoting by $\hat{\boldsymbol{\theta}}_{\lambda} := \arg \min_{\boldsymbol{\theta}} \hat{R}_N(\boldsymbol{\theta})$, we will consider a test error of the form $R_{\text{test}}(\hat{\boldsymbol{\theta}}_{\lambda}) = \mathbb{E} L_{\text{test}}(\langle \hat{\boldsymbol{\theta}}_{\lambda}, \boldsymbol{x}_{\text{new}} \rangle, y_{\text{new}})$.

Because of the rotational invariance the behavior of the empirical risk minimization (5.1) depends on the surrogate model only through two parameters[3] ($\boldsymbol{P}_0^{\perp}$ is the projector orthogonal to $\boldsymbol{\theta}_0$):

$$\beta_0 := \lim_{N, p \to \infty} \frac{\langle \hat{\boldsymbol{\theta}}^{\text{su}}, \boldsymbol{\theta}_0 \rangle}{\|\boldsymbol{\theta}_0\|}, \qquad \beta_s := \lim_{N, p \to \infty} \|\boldsymbol{P}_0^{\perp} \hat{\boldsymbol{\theta}}^{\text{su}}\|_2. \tag{5.2}$$

---

[3]The population risk minimizer $\boldsymbol{\theta}_*$ does not coincide necessarily with $\boldsymbol{\theta}_0$. We have $\boldsymbol{\theta}_* = c_* \boldsymbol{\theta}_0 / \|\boldsymbol{\theta}_0\|$.

## 5.2 ASYMPTOTICS OF THE ESTIMATION ERROR

The high-dimensional asymptotics of the test error is determined by a saddle point of the following Lagrangian (here and below $\boldsymbol{\alpha} := (\alpha_0, \alpha_s, \alpha_\perp)$, $\boldsymbol{\beta} := (\beta_0, \beta_s, 0)$, $\boldsymbol{\alpha}_{0,s} := (\alpha_0, \alpha_s, 0)$):

$$\mathscr{L}(\boldsymbol{\alpha}, \mu, \omega) := \frac{\lambda}{2}\|\boldsymbol{\alpha}\|^2 - \frac{\mu\alpha_\perp^2}{2\delta_0} + \mathbb{E}\Big\{ \min_{u\in\mathbb{R}} \Big[ S(\langle\boldsymbol{\beta}, \boldsymbol{G}\rangle) \, L(\langle\boldsymbol{\alpha}_{0,s}\boldsymbol{G}\rangle + u, Y) + \frac{1}{2}\mu(\alpha_\perp G_\perp - u)^2 \Big] \Big\},$$
$$\boldsymbol{G} = (G_0, G_s, G_\perp) \sim \mathsf{N}(\mathbf{0}, \boldsymbol{I}_3), \qquad Y \sim \mathsf{P}(\,\cdot\,|\,\|\boldsymbol{\theta}_0\|_2 G_0)\,. \tag{5.3}$$

(Expectation with respect to the randomness in $S$ is implicit in the symbol $\mathbb{E}$.)

**Theorem 4.** *Assume $u \mapsto L(u, y)$ is convex, continuous, with at most quadratic growth, and $\lambda > 0$. Further denote by $\boldsymbol{\alpha}^*, \mu^*$ the solution of the following minimax problem ($\boldsymbol{\alpha}^*$ is uniquely defined by this condition)*

$$\min_{\boldsymbol{\alpha}} \max_{\mu\geq 0} \mathscr{L}(\boldsymbol{\alpha}, \mu)\,. \tag{5.4}$$

*Then the following hold in the limit $N, p \to \infty$, with $N/p \to \delta_0$: If $(u, z) \mapsto \int L_{\text{test}}(u; y)\mathsf{P}(\mathrm{d}y|z)$ is a continuous function with at most quadratic growth, we have*

$$\underset{N,p\to\infty}{\text{p-lim}} \, R_{\text{test}}(\hat{\boldsymbol{\theta}}_\lambda) = \mathbb{E}L_{\text{test}}\Big(\alpha_0^* G_0 + \sqrt{(\alpha_s^*)^2 + (\alpha_\perp^*)^2}G; Y\Big)\,, \tag{5.5}$$

*where expectation is taken with respect to the joint distribution of Eq. (5.3). Further, the asymptotic subsampling fraction is given by $n/N \to \gamma = \mathbb{P}(S(\beta_0 G_0 + \beta_s G_s) > 0)$.*

The proof of Theorem 4 is deferred to Appendix L, which also contains corollaries for special cases (see also Appendix M).

## 6 NUMERICAL RESULTS: SYNTHETIC DATA

In this section we present numerical simulations within the synthetic data model introduced in Section 5.1, for binary labels $y_i \in \{+1, -1\}$. Summarizing, we generate isotropic feature vectors $\boldsymbol{x}_i \sim \mathsf{N}(0, \boldsymbol{I}_p)$, and labels with $\mathbb{P}(y_i = +1|\boldsymbol{x}_i) = f(\langle\boldsymbol{\theta}_0, \boldsymbol{x}_i\rangle)$. We then perform data selection, with selection probability

$$\pi(\boldsymbol{x}_i) = \min\Big(c(\gamma)\,p(\langle\hat{\boldsymbol{\theta}}^{\text{su}}, \boldsymbol{x}_i\rangle)^\alpha(1 - p(\langle\hat{\boldsymbol{\theta}}^{\text{su}}, \boldsymbol{x}_i\rangle))^\alpha;\; 1\Big)\,, \tag{6.1}$$

where $p(\langle\hat{\boldsymbol{\theta}}^{\text{su}}, \boldsymbol{x}_i\rangle) = (1 + e^{-2\langle\hat{\boldsymbol{\theta}}^{\text{su}}, \boldsymbol{x}_i\rangle})^{-1}$ is the probability of $y_i = +1$ under the surrogate model. We fix the constant $c(\gamma)$ so that $\sum_{i\leq N} \pi(\boldsymbol{x}_i) = n$.

Hence, $\alpha > 0$ upsamples data points with higher uncertainty under the surrogate model ('difficult' data), while $\alpha < 0$ upsamples data points with lower uncertainty ('easy' data). We then fit ridge regularized logistic regression to the selected data (cf. Eq. (5.1)) and evaluate misclassification error on a hold-out test set. For large-dimension $d$, influence-function based subsampling essentially corresponds to the case $\alpha = 1/2$ of Eq. (6.1) (see Appendix N for proof).

Figure 2 compare the results of simulations (circles) and theoretical predictions (continuous lines) for different $\delta_0, \lambda$. We consider two misspecified models defined by (here $\eta = 0.95 = 1 - \bar{\eta}$, $\zeta = 0.7 = 1 - \bar{\zeta}$): $f$:

$$f_1(z) := \eta\mathbf{1}_{z\geq 0} + \bar{\eta}\mathbf{1}_{z<0}\,, \quad f_2(z) := \bar{\zeta}\mathbf{1}_{z<-0.5} + \bar{\eta}\mathbf{1}_{(-0.5\leq z<0)} + \eta\mathbf{1}_{(0\leq z<0.5)} + \zeta\mathbf{1}_{z\geq 0.5}\,.$$

Here we use an ideal surrogate: $\hat{\boldsymbol{\theta}}^{\text{su}} = \boldsymbol{\theta}_0$. In Appendix N we report similar plots both for well-specified and misspecified models, exploring the effect of signal strength and of imperfect surrogates (cf. Figure 5, 6, 7) [4].

The agreement between theoretical predictions and simulation results is excellent in all the experiments. Theory correctly predicts that, for a suitable choice of $\alpha$, the non-reweighted data selection scheme of Eq. (6.1) achieves nearly identical test error as full data, while using as few as $40\%$ of the

---

[4] The code to reproduce theory and simulations with synthetic data can be found at the github repo.

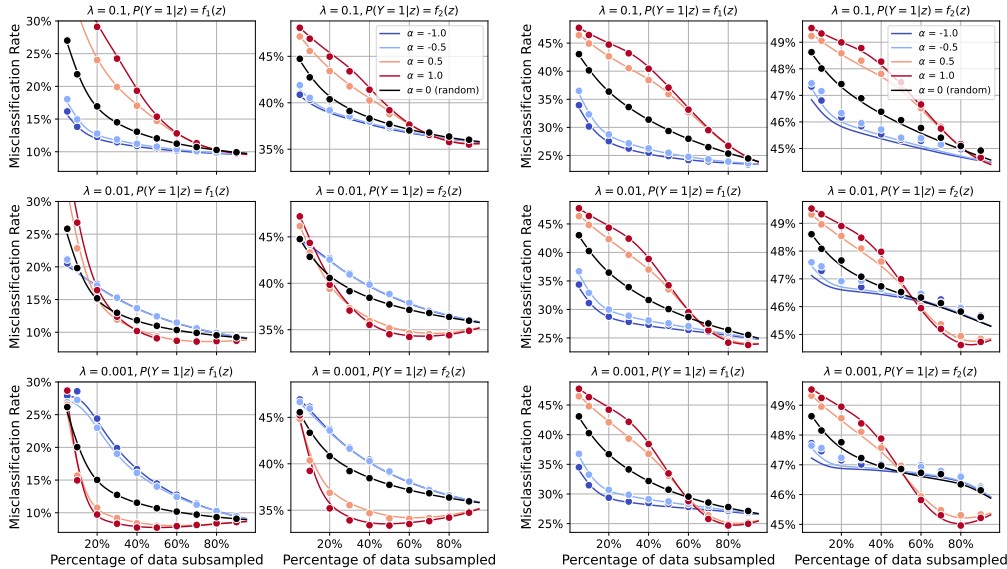

Figure 2: Misclassification error for logistic regression after non-reweighted subsampling, cf. Eq. (6.1), under misspecified model $\mathbb{P}\big(y = 1 \big| z\big) := f(z)$, where $z = \langle \boldsymbol{\theta}_0, \boldsymbol{x} \rangle$. Circles are results of numerical simulations, and continuous lines are theoretical predictions. Colors represent different values of the exponent $\alpha$ in Eq. (6.1). Left plot: $N = 34345$, $p = 932$; Right plot: $N = 6870$, $p = 3000$. Each panel corresponds to a different choice of $f$, $\lambda$. (See main text.)

samples. The optimal choice of $\alpha$ is highly dependent on the setting, with $\alpha < 0$ broadly outperforming $\alpha > 0$ when the number of samples per dimension is smaller (Figure 6). Also, we observe qualitatively different behaviors for misspecified model. Notably: $(i)$ Learning after data selection often outperforms learning on the full sample; $(ii)$ Upsampling 'hard' datapoints (i.e. using $\alpha > 0$) is most often the optimal strategy with respect to the well-specified case. $(iii)$ Upsampling 'easy' datapoints can be beneficial in the small $\gamma$ regime; $(iv)$ For small $\lambda$ and certain choices of $\delta_0, \alpha$ we observe interesting non-monotonicities of error: smaller sample sizes lead to lower misclassification error. This is related to suboptimality of that value of $\alpha$ and scheme (6.1) (see Section 4.3).

# 7 NUMERICAL RESULTS: REAL DATA

For our real-data experiments we used an Autonomous Vehicle (AV) dataset and a binary classification task. As for synthetic data, we use ridge regularized logistic regression and study data selection under the one-parameter scheme of Eq. (6.1). This model was trained and tested on unsupervised features extracted from image data. Appendix O provides the details of the dataset and experiments, and further empirical results supporting the claims.

Figure 1 (left) shows test error for optimal $\lambda$ and $\alpha = 0.5$ fixed, for unbiased and non-reweighted (biased) subsampling, using 'weak' and 'strong' surrogate models. Note that: $(i)$ Restricting to 'unbiased' data selection is unnecessary and sometimes harmful; $(ii)$ The efficacy of weakly supervised data-selection methods is not highly sensitive to the quality of the surrogate (see also Figure 9 in Appendix O).

Figure 1 (right) shows test error for the optimal parameters $\{\lambda, \alpha, \mathsf{surr}, \mathsf{bias}\}$, selected by minimizing the misclassification rate on the validation set. The reported results are computed on the test set. We compare this optimal choice by a 'constant strategy' that uses $\lambda = 0.01$, $\alpha = 0.5$, non-reweighted subsampling and weak surrogate models. This constant strategy performs almost optimally when $N$ is large, and still provides consistent improvements over random subsampling when $N$ is smaller. This strategy reflects influence-function based subsampling, although without reweighting and using a weak surrogate model. This figure highlights that: $(i)$ Selection criteria based on the 'uncertainty' associated to the label of a data point are effective, however influence function can be suboptimal; $(ii)$ Learning after data selection can outperform learning on the full sample.

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

# Appendix

## Table of Contents

## A  STANDARD NOTATIONS

The unit sphere in $d$ dimensions is denoted by $\mathbb{S}^{d-1}$. Given two symmetric matrices $\boldsymbol{A}, \boldsymbol{B} \in \mathbb{R}^{d \times d}$, we write $\boldsymbol{A} \succeq \boldsymbol{B}$ if $\boldsymbol{A} - \boldsymbol{B}$ is positive semidefinite, and $\boldsymbol{A} \succ \boldsymbol{B}$ if $\boldsymbol{A} - \boldsymbol{B}$ is strictly positive definite. We denote by p-$\lim_{n \to \infty} X_n$ the limit in probability of a sequence of random variables $(X_n)_{n \geq 1}$. We use $O_P(\,\cdot\,)$, $o_P(\,\cdot\,)$ and so on for the standard Oh-notation in probability. For instance, given a sequence of random variables $X_N$, and deterministic quantities $b_N$, we have

$$X_N = o_P(b_N) \quad \Leftrightarrow \quad \underset{N \to \infty}{\text{p-lim}} \frac{|X_N|}{b_N} = 0\,. \tag{A.1}$$

## B  LOW-DIMENSIONAL GENERAL ASYMPTOTICS: IDEAL SURROGATE

Recall, the asymptotics of $\hat{\boldsymbol{\theta}}$ depends on the population risk associated to sampling scheme $S$ (that is the expectation of the empirical risk (2.1)):

$$R_S(\boldsymbol{\theta}) := \mathbb{E}\big\{ S(\boldsymbol{x})\, L(\boldsymbol{\theta}; y, \boldsymbol{x}) \big\}\,. \tag{B.1}$$

(In this notation, the argument $S$ indicates the dependence on the *function* that defines the sub-sampling procedure.) The conditional gradient covariance, and conditional Hessian will play an important role, and are defined below:

$$\boldsymbol{G}(\boldsymbol{x}) := \mathbb{E}\big\{ \nabla_{\boldsymbol{\theta}} L(\boldsymbol{\theta}_*; y, \boldsymbol{x}) \nabla_{\boldsymbol{\theta}} L(\boldsymbol{\theta}_*; y, \boldsymbol{x})^{\mathsf{T}} | \boldsymbol{x} \big\}\,, \qquad \boldsymbol{H}(\boldsymbol{x}) := \mathbb{E}\big\{ \nabla_{\boldsymbol{\theta}}^2 L(\boldsymbol{\theta}_*; y, \boldsymbol{x}) | \boldsymbol{x} \big\}\,. \tag{B.2}$$

Our result characterizes the error under weighted quadratic losses $\|\hat{\boldsymbol{\theta}} - \boldsymbol{\theta}_*\|_{\boldsymbol{Q}}^2 := \langle \hat{\boldsymbol{\theta}} - \boldsymbol{\theta}_*, \boldsymbol{Q}(\hat{\boldsymbol{\theta}} - \boldsymbol{\theta}_*) \rangle$, for $\boldsymbol{Q} \in \mathbb{R}^{p \times p}$. This covers both the standard $\ell_2$ estimation error (by setting $\boldsymbol{Q} = \boldsymbol{I}$) and, under smoothness conditions, the test error $R_{\text{test}}(\hat{\boldsymbol{\theta}})$.

We define the asymptotic error coefficient via

$$\rho(S; \boldsymbol{Q}) := \underset{N \to \infty}{\text{p-lim}}\, N \|\hat{\boldsymbol{\theta}} - \boldsymbol{\theta}_*\|_{\boldsymbol{Q}}^2\,, \tag{B.3}$$

whenever the limit exists. The next result is an application of textbook asymptotic statistics, as detailed below.

**Proposition B.1.** *Assume the following:*

A1. *$R_S(\boldsymbol{\theta})$ is uniquely minimized at $\boldsymbol{\theta} = \boldsymbol{\theta}_*$.*

A2. *$\boldsymbol{\theta} \mapsto L(\boldsymbol{\theta}; y, \boldsymbol{x})$ is non-negative, lower semicontinuous. Further, for every $\boldsymbol{u} \in \mathbb{S}^{p-1}$, define $L_\infty(\boldsymbol{u}; y, \boldsymbol{x}) \in [0, \infty]$*

$$L_\infty(\boldsymbol{u}; y, \boldsymbol{x}) = \underset{\substack{\boldsymbol{\theta} \to +\infty \\ \boldsymbol{\theta}/\|\boldsymbol{\theta}\| \to \boldsymbol{u}}}{\liminf}\, L(\boldsymbol{\theta}; y, \boldsymbol{x})\,, \tag{B.4}$$

*and assume $\inf_{\boldsymbol{u} \in \mathbb{S}^{p-1}} \mathbb{E}\big\{ S(\boldsymbol{x}) L_\infty(\boldsymbol{u}; y, \boldsymbol{x}) \big\} > R_S(\boldsymbol{\theta}_*)$.*

A3. *$\boldsymbol{\theta} \mapsto L(\boldsymbol{\theta}; y, \boldsymbol{x})$ is differentiable at $\boldsymbol{\theta}_*$ for $\mathbb{P}$-almost every $(y, \boldsymbol{x})$. Further, for $B$ a neighborhood of $\boldsymbol{\theta}_*$,*

$$\mathbb{E} \sup_{\boldsymbol{\theta}_1 \neq \boldsymbol{\theta}_2 \in B} \left\{ S(\boldsymbol{x})^2 \frac{|L(\boldsymbol{\theta}_1; y, \boldsymbol{x}) - L(\boldsymbol{\theta}_2; y, \boldsymbol{x})|^2}{\|\boldsymbol{\theta}_1 - \boldsymbol{\theta}_2\|_2^2} \right\} < \infty$$

A4. *$\boldsymbol{\theta} \mapsto R_S(\boldsymbol{\theta})$ is twice differentiable at $\boldsymbol{\theta}_*$, with $\nabla^2 R_S(\boldsymbol{\theta}_*) = \mathbb{E}\{ S(\boldsymbol{x}) \boldsymbol{H}(\boldsymbol{x}) \} \succ \boldsymbol{0}$.*

*Then, for any $\boldsymbol{Q} \in \mathbb{R}^{p \times p}$ symmetric, the limit of Eq.* (B.3) *exists and is given by*

$$\rho(S; \boldsymbol{Q}) = \frac{\mathbb{E}\{ S(\boldsymbol{x})^2 \}}{\mathbb{E}\{ S(\boldsymbol{x}) \}^2} \,\mathsf{Tr}\big( \boldsymbol{G}_S \boldsymbol{H}_S^{-1} \boldsymbol{Q} \boldsymbol{H}_S^{-1} \big)\,, \tag{B.5}$$

*where*

$$\boldsymbol{G}_S := \frac{\mathbb{E}\{S(\boldsymbol{x})^2 \, \boldsymbol{G}(\boldsymbol{x})\}}{\mathbb{E}\{S(\boldsymbol{x})^2\}} \, , \qquad \boldsymbol{H}_S := \frac{\mathbb{E}\{S(\boldsymbol{x}) \, \boldsymbol{H}(\boldsymbol{x})\}}{\mathbb{E}\{S(\boldsymbol{x})\}} \, . \tag{B.6}$$

*In particular, if $\boldsymbol{\theta} \mapsto R_{\text{test}}(\boldsymbol{\theta})$ is twice continuously differentiable at $\boldsymbol{\theta}_*$, with $\nabla R_{\text{test}}(\boldsymbol{\theta}_*) = \boldsymbol{0}$, then*

$$R_{\text{test}}(\hat{\boldsymbol{\theta}}) = R_{\text{test}}(\boldsymbol{\theta}_*) + \frac{1}{N} \cdot \rho(S; \boldsymbol{Q}_{\text{test}}) + o_P(1/N) \, , \tag{B.7}$$

$$\boldsymbol{Q}_{\text{test}} := \frac{1}{2} \nabla^2 R_{\text{test}}(\boldsymbol{\theta}_*) \, . \tag{B.8}$$

**Remark B.1.** Key for Proposition B.1 to hold is condition A1, which requires the minimizer of the subsampled population risk $R_S$ to coincide with the minimizer of the original risk $R$. This amounts to say that the data selection scheme is not so biased as to make empirical risk minimization inconsistent. If it does not hold, then the resulting error $\|\hat{\boldsymbol{\theta}} - \boldsymbol{\theta}_*\|_{\boldsymbol{Q}}^2$ will be, in general, of order one.

**Proof of Proposition B.1** Write $S_i(\boldsymbol{x}_i) = s(\boldsymbol{x}_i, U_i)$ where $(U_i)_{i \leq N} \sim_{iid} \mathsf{Unif}([0,1])$ are the independent random seed used to compute $S_i$ (we omit the dependence on the surrogate model). For $\|\boldsymbol{\xi}\| < 1$, define $\boldsymbol{\theta}(\boldsymbol{\xi}) = c(\|\boldsymbol{\xi}\|) \, \boldsymbol{\xi}$, where $c(r) = 1/(1 - r^2)$, and, with an abuse of notation, $R_S(\boldsymbol{\xi}) = R_S(\boldsymbol{\theta}(\boldsymbol{\xi}))$. Finally

$$L_S(\boldsymbol{\xi}; Z_i) := \begin{cases} s(\boldsymbol{x}_i, U_i) \, L(\boldsymbol{\theta}(\boldsymbol{\xi}); y_i, \boldsymbol{x}_i) & \text{if } \|\boldsymbol{\xi}\| < 1, \\ s(\boldsymbol{x}_i, U_i) \, L_\infty(\boldsymbol{\xi}; y_i, \boldsymbol{x}_i) & \text{if } \|\boldsymbol{\xi}\| = 1, \end{cases} \tag{B.9}$$

$$Z_i := \big(U_i, y_i, \boldsymbol{x}_i\big) \, , \tag{B.10}$$

so that $R_S(\boldsymbol{\xi}) = \mathbb{E}L_S(\boldsymbol{\xi}; Z)$ and $\hat{R}_N(\boldsymbol{\xi}) = N^{-1} \sum_{i=1}^N L_S(\boldsymbol{\xi}; Z_i)$ are defined for $\boldsymbol{\xi} \in \mathsf{B}^p(1)$ (the unit ball in $\mathbb{R}^p$). If $\hat{\boldsymbol{\xi}} := \arg\min_{\boldsymbol{\xi} \in \mathsf{B}^p(1)} \hat{R}_N(\boldsymbol{\xi})$, and $\boldsymbol{\xi}_* := \arg\min_{\boldsymbol{\xi} \in \mathsf{B}^p(1)} R_S(\boldsymbol{\xi})$ (the latter is unique by Assumption A1), by van der Vaart (2000, Theorem 5.13), we have $\hat{\boldsymbol{\xi}} \to \boldsymbol{\xi}_*$ almost surely. By Assumption A2, we further have $\|\boldsymbol{\xi}\| < 1$ strictly. Therefore, almost surely, $\hat{\boldsymbol{\theta}} \to \boldsymbol{\theta}_* = \boldsymbol{\theta}(\boldsymbol{\xi}_*)$.

Using van der Vaart (2000, Theorem 5.14) (whose assumptions follow from A3, A4), we get

$$\hat{\boldsymbol{\theta}} - \boldsymbol{\theta}_* = \frac{1}{N} \boldsymbol{H}_S^{-1} \sum_{i=1}^N \nabla_{\boldsymbol{\theta}} L_S(\boldsymbol{\theta}_*; Z_i) + o_P(N^{-1/2}) \, , \tag{B.11}$$

whence

$$\rho(S; \boldsymbol{Q}) = \mathbb{E} \, \mathsf{Tr}\Big( \nabla_{\boldsymbol{\theta}} L_S(\boldsymbol{\theta}_*; Z_1) \nabla_{\boldsymbol{\theta}} L_S(\boldsymbol{\theta}_*; Z_1)^\mathsf{T} \boldsymbol{H}_S^{-1} \boldsymbol{Q} \boldsymbol{H}_S^{-1} \Big) \, . \tag{B.12}$$

The claim follows simply by substituting the expression for $\nabla_{\boldsymbol{\theta}} L_S(\boldsymbol{\theta}_*; Z_1)$.

## C  PROOF OF PROPOSITION 4.1

As mentioned in the main text, this result (in slightly different form) appears already in the literature (Ting & Brochu, 2018; Wang et al., 2018; Ai et al., 2021). We nevertheless present a proof for the reader's convenience.

First of all notice that, for $S$ unbiased we have $\mathbb{E}\{S(\boldsymbol{x})|\boldsymbol{x}\} = 1$ and therefore $\boldsymbol{H}_S = \boldsymbol{H}$. Eq. (B.5) yields

$$\rho(S; \boldsymbol{Q}) = \mathbb{E}\{S(\boldsymbol{x})^2 Z(\boldsymbol{x})\} \, , \quad Z(\boldsymbol{x}) = \mathsf{Tr}\big( \boldsymbol{G}(\boldsymbol{x}) \boldsymbol{H}^{-1} \boldsymbol{Q} \boldsymbol{H}^{-1} \big) \, ,$$

We can always write $S(\boldsymbol{x}) = S_+(\boldsymbol{x}) \, I(\boldsymbol{x})$, where $S_+(\boldsymbol{x}) > 0$ almost surely and, conditionally on $\boldsymbol{x}$, $S_+(\boldsymbol{x})$ is independent of $I(\boldsymbol{x}) \in \{0,1\}$ with $\mathbb{P}(I(\boldsymbol{x}) = 1|\boldsymbol{x}) = \pi(\boldsymbol{x})$. The unbiasedness constraint translates into $\mathbb{E}(S_+(\boldsymbol{x})|\boldsymbol{x}) = 1/\pi(\boldsymbol{x})$. Hence

$$\begin{aligned} \rho(S; \boldsymbol{Q}) &= \mathbb{E}\{S_+(\boldsymbol{x})^2 \pi(\boldsymbol{x}) Z(\boldsymbol{x})\} \\ &\geq \mathbb{E}\{\mathbb{E}\{S_+(\boldsymbol{x})|\boldsymbol{x}\}^2 \pi(\boldsymbol{x}) Z(\boldsymbol{x})\} \\ &= \mathbb{E}\Big\{ \frac{Z(\boldsymbol{x})}{\pi(\boldsymbol{x})} \Big\} \, , \end{aligned}$$

where the lower bound holds with equality if and only if $S_+(\boldsymbol{x}) = 1/\pi(\boldsymbol{x})$ almost surely.

The optimal $\pi$ is determined by the convex optimization problem

$$\text{minimize} \quad \mathbb{E}\Big\{ \frac{Z(\boldsymbol{x})}{\pi(\boldsymbol{x})} \Big\}, \tag{C.1}$$

$$\text{subj. to} \quad \mathbb{E}\{\pi(\boldsymbol{x})\} = \gamma, \tag{C.2}$$

$$\pi(\boldsymbol{x}) \in [0,1] \; \forall \boldsymbol{x}. \tag{C.3}$$

By duality, there exists a constant $\lambda = \lambda(\gamma)$, such that the optimum of the above problem is the solution of

$$\text{minimize} \quad \mathbb{E}\Big\{ \frac{Z(\boldsymbol{x})}{\pi(\boldsymbol{x})} - \lambda\,\pi(\boldsymbol{x}) \Big\}, \tag{C.4}$$

$$\text{subj. to} \quad \pi(\boldsymbol{x}) \in [0,1] \quad \forall \boldsymbol{x}. \tag{C.5}$$

This yields the claimed optimum $\pi_{\text{unb}}$.

**Remark C.1.** Besides computing expectations with respect to the conditional distribution of $y$ given $\boldsymbol{x}$, evaluating the score $Z$ requires to be able to approximate $\boldsymbol{H} = \mathbb{E}\{\boldsymbol{H}(\boldsymbol{x})\}$. However, this can be consistently estimated e.g. using the empirical mean $\hat{\boldsymbol{H}} := N^{-1} \sum_{i=1}^{N} \boldsymbol{H}(\boldsymbol{x}_i)$. We will neglect errors involved in such approximations.

## D  PROOF OF PROPOSITION 4.2

For claim (1), it is easy to compute $\rho_{\text{rand}}(\boldsymbol{Q}) = \gamma^{-1}\mathbb{E}(Z)$. Therefore, the gain derived from optimal subsampling can be written as

$$\frac{\rho_{\text{unb}}(\boldsymbol{Q})}{\rho_{\text{rand}}(\boldsymbol{Q})} = \frac{\mathbb{E}(Z/\pi_{\text{unb}})\mathbb{E}(\pi_{\text{unb}})}{\mathbb{E}((Z/\pi_{\text{unb}}) \cdot \pi_{\text{unb}})} \le 1. \tag{D.1}$$

The inequality above is due to the fact that $\pi_{\text{unb}}$ and $Z/\pi_{\text{unb}}$ are both non-decreasing functions of $Z$. It is strict unless the following happens with probability one, for i.i.d. copies $Z_1, Z_2$ of $Z$: $\min(1; cZ_1^{1/2}) = \min(1; cZ_2^{1/2})$ or $\max(Z_1; Z_1^{1/2}/c) = \max(Z_2; Z_2^{1/2}/c)$ However, this happens only if either $Z$ is almost surely a constant or if $cZ \le 1$ almost surely. The latter implies $\gamma = 1$.

To prove claim (2), note that the mapping $\pi \mapsto \mathbb{E}\{Z(\boldsymbol{x})/\pi(\boldsymbol{x})\}$ appearing in Eq. (4.5) is monotone with respect to the partial ordering of pointwise domination. Therefore, we can replace the constraint $\mathbb{E}\pi(\boldsymbol{x}) = \gamma$ by $\mathbb{E}\pi(\boldsymbol{x}) \le \gamma$, whence monotonicity trivially follows.

## E  BIASED DATA SELECTION: DEFINITIONS

Among data selection schemes, a special role is played by those schemes in which the only randomness is the choice of whether or not to select a certain sample $i$. We refer to these schemes as 'simple.'

**Definition E.1** (Simple data selection schemes). *A scheme is* simple *if there exist function $w, \pi$ such that*

$$S(\boldsymbol{x}) = \begin{cases} w(\boldsymbol{x}) & \text{with probability } \pi(\boldsymbol{x}), \\ 0 & \text{otherwise}, \end{cases} \tag{E.1}$$

*We denote the set of simple sampling schemes by $\mathscr{S} \subseteq \mathscr{A}$, where $\mathscr{A}$ is the set of all data selection schemes.*

As a reminder, we can identify any simple data selection scheme with the pair $(\pi, w)$, hence we will occasionally abuse notation and write $\rho(\pi, w; \boldsymbol{Q})$ for $\rho(S; \boldsymbol{Q})$.

**Lemma E.2.** *For $\boldsymbol{Q} \succeq 0$, we have:* $\inf_{S \in \mathscr{A}} \rho(S; \boldsymbol{Q}) = \inf_{S \in \mathscr{S}} \rho(S; \boldsymbol{Q})$.

$$\inf_{S \in \mathscr{A}} \rho(S; \boldsymbol{Q}) = \inf_{S \in \mathscr{S}} \rho(S; \boldsymbol{Q}). \tag{E.2}$$

We can therefore restrict ourselves to simple schemes without loss.

**Proof of Lemma E.2** As in the previous section, we can always $S(\boldsymbol{x}) = S_+(\boldsymbol{x}) \, I(\boldsymbol{x})$, where $S_+(\boldsymbol{x}) > 0$ almost surely and, conditionally on $\boldsymbol{x}$, $S_+(\boldsymbol{x})$ is independent of $I(\boldsymbol{x}) \in \{0, 1\}$. Further

$$\mathbb{P}(I(\boldsymbol{x}) = 1 | \boldsymbol{x}) = \pi(\boldsymbol{x}), \tag{E.3}$$

$$\mathbb{E}(S_+(\boldsymbol{x}) | \boldsymbol{x}) = w(\boldsymbol{x}). \tag{E.4}$$

Simple schemes correspond to the case in which $S_+(\boldsymbol{x}) = w(\boldsymbol{x})$ is non-random

Recall the formula (B.5) for the asymptotic error coefficient $\rho(S; \boldsymbol{Q})$, which we rewrite here as

$$\rho(S; \boldsymbol{Q}) = \mathsf{Tr}\Big(\mathbb{E}\big\{S_+^2(\boldsymbol{x})\pi(\boldsymbol{x})\boldsymbol{G}(\boldsymbol{x})\big\}\tilde{\boldsymbol{H}}_{w,\pi}^{-1}\boldsymbol{Q}\tilde{\boldsymbol{H}}_{w,\pi}^{-1}\Big), \tag{E.5}$$

$$\tilde{\boldsymbol{H}}_{w,\pi} := \mathbb{E}\big\{w(\boldsymbol{x})\pi(\boldsymbol{x})\boldsymbol{H}(\boldsymbol{x})\big\}. \tag{E.6}$$

By Jensen inequality (using the fact that $\boldsymbol{G}(\boldsymbol{x}), \tilde{\boldsymbol{H}}_{w,\pi}, \boldsymbol{Q} \succeq \boldsymbol{0}$), we get

$$\rho(S; \boldsymbol{Q}) \geq \mathsf{Tr}\Big(\mathbb{E}\big\{w(\boldsymbol{x})^2\pi(\boldsymbol{x})\boldsymbol{G}(\boldsymbol{x})\big\}\tilde{\boldsymbol{H}}_{w,\pi}^{-1}\boldsymbol{Q}\tilde{\boldsymbol{H}}_{w,\pi}^{-1}\Big), \tag{E.7}$$

and simply note that the right hand side is achieved by the simple scheme.

# F    PROOF OF PROPOSITION 4.3

Recall the general formula (B.5) for the asymptotic error coefficient $\rho(S; \boldsymbol{Q})$. For a non-reweighting scheme with selection probability $\pi$, with an abuse of notation we write $\rho(S; \boldsymbol{Q})$ as $\rho(\pi; \boldsymbol{Q})$ (we also used $\rho(\pi, 1; \boldsymbol{Q})$ in the main text). Explicitly

$$\rho(\pi; \boldsymbol{Q}) = \mathsf{Tr}\Big(\mathbb{E}\big\{\pi(\boldsymbol{x})\boldsymbol{G}(\boldsymbol{x})\big\}\mathbb{E}\big\{\pi(\boldsymbol{x})\boldsymbol{H}(\boldsymbol{x})\big\}^{-1}\boldsymbol{Q}\mathbb{E}\big\{\pi(\boldsymbol{x})\boldsymbol{H}(\boldsymbol{x})\big\}^{-1}\Big). \tag{F.1}$$

Notice that this is defined for $\mathbb{E}\big\{\pi(\boldsymbol{x})\boldsymbol{H}(\boldsymbol{x})\big\} \succ \boldsymbol{0}$. We extent it to $\mathbb{E}\big\{\pi(\boldsymbol{x})\boldsymbol{H}(\boldsymbol{x})\big\} \succeq \boldsymbol{0}$ by letting

$$\rho(\pi; \boldsymbol{Q}) = \lim_{\lambda \to 0+} \mathsf{Tr}\Big(\mathbb{E}\big\{\pi(\boldsymbol{x})\boldsymbol{G}(\boldsymbol{x})\big\}\big(\lambda\boldsymbol{I} + \mathbb{E}\big\{\pi(\boldsymbol{x})\boldsymbol{H}(\boldsymbol{x})\big\}\big)^{-1}\boldsymbol{Q}\big(\lambda\boldsymbol{I} + \mathbb{E}\big\{\pi(\boldsymbol{x})\boldsymbol{H}(\boldsymbol{x})\big\}\big)^{-1}\Big). \tag{F.2}$$

We want to minimize this function over $\pi$ subject to the convex constraints $\mathbb{E}\pi(\boldsymbol{x}) = \gamma$, $\pi(\boldsymbol{x}) \in [0, 1]$ for all $\boldsymbol{x}$.

We claim that a minimizer $\pi_{\mathrm{nr}}$ always exists. To this end, we view $\rho(\pi; \boldsymbol{Q}) = F(\nu)$ as a function of the probability measure $\nu(\mathrm{d}\boldsymbol{x}) = \pi(\boldsymbol{x})\mathbb{P}(\mathrm{d}\boldsymbol{x})/\gamma$. In other words $F$ is a function of the space of probability measures, whose Radon-Nikodym derivative with respect to $\mathbb{P}$ is upper bounded by $1/\gamma$. This domain is uniformly tight. Further, if $\nu_n$ is a sequence in this space and $\nu_n \Rightarrow \nu_\infty$ (weak convergence), it follows by the Portmanteau's theorem that $\nu_\infty$ has also Radon-Nikodym derivative with respect to $\mathbb{P}$ that is upper bounded by $1/\gamma$. Hence this domain is compact by Prokhorov's theorem. Finally $\nu \mapsto \int \boldsymbol{G}(\boldsymbol{x}) \, \nu(\mathrm{d}\boldsymbol{x})$ and $\nu \mapsto \int \boldsymbol{H}(\boldsymbol{x}) \, \nu(\mathrm{d}\boldsymbol{x})$ are continuous in the topology of weak convergence (because $\boldsymbol{G}(\boldsymbol{x}), \boldsymbol{H}(\boldsymbol{x})$ are continuous by assumption), and therefore $\nu \mapsto F(\nu)$ is lower semi-continuous. Hence, there exists a minimizer $\nu_{\mathrm{nr}}(\mathrm{d}\boldsymbol{x}) = \pi_{\mathrm{nr}}(\boldsymbol{x})\mathbb{P}(\mathrm{d}\boldsymbol{x})/\gamma$, with $\pi_{\mathrm{nr}}(\boldsymbol{x}) \in [0, 1]$.

Given any minimizer $\pi_{\mathrm{nr}}$, and any other feasible $\pi$, let $\pi_t := (1 - t)\pi_{\mathrm{nr}} + t\pi$. By assumption $\boldsymbol{H}_{\pi_{\mathrm{nr}}} \succ \boldsymbol{0}$ strictly. Then

$$\rho(\pi_t; \boldsymbol{Q}) = \rho(\pi_{\mathrm{nr}}; \boldsymbol{Q}) + t \int \big(\pi(\boldsymbol{x}) - \pi_{\mathrm{nr}}(\boldsymbol{x})\big) Z(\boldsymbol{x}; \pi_{\mathrm{nr}}) \, \mathbb{P}(\mathrm{d}\boldsymbol{x}) + o(t). \tag{F.3}$$

Therefore it must be true that, for any feasible $\pi$,

$$J(\pi; \pi_{\mathrm{nr}}) := \int \big(\pi(\boldsymbol{x}) - \pi_{\mathrm{nr}}(\boldsymbol{x})\big) Z(\boldsymbol{x}; \pi_{\mathrm{nr}}) \, \mathbb{P}(\mathrm{d}\boldsymbol{x}) \geq 0. \tag{F.4}$$

Let $Q_\varepsilon := \{\boldsymbol{x} \in \mathbb{R}^d : \pi_{\mathrm{nr}}(\boldsymbol{x}) \in (\varepsilon, 1-\varepsilon)\}$. The claim (4.10) is is implied by the following statement: $Z(\boldsymbol{x}; \pi_{\mathrm{nr}})$ is almost surely constant on $Q_\varepsilon$ for each $\varepsilon > 0$. Assume by contradiction that there exists

$\varepsilon > 0$ and $z_0 \in \mathbb{R}$ such that $\mathbb{P}(\boldsymbol{x} \in Q_\varepsilon : Z(\boldsymbol{x}; \pi_{\mathrm{nr}}) \geq z_0) = p_+ > 0$, $\mathbb{P}(\boldsymbol{x} \in Q_\varepsilon : Z(\boldsymbol{x}; \pi_{\mathrm{nr}}) < z_0) = p_- > 0$. Let $Q_+ := \{\boldsymbol{x} \in Q_\varepsilon : Z(\boldsymbol{x}; \pi_{\mathrm{nr}}) \geq z_0\}$, $Q_- := \{\boldsymbol{x} \in Q_\varepsilon : Z(\boldsymbol{x}; \pi_{\mathrm{nr}}) < z_0\}$. Define

$$\pi(\boldsymbol{x}) = \begin{cases} \pi_{\mathrm{nr}}(\boldsymbol{x}) - p_-\varepsilon & \text{if } \boldsymbol{x} \in Q_+, \\ \pi_{\mathrm{nr}}(\boldsymbol{x}) + p_+\varepsilon & \text{if } \boldsymbol{x} \in Q_-, \\ \pi_{\mathrm{nr}}(\boldsymbol{x}) & \text{otherwise.} \end{cases}$$

It is easy to check that $\pi$ is feasible and

$$J(\pi; \pi_{\mathrm{nr}}) = -p_-\varepsilon \int_{Q_+} Z(\boldsymbol{x}; \pi_{\mathrm{nr}}) \, \mathbb{P}(\mathrm{d}\boldsymbol{x}) + p_+\varepsilon \int_{Q_-} Z(\boldsymbol{x}; \pi_{\mathrm{nr}}) \, \mathbb{P}(\mathrm{d}\boldsymbol{x})$$

$$= -p_+ p_-\varepsilon \Big\{ \mathbb{E}\big(Z(\boldsymbol{x}; \pi_{\mathrm{nr}})\big|\boldsymbol{x} \in Q_+\big) - \mathbb{E}\big(Z(\boldsymbol{x}; \pi_{\mathrm{nr}})\big|\boldsymbol{x} \in Q_-\big) \Big\} < 0,$$

thus yielding a contradiction with Eq. (F.4).

Finally, we note that the stated form of $\rho_{\mathrm{nr}}$ follows from Eq. (F.1).

**Remark F.1.** The condition $\mathbb{E}\{\nabla_{\boldsymbol{\theta}} L(\boldsymbol{\theta}_*; y, \boldsymbol{x})|\boldsymbol{x}\} = \boldsymbol{0}$ (almost surely) could be eliminated at the cost of enforcing the constraint $\mathbb{E}\{\nabla_{\boldsymbol{\theta}} L(\boldsymbol{\theta}_*; y, \boldsymbol{x})\pi(\boldsymbol{x})\} = \boldsymbol{0}$. This would result in a more complicated expression for the data selection rule: we will not pursue this generalization.

## G  DIFFERENCES BETWEEN UNBIASED AND BIASED DATA SELECTION: EXAMPLES

We provide concrete examples for optimal biased data selection schemes for two specific cases contrasting with optimal biased data selection schemes under low-dimensional asymptotics.

### G.1  LINEAR REGRESSION: PROOF OF THEOREM 1

Continuing from Example 4.3, note that the condition $\mathbb{E}\{\nabla_{\boldsymbol{\theta}} L(\boldsymbol{\theta}_*; y, \boldsymbol{x})|\boldsymbol{x}\} = \boldsymbol{0}$ holds. We now have $\boldsymbol{G}(\boldsymbol{x}) = \tau^2 \boldsymbol{x}\boldsymbol{x}^\mathsf{T}$, $\boldsymbol{H}(\boldsymbol{x}) = \boldsymbol{x}\boldsymbol{x}^\mathsf{T}$. Since rescaling $\boldsymbol{G}$ does not change the selection rule, we can redefine $\boldsymbol{G}(\boldsymbol{x}) = \boldsymbol{x}\boldsymbol{x}^\mathsf{T}$, whence $\boldsymbol{H}_\pi = \boldsymbol{G}_\pi = \boldsymbol{\Sigma}_\pi$, where $\boldsymbol{\Sigma}_\pi$ is the population covariance of the subsampled feature vectors.

A simple calculation shows

$$Z(\boldsymbol{x}; \pi) = \langle \boldsymbol{x}, \boldsymbol{\Sigma}_\pi^{-1} \boldsymbol{Q} \boldsymbol{\Sigma}_\pi^{-1} \boldsymbol{x} \rangle, \tag{G.1}$$

$$\pi_{\mathrm{nr}}(\boldsymbol{x}) = \begin{cases} 1 & \text{if } \langle \boldsymbol{x}, \boldsymbol{\Sigma}_{\pi_{\mathrm{nr}}}^{-1} \boldsymbol{Q} \boldsymbol{\Sigma}_{\pi_{\mathrm{nr}}}^{-1} \boldsymbol{x} \rangle > \lambda, \\ 0 & \text{if } \langle \boldsymbol{x}, \boldsymbol{\Sigma}_{\pi_{\mathrm{nr}}}^{-1} \boldsymbol{Q} \boldsymbol{\Sigma}_{\pi_{\mathrm{nr}}}^{-1} \boldsymbol{x} \rangle < \lambda. \end{cases} \tag{G.2}$$

In other words, this scheme selects all data that lay outside a certain ellipsoid. The shape of the ellipsoid is determined self-consistently by the covariance $\boldsymbol{\Sigma}_{\pi_{\mathrm{nr}}}$ of points outside the ellipsoid.

Let emphasize two differences with respect to the standard leverage-score approach of Section 4.1

$(i)$  As for general non-reweighing schemes, selection is essentially deterministic, $\pi_{\mathrm{nr}}(\boldsymbol{x}) \in \{0, 1\}$ (in particular, if the distribution of $\boldsymbol{x}$ has a density, then $\pi_{\mathrm{nr}}(\boldsymbol{x}) \in \{0, 1\}$ with probability one).

$(ii)$  The original covariance $\boldsymbol{\Sigma}$ is replaced by the covariance of selected data $\boldsymbol{\Sigma}_\pi$. As anticipated in Section 3, the selected set depends on $\gamma$ in a nontrivial way.

Given that the leverage score of datapoint $\boldsymbol{x}_i$ measures how different is $\boldsymbol{x}_i$ from the other data, the modified score of Eq. (G.1) can be interpreted as measuring how different is $\boldsymbol{x}_i$ from selected data.

**Example G.1** (One-dimensional covariates). In order to get a more concrete understanding of the difference with respect to unbiased data selection, consider one-dimensional covariates $x_i \sim \mathsf{P}_X$ with $\mathsf{P}_X$ of mean zero. We study prediction error, i.e. $\boldsymbol{Q} = \boldsymbol{\Sigma}$. Let $X_M$ be the supremum of the support of $\mathsf{P}_{|X|}$, which we assume finite, and $X$ a sample from $\mathsf{P}_X$. In this case $Z(x) = \tau^2 x^2/\Sigma$. Provided $\gamma \leq \mathbb{E}|X|/X_M$, we have $c(\gamma) = \gamma \mathbb{E}(X^2)^{1/2}/(\tau\mathbb{E}|X|)$ and the optimal asymptotic error for unbiased subsampling is

$$\rho_{\mathrm{unb}}(\Sigma) = \frac{\tau^2}{\gamma} \cdot \frac{(\mathbb{E}|X|)^2}{\mathbb{E}(X^2)}. \tag{G.3}$$

On the other hand, assuming $\mathsf{P}_X$ has a density, the optimal non-reweighting rule takes the form $\pi_*(x) = \mathbf{1}(|x| \geq r(\gamma))$. The coefficient $r = r(\gamma)$ is fixed by $\gamma = \mathbb{P}(|X| \geq r)$. The optimal asymptotic error for non-reweighting subsampling is

$$\rho_{\mathrm{nr}}(\Sigma) = \tau^2 \cdot \frac{\mathbb{E}(X^2)}{\mathbb{E}(X^2 \mathbf{1}_{|X| \geq r})} \,. \tag{G.4}$$

The ratio of biased to unbiased error then takes the form

$$\frac{\rho_{\mathrm{nr}}(\Sigma)}{\rho_{\mathrm{unb}}(\Sigma)} = \frac{\mathbb{E}(X^2)^2 \mathbb{P}(|X| \geq r)}{(\mathbb{E}|X|)^2 \mathbb{E}(X^2 \mathbf{1}_{|X| \geq r})} \,, \qquad \gamma \leq \frac{\mathbb{E}|X|}{X_M} \,. \tag{G.5}$$

It is easy to come up with examples in which this ratio is smaller than one. For instance if $\mathsf{P}_X = \mathsf{Unif}([-X_M, X_M])$, then

$$\frac{\rho_{\mathrm{nr}}(\Sigma)}{\rho_{\mathrm{unb}}(\Sigma)} = \frac{4\gamma}{3(1 - (1 - \gamma)^3)} \,, \qquad \gamma \leq \frac{1}{2} \,. \tag{G.6}$$

More generally, as $\gamma \to 0$, we get

$$\frac{\rho_{\mathrm{nr}}(\Sigma)}{\rho_{\mathrm{unb}}(\Sigma)} = \frac{\mathbb{E}(X^2)^2}{(\mathbb{E}|X|)^2 X_M^2} + o(1) \,, \quad \gamma \to 0, \tag{G.7}$$

Notice that this ratio is always smaller than one (by Hölder's inequality) and can be arbitrarily small. For instance $\mathsf{P}_X(\mathrm{d}x) = C_{M,\alpha}|x|^{-\alpha} \mathbf{1}_{1 \leq |x| \leq X_M}, \alpha > 3$ then the above ratio is

$$\frac{\rho_{\mathrm{nr}}(\Sigma)}{\rho_{\mathrm{unb}}(\Sigma)} = \left( \frac{\alpha - 2}{\alpha - 3} \cdot \frac{1 - X_M^{-\alpha+3}}{1 - X_M^{-\alpha+2}} \right)^2 \frac{1}{X_M^2} + o(1) \,. \tag{G.8}$$

To be concrete, for $\alpha = 4$, $X_M = 10$, unbiased is suboptimal by a factor larger than 30.

**Example G.2** (Elliptical covariates)**.** The one-dimensional example above is easily generalized to higher dimensions. Consider $\boldsymbol{x}_i = \boldsymbol{\Sigma}^{1/2} \boldsymbol{z}_i$ where $\boldsymbol{z}_i$ are spherically symmetric, namely $\boldsymbol{z}_i = r_i \boldsymbol{u}_i$ with $(r_i, \boldsymbol{u}_i) \sim \mathsf{P}_R \otimes \mathsf{Unif}(\mathbb{S}^{d-1}(\sqrt{d}))$. If we consider test error (and therefore $\boldsymbol{Q} = \boldsymbol{\Sigma}$), then it is a symmetry argument shows that, for optimal $\pi$ the modified leverage score (G.1) coincides with the original one

$$Z(\boldsymbol{x}; \pi_*) = \langle \boldsymbol{x}, \boldsymbol{\Sigma}^{-1} \boldsymbol{x} \rangle = \|\boldsymbol{z}\|_2^2 = r^2 \,. \tag{G.9}$$

We therefore recover the one-dimensional case with $\mathsf{P}_{|X|}$ replaced by $\mathsf{P}_R$.

### G.2 Generalized Linear Models

Continuing from Example 4.2, we note that $\mathbb{E}\{y|\boldsymbol{x}\} = \phi'(\langle \boldsymbol{\theta}_*, \boldsymbol{x} \rangle)$. Therefore $\mathbb{E}\{\nabla_{\boldsymbol{\theta}} L(\boldsymbol{\theta}_*; y, \boldsymbol{x})|\boldsymbol{x}\} = \mathbf{0}$. Further

$$\boldsymbol{G}(\boldsymbol{x}) = \mathbb{E}\{(y - \phi'(\langle \boldsymbol{\theta}_*, \boldsymbol{x} \rangle))^2 |\boldsymbol{x}\} \cdot \boldsymbol{x}\boldsymbol{x}^\mathsf{T} = \phi''(\langle \boldsymbol{\theta}_*, \boldsymbol{x} \rangle) \cdot \boldsymbol{x}\boldsymbol{x}^\mathsf{T} = \boldsymbol{H}(\boldsymbol{x}) \,. \tag{G.10}$$

whence we have the following generalization of the score (G.1):

$$Z(\boldsymbol{x}; \pi) = \phi''(\langle \boldsymbol{\theta}_*, \boldsymbol{x} \rangle) \cdot \langle \boldsymbol{x}, \boldsymbol{H}_\pi^{-1} \boldsymbol{Q} \boldsymbol{H}_\pi^{-1} \boldsymbol{x} \rangle \,. \tag{G.11}$$

Note that this is similar to the score derived in the unbiased case, cf. Eq. (4.7), with two important differences that we already encountered for linear regression: $(i)$ The selection process is essentially deterministic: a datapoint is selected if $Z(\boldsymbol{x}_i; \pi) > \lambda$ and not selected if $Z(\boldsymbol{x}_i; \pi) < \lambda$; $(ii)$ The score is computed with respect to the selected data, namely $\boldsymbol{H}$ is replaced by $\boldsymbol{H}_\pi$. The effect of replacing $\boldsymbol{H}$ by $\boldsymbol{H}_\pi$ is illustrated on a toy data distribution in Figure 3.

We observe that at high $\gamma$, the two selection procedures are very similar. In contrast, at low $\gamma$, selection based on $\boldsymbol{H}$ is always biased towards selecting "hard samples" (in directions roughly orthogonal to $\boldsymbol{\theta}_*$), whereas selection based on $\boldsymbol{H}_\pi$ takes into account geometry of selected subset and keeps samples in a more diverse set of directions.

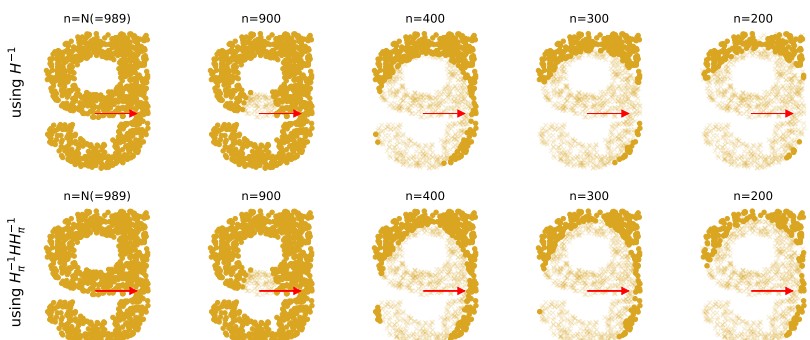

Figure 3: Data selection in a logistic model (here we optimize test error with respect to log-loss). Covariates are bi-dimensional and uniformly distributed over the letter **g**. The red arrow corresponds to the true parameter vector $\boldsymbol{\theta}^*$. Selected points are dark yellow (circles) and non selected ones are light yellow (crosses). Top row: selecting data with largest value of the influence function. Bottom: optimal non-reweighting selection scheme.

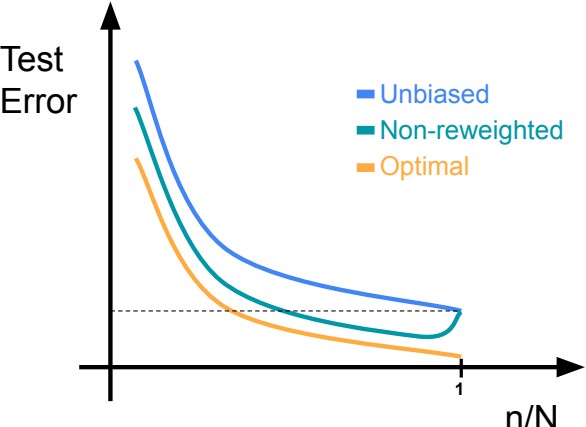

Figure 4: Cartoon of the monotonicity properties of various data-selection schemes.

We also note that, for GLMs, the optimal asymptotic error coefficient of Proposition 4.3 takes the particularly simple form

$$\rho_{\mathrm{nr}}(\boldsymbol{Q}) = \frac{1}{\gamma} \inf_{\pi:\, \mathbb{E}\pi=\gamma} \mathsf{Tr}(\boldsymbol{H}_\pi^{-1}\boldsymbol{Q}) \,. \tag{G.12}$$

In the case of well-specified GLMs, biased data selection cannot improve over full sample estimation. Indeed, $\rho_{\mathrm{nr}}(\boldsymbol{Q}) = \mathsf{Tr}(\mathbb{E}\{\boldsymbol{H}(\boldsymbol{x})\pi_{\mathrm{nr}}(\boldsymbol{x})\}^{-1}\boldsymbol{Q})$, and $\mathbb{E}\{\boldsymbol{H}(\boldsymbol{x})\pi_{\mathrm{nr}}(\boldsymbol{x})\} \preceq \mathbb{E}\{\boldsymbol{H}(\boldsymbol{x})\}$. On the other hand, it is clear that $\rho_{\mathrm{nr}}(\boldsymbol{Q}) \leq \rho_{\mathrm{rand}}(\boldsymbol{Q})$ (because $\rho_{\mathrm{rand}}(\boldsymbol{Q})$ corresponds to a special choice of $\pi$ on the right-hand side of Eq. (G.12)). Further, the inequality is strict (namely, $\rho_{\mathrm{nr}}(\boldsymbol{Q}) < \rho_{\mathrm{rand}}(\boldsymbol{Q})$) except (possibly) on a the degenerate case in which $Z(\boldsymbol{x};\pi)$ is constant on a set of positive measure.

## H   MONOTONICITY PROPERTIES OF BIASED DATA SELECTION

Figure 4 provides a cartoon illustration of the following monotonicity properties:

- For unbiased (reweighted) data selection, we saw in Section 4.1 that the asymptotic test error is always monotone in the size of the subsample $n$ (at least within the low-dimensional setting studied here). In particular, full sample data ERM cannot have worse test error than ERM on selected data.

- Consider then the optimal data selection scheme with reweighting. We claim that this is also monotone. Indeed given target sample sizes $n_1 < n_2$, one can simulate data selection

at sample size $n_1$ by first selecting $n_2$ samples and then setting to 0 the weights of $n_2 - n_1$ samples.

However, for $n = N$, this scheme does not reduce to unweighted ERM, but to optimally weighted ERM. As a consequence, this monotonicity property does not imply that original unweighted ERM on the full sample has better test error than weighted ERM on a data-selected subsample.

- The main result of the next section will be a proof that non-monotonicity is possible in a neighborhood of $\gamma = 1$ for non-reweighting schemes.

# I  BEATING FULL-SAMPLE ESTIMATION USING DATA SELECTION: LOW-DIMENSIONAL ASYMPTOTICS

We next state theorems and corresponding proofs showing that non-monotonicity is possible in a neighborhood of $\gamma = 1$ for non-reweighting schemes under low-dimensional asymptotics.

**Theorem 5.** *Under the setting of Proposition 4.3, further assume $\boldsymbol{H} \succ \boldsymbol{0}$, $\mathbb{E}\{\|\boldsymbol{G}(\boldsymbol{x})\|_{\mathrm{op}}^4\} < \infty$, $\mathbb{E}\{\|\boldsymbol{H}(\boldsymbol{x})\|_{\mathrm{op}}^4\} < \infty$. Then, there exists a constant $C$ such that, for any $\lambda \in \mathbb{R}$ such that $\mathbb{P}(Z_{\boldsymbol{Q}}(\boldsymbol{x}; 1) < \lambda) > 0$, we have*

$$\rho_{\mathrm{nr}}(\boldsymbol{Q}; \gamma) \le \rho_{\mathrm{nr}}(\boldsymbol{Q}; 1) - (1 - \gamma)\mathbb{E}\big\{Z_{\boldsymbol{Q}}(\boldsymbol{x}; 1)\big|Z_{\boldsymbol{Q}}(\boldsymbol{x}; 1) < \lambda\big\} + C(1 - \gamma)^{3/2}, \tag{I.1}$$

$$Z_{\boldsymbol{Q}}(\boldsymbol{x}; 1) := -\mathsf{Tr}\big\{\boldsymbol{G}(\boldsymbol{x})\boldsymbol{H}^{-1}\boldsymbol{Q}\boldsymbol{H}^{-1}\big\} + 2\mathsf{Tr}\big\{\boldsymbol{H}(\boldsymbol{x})\boldsymbol{H}^{-1}\boldsymbol{Q}\boldsymbol{H}^{-1}\boldsymbol{G}\boldsymbol{H}^{-1}\big\}. \tag{I.2}$$

*Further*

(a) *If $\partial_\gamma \rho_{\mathrm{nr}}(\boldsymbol{Q}; 1) = -\mathrm{ess\,inf}\, Z_{\boldsymbol{Q}}(\boldsymbol{x}; 1)$. (Note that this is potentially equal to $+\infty$.)*

(b) *If $\mathbb{P}(Z_{\boldsymbol{Q}}(\boldsymbol{x}; 1) < 0) > 0$, then there exists $\gamma_0 = \gamma_0(d) < 1$ such that*

$$\gamma \in (\gamma_0(d), 1) \quad \Rightarrow \quad \rho_{\mathrm{nr}}(\boldsymbol{Q}; \gamma) < \rho_{\mathrm{nr}}(\boldsymbol{Q}; 1). \tag{I.3}$$

We next construct specific cases in which $\mathbb{P}(Z_{\boldsymbol{Q}}(\boldsymbol{x}; 1) < 0) > 0$ for the case of misspecified linear model, namely $(y_i, \boldsymbol{x}_i) \in \mathbb{R} \times \mathbb{R}^d$ with

$$\mathbb{P}(y_i \in A|\boldsymbol{x}_i) = \mathsf{P}(A|\langle\boldsymbol{\theta}_0, \boldsymbol{x}_i\rangle), \tag{I.4}$$

where $\boldsymbol{\theta}_0 \in \mathbb{R}^d$ is a fixed vector. We will show that both in the case of linear regression and logistic regression, there are choices of the conditional distribution $\mathsf{P}$, for which $\gamma \mapsto \rho(\boldsymbol{Q}; \gamma)$ is strictly increasing near $\gamma = 1$. In this cases, training on a selected subsample provably helps.

**Theorem 6.** *Assume $\boldsymbol{x}_i \sim \mathsf{N}(0, \boldsymbol{I}_d)$, and $y_i$ distributed according to Eq. (4.8). Let $\boldsymbol{Q} = \boldsymbol{H} := \nabla^2 R(\boldsymbol{\theta}_*)$, (as before, $\boldsymbol{\theta}_* := \arg\min_{\boldsymbol{\theta}} R(\boldsymbol{\theta})$). Then in each of the following cases, there exists $\mathsf{P}(A|t)$ such that Eq. (I.3) holds:*

(a) *Least squares, i.e. $L(\boldsymbol{\theta}; y, \boldsymbol{x}) = (y - \langle\boldsymbol{\theta}, \boldsymbol{x}\rangle)^2/2$, $L_{\mathrm{test}} = L$.*

(b) *Logistic regression, whereby $y_i \in \{+1, -1\}$, $L(\boldsymbol{\theta}; y_i, \boldsymbol{x}_i) = -y_i\langle\boldsymbol{\theta}, \boldsymbol{x}_i\rangle + \phi(\langle\boldsymbol{\theta}, \boldsymbol{x}_i\rangle)$, $\phi(t) = \log(e^t + e^{-t})$, $L_{\mathrm{test}} = L$.*

The proofs of Theorem 5 and Theorem 6 are presented next.

## I.1  PROOF OF THEOREM 5

Write $\tilde{\boldsymbol{G}}_\pi = \mathbb{E}\{\boldsymbol{G}(\boldsymbol{x})\pi(\boldsymbol{x})\}$, and similarly for $\tilde{\boldsymbol{H}}_\pi$. Defining $\overline{\pi}(\boldsymbol{x}) := 1 - \pi(\boldsymbol{x})$, and using $\mathbb{E}\{\overline{\pi}(\boldsymbol{x})\} = 1 - \gamma$, we get

$$\tilde{\boldsymbol{G}}_\pi = \boldsymbol{G} - \mathbb{E}\{\boldsymbol{G}(\boldsymbol{x})\overline{\pi}(\boldsymbol{x})\}, \quad \|\mathbb{E}\{\boldsymbol{G}(\boldsymbol{x})\overline{\pi}(\boldsymbol{x})\}\|_{\mathrm{op}} \le \mathbb{E}\{\|\boldsymbol{G}(\boldsymbol{x})\|_{\mathrm{op}}^4\}^{1/4}(1 - \gamma)^{3/4}, \tag{I.5}$$

and similarly for $\tilde{\boldsymbol{H}}_\pi$.

Using the form of $\rho_{\mathrm{nr}}$ in Proposition 4.3 and Taylor expansion (recall $\boldsymbol{H} \succ \boldsymbol{0}$ by assumption), we get, for any $\pi$ satisfying $\mathbb{E}\pi(\boldsymbol{x}) = \gamma$,

$$\rho_{\mathrm{nr}}(\boldsymbol{Q}; \gamma) \le \rho_{\mathrm{nr}}(\boldsymbol{Q}; 1) + \mathbb{E}\{Z_{\boldsymbol{Q}}(\boldsymbol{x}; 1)\overline{\pi}(\boldsymbol{x})\} + O\big(\|\mathbb{E}\{\boldsymbol{G}(\boldsymbol{x})\overline{\pi}(\boldsymbol{x})\}\|_{\mathrm{op}}^2 + \|\mathbb{E}\{\boldsymbol{H}(\boldsymbol{x})\overline{\pi}(\boldsymbol{x})\}\|_{\mathrm{op}}^2\big)$$

$$= \rho_{\mathrm{nr}}(\boldsymbol{Q}; 1) + \mathbb{E}\{Z_{\boldsymbol{Q}}(\boldsymbol{x}; 1)\overline{\pi}(\boldsymbol{x})\} + O\big((1 - \gamma)^{3/2}\big),$$

where $Z_{\boldsymbol{Q}}(\boldsymbol{x}; 1)$ is defined in the statement of the theorem.

For $\lambda$ as in the statement, and all $(1 - \gamma)$ small enough, we can take $\overline{\pi}(\boldsymbol{x}) = (1 - \gamma)/\mathbb{P}(Z_{\boldsymbol{Q}}(\boldsymbol{x}; 1) < \lambda)$ if $Z_{\boldsymbol{Q}}(\boldsymbol{x}; 1) < \lambda$, and $\overline{\pi}(\boldsymbol{x}) = 0$ otherwise. This immediately implies Eq. (I.1) whence point $(b)$ follows.

In order to prove point $(a)$, note that, by definition, for any $\lambda > \operatorname{ess\,inf} Z_{\boldsymbol{Q}}(\boldsymbol{x})$, we have $\mathbb{P}(Z_{\boldsymbol{Q}}(\boldsymbol{x}) < \lambda) > 0$ and therefore Eq. (I.1) implies, for all $1 - \gamma$ small enough

$$\rho_{\mathrm{nr}}(\boldsymbol{Q}; \gamma) \le \rho_{\mathrm{nr}}(\boldsymbol{Q}; 1) + \lambda(1 - \gamma) + C(1 - \gamma)^{3/2} \,. \tag{I.6}$$

This implies $\partial_{\gamma}\rho_{\mathrm{nr}}(\boldsymbol{Q}; 1) \ge -\lambda$ and therefore $\partial_{\gamma}\rho_{\mathrm{nr}}(\boldsymbol{Q}; 1) \ge -\operatorname{ess\,inf} Z_{\boldsymbol{Q}}(\boldsymbol{x})$.

Alternatively, assume $\operatorname{ess\,inf} Z_{\boldsymbol{Q}}(\boldsymbol{x}) = \lambda_* > -\infty$. Let $\pi_{\mathrm{nr},\gamma}$ achieve the infimum in Proposition 4.3 (Proposition 4.3 ensures that such $\pi_{\mathrm{nr},\gamma}$ exists). By the above argument (letting $\overline{\pi}_{\mathrm{nr},\gamma}(\boldsymbol{x}) = 1 - \pi_{\mathrm{nr},\gamma}(\boldsymbol{x})$)

$$\rho_{\mathrm{nr}}(\boldsymbol{Q}; \gamma) = \rho_{\mathrm{nr}}(\boldsymbol{Q}; 1) + \mathbb{E}\{Z_{\boldsymbol{Q}}(\boldsymbol{x}; 1)\overline{\pi}_{\mathrm{nr},\gamma}(\boldsymbol{x})\} + O\big(\|\mathbb{E}\{\boldsymbol{G}(\boldsymbol{x})\overline{\pi}(\boldsymbol{x})\}\|_{\mathrm{op}}^2 + \|\mathbb{E}\{\boldsymbol{H}(\boldsymbol{x})\overline{\pi}(\boldsymbol{x})\}\|_{\mathrm{op}}^2\big)$$
$$\ge \rho_{\mathrm{nr}}(\boldsymbol{Q}; 1) + \lambda_*(1 - \gamma) + O\big((1 - \gamma)^{3/2}\big) \,.$$

This yields $\partial_{\gamma}\rho_{\mathrm{nr}}(\boldsymbol{Q}; 1) \le -\lambda_*$.

## I.2 PROOF OF THEOREM 6(A)

Both claims of the theorem follow if we can provide an example such that $Z_{\boldsymbol{Q}}(\boldsymbol{x}; 1) < 0$ with strictly positive probability, in the case $\boldsymbol{Q} = \boldsymbol{H}$. Specializing to that case, we have

$$Z_{\boldsymbol{H}}(\boldsymbol{x}; 1) := -\mathsf{Tr}\big\{\boldsymbol{G}(\boldsymbol{x})\boldsymbol{H}^{-1}\big\} + 2\mathsf{Tr}\big\{\boldsymbol{H}(\boldsymbol{x})\boldsymbol{H}^{-1}\boldsymbol{G}\boldsymbol{H}^{-1}\big\} \,. \tag{I.7}$$

In the following we will simplify notations and write $Z(\boldsymbol{x}) = Z_{\boldsymbol{H}}(\boldsymbol{x}; 1)$.

We next consider the linear regression setting of point $(a)$. Without loss of generality, we will assume $\|\boldsymbol{\theta}_0\| = 1$. By rotation invariance, the population risk minimizer has the form $\boldsymbol{\theta}_* = \alpha_* \boldsymbol{\theta}_0$. The coefficient $\alpha_*$ is fixed by

$$\boldsymbol{0} = \nabla R_S(\alpha_* \boldsymbol{\theta}_0) = \mathbb{E}\big\{(y - \alpha_* \langle \boldsymbol{\theta}_0, \boldsymbol{x} \rangle)\boldsymbol{x}\big\} \,.$$

The only non-zero component of this equation is the one along $\boldsymbol{\theta}_0$. Projecting along this direction, we get (for $G \sim \mathsf{N}(0, 1)$, $Y \sim \mathsf{P}(\,\cdot\,|G)$):

$$\mathbb{E}\big\{(Y - \alpha_* G)G\big\} = 0 \,.$$

We next compute

$$\boldsymbol{G}(\boldsymbol{x}) = \mathbb{E}\big\{(y - \langle \boldsymbol{\theta}_*, \boldsymbol{x} \rangle)^2 \big| \boldsymbol{x}\big\}\, \boldsymbol{x}\boldsymbol{x}^\mathsf{T} \,, \quad \boldsymbol{H}(\boldsymbol{x}) = \boldsymbol{x}\boldsymbol{x}^\mathsf{T} \,. \tag{I.8}$$

Note that we can rewrite the first equation as

$$\boldsymbol{G}(\boldsymbol{x}) = f(\langle \boldsymbol{\theta}_0, \boldsymbol{x} \rangle)\, \boldsymbol{x}\boldsymbol{x}^\mathsf{T} \,, \tag{I.9}$$

$$f(t) := \int (y - \alpha_* t)^2 \,\mathsf{P}(\mathrm{d}y|t) \,. \tag{I.10}$$

Taking expectation with respect to $\boldsymbol{x}$

$$\boldsymbol{G} = a\boldsymbol{I}_d + b\boldsymbol{\theta}_0\boldsymbol{\theta}_0^\mathsf{T} \,, \quad \boldsymbol{H} = \boldsymbol{I}_d \,, \tag{I.11}$$

$$a := \mathbb{E}\big\{(Y - \alpha_* G)^2\big\} \,, \quad b := \mathbb{E}\big\{(Y - \alpha_* G)^2(G^2 - 1)\big\} \,. \tag{I.12}$$

Substituting in Eq. (I.7), we get

$$Z(\boldsymbol{x}) = -f(\langle \boldsymbol{\theta}_0, \boldsymbol{x} \rangle)\|\boldsymbol{x}\|^2 + 2a\|\boldsymbol{x}\|^2 + 2b\langle \boldsymbol{\theta}_0, \boldsymbol{x} \rangle^2$$
$$= \langle \boldsymbol{\theta}_0, \boldsymbol{x} \rangle^2 \big[ -f(\langle \boldsymbol{\theta}_0, \boldsymbol{x} \rangle) + 2\,\mathbb{E}\{f(G)G^2\}\big] \tag{I.13}$$
$$+ \big\|\boldsymbol{P}_0^{\perp}\boldsymbol{x}\big\|_2^2 \big[ -f(\langle \boldsymbol{\theta}_0, \boldsymbol{x} \rangle) + 2\,\mathbb{E}\{f(G)\}\big] \,.$$

It is easy to construct examples in which $\operatorname{ess\,inf} Z(\boldsymbol{x}) = -\infty$. For instance, take $\mathsf{P}(\,\cdot\,|t) = \delta_{h(t)}$, $h(t) = t + c(t^3 - 3t)$ for some $c > 0$ (no noise). Then we get $\alpha_* = 1$ and $f(t) = c^2(t^3 - 2t)^2$. Let $t_0$ be such that $f(t) > 2\mathbb{E}\{G^2 f(G)\}$ for all $t \ge t_0$. Then the claim follows since

$$\mathbb{P}\Big(\boldsymbol{x} : \langle \boldsymbol{\theta}_0, \boldsymbol{x} \rangle \in [t_0, t_0 + 1], \big\|\boldsymbol{P}_0^{\perp}\boldsymbol{x}\big\|_2 > M\Big) > 0 \,, \tag{I.14}$$

for any $M$, and Eq. (I.13) yields $Z(\boldsymbol{x}) < 0$ on the above event for all $M$ large enough. In fact, we also have $Z(\boldsymbol{x}) < -c$ for any $c$, by taking $M$ sufficiently large.

### I.3 PROOF OF THEOREM 6(B)

We next consider a misspecified generalized linear model with $y_i \in \{+1, -1\}$ and

$$\mathsf{P}(+1|z) = 1 - \mathsf{P}(-1|z) = f(z) \,. \tag{I.15}$$

We set $u(t) := 2f(t) - 1$. It is simple to compute

$$\nabla R(\boldsymbol{\theta}) = -\mathbb{E}\big\{(u(\langle \boldsymbol{\theta}_0, \boldsymbol{x}\rangle) - \phi'(\langle \boldsymbol{\theta}, \boldsymbol{x}\rangle))\boldsymbol{x}\big\} \,.$$

In particular $\nabla R(\boldsymbol{\theta}) = -\mathbb{E}\big\{(u(G) - \phi'(G))G\big\}\boldsymbol{\theta}_0$, where expectation is with respect to $G \sim \mathsf{N}(0, 1)$. We will impose the condition

$$\mathbb{E}\big\{Gu(G)\big\} = \mathbb{E}\big\{G\phi'(G))\big\} \,.$$

so that the empirical risk minimizer is $\boldsymbol{\theta}_* = \boldsymbol{\theta}_0$.

Next we compute

$$\boldsymbol{G}(\boldsymbol{x}) = g(\langle \boldsymbol{\theta}_*, \boldsymbol{x}\rangle) \, \boldsymbol{x}\boldsymbol{x}^\mathsf{T} \,, \qquad \boldsymbol{H}(\boldsymbol{x}) = h(\langle \boldsymbol{\theta}_*, \boldsymbol{x}\rangle) \, \boldsymbol{x}\boldsymbol{x}^\mathsf{T} \,, \tag{I.16}$$

where

$$g(t) := (u(t) - \phi'(t))^2 + 1 - u(t)^2 \,, \quad h(t) := 1 - \phi'(t)^2 \,. \tag{I.17}$$

Performing the Gaussian integral, we get

$$\boldsymbol{G} = a_G \boldsymbol{I}_d + b_G \boldsymbol{\theta}_* \boldsymbol{\theta}_*^\mathsf{T} \,, \qquad \boldsymbol{H} = a_H \boldsymbol{I}_d + b_H \boldsymbol{\theta}_* \boldsymbol{\theta}_*^\mathsf{T} \,, \tag{I.18}$$

where $a_G := \mathbb{E}g(G)$, $b_G := \mathbb{E}g''(G)$, and similarly for $\boldsymbol{H}$. We thus get

$$\boldsymbol{H}^{-1} = c_1 \boldsymbol{I}_d + c_1' \boldsymbol{\theta}_* \boldsymbol{\theta}_*^\mathsf{T} \,, \qquad \boldsymbol{H}^{-1}\boldsymbol{G}\boldsymbol{H}^{-1} = c_2 \boldsymbol{I}_d + c_2' \boldsymbol{\theta}_* \boldsymbol{\theta}_*^\mathsf{T} \,, \tag{I.19}$$

for some constants $c_i, c_i'$ that are dimension-independent functions of $a_H, b_H, a_G, b_G$. Substituting in the formula for $Z(\boldsymbol{x}) = Z_{\boldsymbol{H}}(\boldsymbol{x}; 1)$, cf. Eq. (I.7), we get

$$Z(\boldsymbol{x}) := -g(\langle \boldsymbol{\theta}_*, \boldsymbol{x}\rangle) \, \langle \boldsymbol{x}, (c_1 \boldsymbol{I}_d + c_1' \boldsymbol{\theta}_* \boldsymbol{\theta}_*^\mathsf{T})\boldsymbol{x}\rangle + 2h(\langle \boldsymbol{\theta}_*, \boldsymbol{x}\rangle) \, \langle \boldsymbol{x}, (c_2 \boldsymbol{I}_d + c_2' \boldsymbol{\theta}_* \boldsymbol{\theta}_*^\mathsf{T})\boldsymbol{x}\rangle \,. \tag{I.20}$$

For large $d$, we have

$$\langle \boldsymbol{x}, (c\boldsymbol{I}_d + c'\boldsymbol{\theta}_* \boldsymbol{\theta}_*^\mathsf{T})\boldsymbol{x}\rangle = c\,d + o_P(1) \,, \tag{I.21}$$

and therefore

$$\frac{1}{d}Z(\boldsymbol{x}) = -c_1 g(\langle \boldsymbol{\theta}_*, \boldsymbol{x}\rangle) + c_2 h(\langle \boldsymbol{\theta}_*, \boldsymbol{x}\rangle) + o_P(d) \,. \tag{I.22}$$

Note that $G = \langle \boldsymbol{\theta}_*, \boldsymbol{x}\rangle \sim \mathsf{N}(0, 1)$. In particular, its distribution is $d$-independent. It is therefore sufficient to prove that $-c_1 g(G) + 2c_2 h(G) < 0$ with strictly positive probability, since this implies $Z(\boldsymbol{x})$ with strictly positive probability for all $d$ large enoug. It is simple to compute $c_1 = 1/a_H$, $c_2 = a_G/a_H^2$. Therefore it is sufficient to prove the following

> **Claim:** We can choose $f$ so that Eq. (I.16) is satisfied and $-a_H g(G) + 2a_G h(G) < 0$ with strictly positive probability.

To prove this claim, it is convenient to define the random variables $W = \phi'(G)$, $M = u(G)$ and note that the distribution of $W$ is symmetric around 0, and has support $(-1, 1)$.

We have $g(G) = (M - W)^2 + 1 - M^2$, $h(G) = 1 - W^2$. Therefore, the conditions of the claim are equivalent to (the second condition is understood to hold with strictly positive probability)

$$\mathbb{E}\big\{(M - W)\,\psi(W)\big\} = 0 \,, \tag{I.23}$$

$$a_H \cdot \big[(M - W)^2 + 1 - M^2\big] > 2a_G \cdot \big[1 - W^2\big] \,, \tag{I.24}$$

where $\psi = (\phi')^{-1}$ (functional inverse), and

$$a_G = \mathbb{E}\big\{(M - W)^2 + 1 - M^2\big\} \,, \quad a_H = \mathbb{E}\big\{1 - W^2\big\} \,. \tag{I.25}$$

We will further restrict ourselves to construct these random variables so that $2a_G = a_H$, i.e.

$$\mathbb{E}\{1 - W^2 - 4W(M - W)\} = 0. \tag{I.26}$$

Next define $\varphi : \mathbb{R} \to \mathbb{R}$ by $\varphi(x) = u(x) - \phi'(x)$, whence $M - W = \varphi(G)$, $W = \phi'(G) = \tanh(G)$. Therefore, it is sufficient to construct $\varphi : \mathbb{R} \to \mathbb{R}$ such that, for some $x_0 \in \mathbb{R}$, $\varepsilon > 0$,

$$\mathbb{E}\{\varphi(G) \tanh(G)\} = b_0, \tag{I.27}$$

$$\mathbb{E}\{\varphi(G)G\} = 0, \tag{I.28}$$

$$\varphi(x)^2 + 1 - (\tanh(x) + \varphi(x))^2 > 1 - \tanh(x)^2 \quad \text{for all } x \in (x_0 - \varepsilon, x_0 + \varepsilon), \tag{I.29}$$

$$-1 - \tanh(x) < \varphi(x) < 1 - \tanh(x) \qquad \text{for all } x \in \mathbb{R}. \tag{I.30}$$

where $b_0 = \mathbb{E}\{1 - \tanh(G)^2\}/4$ is a constant $b_0 \in (0, 1/4)$. Note that Eq. (I.29) is sufficient because the law of $G$ is supported on the whole real line. Simplifying Eq. (I.29), and assuming $\varphi$ is continuous, we are led to

$$\mathbb{E}\{\varphi(G) \tanh(G)\} = b_0, \tag{I.31}$$

$$\mathbb{E}\{\varphi(G)G\} = 0, \tag{I.32}$$

$$x\varphi(x) < 0 \qquad \text{for some } x \in \mathbb{R}, \tag{I.33}$$

$$-1 - \tanh(x) < \varphi(x) < 1 - \tanh(x) \quad \text{for all } x \in \mathbb{R}. \tag{I.34}$$

We complete the proof by the following result.

**Lemma I.1.** *There exists $\varphi : \mathbb{R} \to \mathbb{R}$ continuous satisfying conditions* (I.31) *to* (I.34) *above.*

*Proof.* We define $\mathsf{F} : L^2(\mathsf{N}(0, 1)) \to \mathbb{R}^2$ by $\mathsf{F}(\varphi) = (\mathbb{E}\{\varphi(G) \tanh(G)\}, \mathbb{E}\{\varphi(G)G\})$, and

$$\mathcal{C} := \left\{ \varphi \in C(\mathbb{R}) : \forall x \in \mathbb{R} \ \ \max(-1 - \tanh(x), -1) < \varphi(x) < \min(1 - \tanh(x), 1) \right\}. \tag{I.35}$$

In particular, any $\varphi \in \mathcal{C}$ satisfies condition (I.34). We claim that there exists an open set $B \subseteq \mathbb{R}^2$, such that $(b_0, 0) \in B$ and $\mathsf{F}(\mathcal{C}) \supseteq B$ (i.e., for every $\boldsymbol{x} \in B$, there exists $\varphi \in \mathcal{C}$ such that $\mathsf{F}(\varphi) = \boldsymbol{x}$).

In order to prove this claim, note that $\mathsf{F}$ is a continuous linear map and $\mathcal{C}$ is convex, whence $\mathsf{F}(\mathcal{C})$ is convex. Hence, it is sufficient to exhibits points $\varphi_1, \varphi_2, \varphi_3 \in \overline{\mathcal{C}}$ (the closure is in $L^2(\mathsf{N}(0, 1))$ of $\mathcal{C}$), such that $(b_0, 0)$ is in the interior of the convex hull of $\{\mathsf{F}(\varphi_i) : i \leq 3\}$. We use the following functions:

1. $\varphi_1(x) = 0$: $\mathsf{F}(\varphi_1) = (0, 0)$.

2. $\varphi_2(x) = \text{sign}(x)(1 - \tanh(|x|))$:

$$\mathsf{F}_1(\varphi_2) = \mathbb{E}\{\tanh|G|(1 - \tanh|G|)\} =: b_1 > b_0, \tag{I.36}$$

$$\mathsf{F}_2(\varphi_2) = \mathbb{E}\{|G|(1 - \tanh|G|)\} > 0, \tag{I.37}$$

   (Numerically, $b_1 \approx 0.16168$, $b_0 \approx 0.15143$.)

3. $\varphi_3(x) = \varphi_2(x) - M^{-1/2} q(x - M)$, where $q(x)$ is a continuous non-negative function supported on $[-1, 1]$, with $\int q(x)\, \mathrm{d}x > 0$. We have $\mathsf{F}_1(\varphi_3) = \mathsf{F}_1(\varphi_2) - \Theta(M^{-1/2})$ and $\mathsf{F}_2(\varphi_3) = \mathsf{F}_2(\varphi_2) - \Theta(M^{1/2})$. Hence for a sufficiently large $M$, $\mathsf{F}_1(\varphi_3) > b_0$, $\mathsf{F}_2(\varphi_3) < 0$.

This proves the claim, and in particular we can satisfy conditions (I.31), (I.32), (I.34), since $(b_0, 0) \in B$. In order to show that we can satisfy also condition (I.33), let $q$ be defined as above and further such that $q(x) \leq 1/2$ for all $x$. Consider the sequence of functions indexed by $k \in \mathbb{N}$:

$$\varphi_k(x) = \overline{\varphi}_k(x) - q(x - k). \tag{I.38}$$

First note that, if $\overline{\varphi}_k \in \mathcal{C}$, then $\varphi_k$ satisfies conditions (I.33), (I.34). Therefore, we are left to prove that, for some $k$, there exists $\overline{\varphi}_k \in \mathcal{C}$ that satisfies

$$\mathsf{F}_1(\overline{\varphi}_k) = b_0 + \mathbb{E}\{q(G - k) \tanh(G)\} =: b_0 + \delta_{1,k}, \tag{I.39}$$

$$\mathsf{F}_2(\overline{\varphi}_k) = \mathbb{E}\{q(G - k) G\} =: \delta_{2,k}. \tag{I.40}$$

By dominated convergence, we have $\boldsymbol{\delta}_k = (\delta_{1,k}, \delta_{2,k}) \to 0$ as $k \to \infty$, and therefore $(b_0, 0) + \boldsymbol{\delta}_k \in B$ for all $k$ sufficiently large, whence the existence of $\overline{\varphi}_k$ for such $k$ follows from the first part of the proof. $\qquad \square$

## J    PROOF OF THEOREM 3

Let $\pi_{\mathrm{unb}}$ be the optimal unbiased sampling probability (see Proposition 4.1), with corresponding weight $w_{\mathrm{unb}}(\boldsymbol{x}) = 1/\pi_{\mathrm{unb}}(\boldsymbol{x})$. Since $\boldsymbol{G}(\boldsymbol{x}) = \boldsymbol{H}(\boldsymbol{x})$ is almost surely bounded (Assumption A2), it follows that $Z(\boldsymbol{x}) = \mathsf{Tr}(\boldsymbol{G}(\boldsymbol{x})\boldsymbol{H}^{-1})$ is bounded as well. Therefore, there exists $\gamma_0 > 0$ such that, for all $\gamma \in (0, \gamma_0)$, $\pi_{\mathrm{unb}}(\boldsymbol{x}) = c(\gamma)Z(\boldsymbol{x})^{1/2}$.

For a bounded function $\varphi$, consider the alternative weight $w_\varepsilon(\boldsymbol{x}) = (1 + \varepsilon\varphi(\boldsymbol{x}))/\pi_{\mathrm{unb}}(\boldsymbol{x})$, which is well defined for all $\varepsilon$ small enough. Recall that we denote by $\rho(\pi, w; \boldsymbol{Q})$ the asymptotic estimation error coefficient for the simple scheme $(\pi, w)$. To linear order in $\varepsilon$, we have

$$\rho(\pi_{\mathrm{unb}}, w_\varepsilon; \boldsymbol{H}) - \rho(\pi_{\mathrm{unb}}, w_0; \boldsymbol{H}) = -2\varepsilon\, \mathbb{E}\{\mathcal{W}(\boldsymbol{x})\, \varphi(\boldsymbol{x})\} + o(\varepsilon)\,. \tag{J.1}$$

where

$$\mathcal{W}(\boldsymbol{x}) := \mathsf{Tr}\Big(\mathbb{E}_{\boldsymbol{x}'}\Big(\frac{\boldsymbol{G}(\boldsymbol{x}')}{\pi_{\mathrm{unb}}(\boldsymbol{x}')}\Big)\boldsymbol{H}^{-1}\boldsymbol{H}(\boldsymbol{x})\boldsymbol{H}^{-1}\Big) - \mathsf{Tr}\Big(\frac{\boldsymbol{G}(\boldsymbol{x})}{\pi_{\mathrm{unb}}(\boldsymbol{x})}\boldsymbol{H}^{-1}\Big)\,. \tag{J.2}$$

Assume by contradiction $\rho(\pi_*, w_\varepsilon; \boldsymbol{H}) \geq \rho(\pi_*, w_-; \boldsymbol{H})$ for every $\varphi$, $\varepsilon$ such that $w_\varepsilon \geq 0$. By Eq. (J.1), this implies $\mathcal{W}(\boldsymbol{x}) = 0$ for $\mathbb{P}$-almost every $\boldsymbol{x}$. Using assumption A1, and defining $\boldsymbol{M}(\boldsymbol{x}) := \boldsymbol{H}^{-1/2}\boldsymbol{G}(\boldsymbol{x})\boldsymbol{H}^{-1/2}$, we get

$$\mathcal{W}(\boldsymbol{x}) = \mathsf{Tr}\Big(\mathbb{E}_{\boldsymbol{x}'}\Big(\frac{\boldsymbol{M}(\boldsymbol{x}')}{\pi_{\mathrm{unb}}(\boldsymbol{x}')}\Big)\boldsymbol{M}(\boldsymbol{x})\Big) - \frac{1}{\pi_{\mathrm{unb}}(\boldsymbol{x})}\mathsf{Tr}\big(\boldsymbol{M}(\boldsymbol{x})\big)\,. \tag{J.3}$$

we have $\pi_{\mathrm{unb}}(\boldsymbol{x}) = c(\gamma)\mathsf{Tr}\big(\boldsymbol{M}(\boldsymbol{x})\big)^{1/2}$ and therefore $\mathcal{W}(\boldsymbol{x}) = \mathcal{W}_0(\boldsymbol{x}) \cdot \mathsf{Tr}\big(\boldsymbol{M}(\boldsymbol{x})\big)^{1/2}/c(\gamma)$, where

$$\mathcal{W}_0(\boldsymbol{x}) = \mathsf{Tr}\Big(\mathbb{E}_{\boldsymbol{x}'}\Big(\frac{\boldsymbol{M}(\boldsymbol{x}')}{\mathsf{Tr}\big(\boldsymbol{M}(\boldsymbol{x}')\big)^{1/2}}\Big) \cdot \frac{\boldsymbol{M}(\boldsymbol{x})}{\mathsf{Tr}\big(\boldsymbol{M}(\boldsymbol{x})\big)^{1/2}}\Big) - 1 \tag{J.4}$$

$$= \mathsf{Tr}\big(\overline{\boldsymbol{W}} \cdot \boldsymbol{W}(\boldsymbol{x})\big) - 1\,.$$

Here we defined $\boldsymbol{W}(\boldsymbol{x}) := \boldsymbol{M}(\boldsymbol{x})/\mathsf{Tr}\big(\boldsymbol{M}(\boldsymbol{x})\big)^{1/2}, \overline{\boldsymbol{W}} := \mathbb{E}\boldsymbol{W}(\boldsymbol{x})$.

Since $\mathcal{W}(\boldsymbol{x}) = 0$ almost surely, and $\boldsymbol{G}(\boldsymbol{x}) \neq \boldsymbol{0}$ almost surely (by assumption A2), we must have $\mathsf{Tr}\big(\overline{\boldsymbol{W}} \cdot \boldsymbol{W}(\boldsymbol{x})\big) = 1$ almost surely, which contradicts the assumption of $\boldsymbol{G}(\boldsymbol{x})/\mathsf{Tr}(\boldsymbol{G}(\boldsymbol{x})\boldsymbol{H}^{-1})^{-1/2}$ not lying on an affine subspace (Assumption A3).

**Example J.1** (Generalized linear models)**.** Consider again the GLM model of Example 4.2 (log-likelihood loss). Recall that in this case

$$\boldsymbol{G}(\boldsymbol{x}) = \boldsymbol{H}(\boldsymbol{x}) = \phi''(\langle\boldsymbol{\theta}_*, \boldsymbol{x}\rangle)\,\boldsymbol{x}\boldsymbol{x}^{\mathsf{T}}\,. \tag{J.5}$$

Hence condition A1 of Theorem 3 is satisfied, and A4 is also always satisfied. If $\mathbb{P}(\mathrm{d}\boldsymbol{x})$ is supported on $\|\boldsymbol{x}\|_2 \leq M$, and $\mathbb{P}(\boldsymbol{x} = 0) = 0$ then condition A2 is also satisfied. (Note that $\phi''(t) > 0$ for any $t > 0$ unless $\nu_0 = \delta_c$ is a point mass, which is a degenerate case.) Finally, for condition A3 note that

$$\frac{\boldsymbol{G}(\boldsymbol{x})}{\mathsf{Tr}(\boldsymbol{G}(\boldsymbol{x})\boldsymbol{H}^{-1})^{1/2}} = \frac{\boldsymbol{x}\boldsymbol{x}^{\mathsf{T}}}{\langle\boldsymbol{x}, \boldsymbol{H}\boldsymbol{x}\rangle}\,. \tag{J.6}$$

Therefore, a sufficient condition for A3 is that the support of $\mathbb{P}(\mathrm{d}\boldsymbol{x})$ contains an arbitrarily small ball $\mathsf{B}^d(\boldsymbol{x}_0; \varepsilon) \subseteq \mathbb{R}^d$,

## K    LOW-DIMENSIONAL ASYMPTOTICS: IMPERFECT SURROGATES

In this section, we discuss the imperfect surrogates under low-dimensional asymptotics. The main conclusion of this section is that plug-in schemes can be suboptimal and weaker surrogate models can indeed lead to better data selection.

Throughout, we will use the notation $\mathbb{E}_{\mathrm{su}}\{F(y, \boldsymbol{x})|\boldsymbol{x}\} = \int F(y, \boldsymbol{x})\, \mathsf{P}_{\mathrm{su}}(\mathrm{d}y|\boldsymbol{x})$, and similarly for $\mathbb{P}_{\mathrm{su}}(\,\cdot\,|\boldsymbol{x})$.

## K.1 PLUGIN SCHEMES

The simplest approach to utilize an imperfect surrogate proceeds as follows: $(i)$ Choose a data selection scheme under the assumption of ideal surrogate; $(ii)$ Replace the conditional expectations $\mathbb{E}\{F(y, \boldsymbol{x})|\boldsymbol{x}\}$ in that scheme by expectations with respect to the surrogate model $\mathbb{E}_{\mathrm{su}}\{F(y, \boldsymbol{x})|\boldsymbol{x}\}$; $(iii)$ Replace expectations over $\boldsymbol{x}$ by expectation over the data sample.

In particular, revisiting the schemes of Sections 4.1 and 4.2, we obtain the following:

*Plugin unbiased data selection.* We form

$$\boldsymbol{G}_{\mathrm{su}}(\boldsymbol{x}) := \mathbb{E}_{\mathrm{su}}\big\{\nabla_{\boldsymbol{\theta}} L(\hat{\boldsymbol{\theta}}^{\mathrm{su}}; y, \boldsymbol{x})\nabla_{\boldsymbol{\theta}} L(\hat{\boldsymbol{\theta}}^{\mathrm{su}}; y, \boldsymbol{x})^{\mathsf{T}}|\boldsymbol{x}\big\}, \tag{K.1}$$

$$\boldsymbol{H}_{\mathrm{su}}(\boldsymbol{x}) := \mathbb{E}_{\mathrm{su}}\big\{\nabla_{\boldsymbol{\theta}}^2 L(\hat{\boldsymbol{\theta}}^{\mathrm{su}}; y, \boldsymbol{x})|\boldsymbol{x}\big\}, \tag{K.2}$$

and subsample according to

$$\pi(\boldsymbol{x}) = \min\left(1; c(\gamma)\, Z_{\mathrm{su}}(\boldsymbol{x})^{1/2}\right), \tag{K.3}$$

$$Z_{\mathrm{su}}(\boldsymbol{x}) := \mathsf{Tr}\big(\boldsymbol{G}_{\mathrm{su}}(\boldsymbol{x})\boldsymbol{H}_{1,\mathrm{su}}^{-1}\boldsymbol{Q}\boldsymbol{H}_{1,\mathrm{su}}^{-1}\big), \tag{K.4}$$

$$\boldsymbol{H}_{1,\mathrm{su}} := \mathbb{E}\{\boldsymbol{H}_{\mathrm{su}}(\boldsymbol{x})\}. \tag{K.5}$$

We then reweight each selected sample proportionally to $1/\pi(\boldsymbol{x})$. Note that:

- The 'true' parameters vector $\boldsymbol{\theta}_*$ appearing in $\nabla_{\boldsymbol{\theta}} L(\,\cdot\,; y, \boldsymbol{x})$, $\nabla_{\boldsymbol{\theta}}^2 L(\,\cdot\,; y, \boldsymbol{x})$ was replaced by an estimate obtained from the surrogate model. In certain applications $\hat{\boldsymbol{\theta}}^{\mathrm{su}}$ can be 'read off' the surrogate model itself. In general, we can define it via $\hat{\boldsymbol{\theta}}^{\mathrm{su}} :=$ $\arg\min \sum_{i=1}^N L(\boldsymbol{\theta}; y_i^{\mathrm{su}}, \boldsymbol{x}_i)$, where $(y_i^{\mathrm{su}})_{i\le N}$ are drawn independently according to $y_i \sim \mathsf{P}_{\mathrm{su}}(\,\cdot\,|\boldsymbol{x}_i)$.

- The matrix $\boldsymbol{H}_{1,\mathrm{su}}$ can be replaced its empirical version $\hat{\boldsymbol{H}}_{1,\mathrm{su}} := N^{-1} \sum_{i=1}^N \boldsymbol{H}_{\mathrm{su}}(\boldsymbol{x}_i)$.

*Plugin non-reweighting data selection.* In this case we select samples such that $Z_{\mathrm{su}}(\boldsymbol{x}; \pi) > \lambda$, cf. Eq. (4.10), where

$$Z(\boldsymbol{x}; \pi) := -\mathsf{Tr}\big\{\boldsymbol{G}_{\mathrm{su}}(\boldsymbol{x})\boldsymbol{H}_{\mathrm{su},\pi}^{-1}\boldsymbol{Q}\boldsymbol{H}_{\mathrm{su},\pi}^{-1}\big\} + 2\mathsf{Tr}\big\{\boldsymbol{H}_{\mathrm{su}}(\boldsymbol{x})\boldsymbol{H}_{\mathrm{su},\pi}^{-1}\boldsymbol{Q}\boldsymbol{H}_{\mathrm{su},\pi}^{-1}\boldsymbol{G}_{\mathrm{su},\pi}\boldsymbol{H}_{\mathrm{su},\pi}^{-1}\big\}, \tag{K.6}$$

and $\boldsymbol{H}_{\mathrm{su},\pi} := \mathbb{E}\{\boldsymbol{H}_{\mathrm{su}}(\boldsymbol{x})\pi(\boldsymbol{x})\}/\mathbb{E}\{\pi(\boldsymbol{x})\}$ and similarly for $\boldsymbol{G}_{\mathrm{su},\pi}$. Again, expectations over $\boldsymbol{x}$ are replaced by averages over the $N$ samples.

Plugin approaches are natural and easy to define, and in fact we will use them in our simulations. However, we will show that they can be suboptimal. Before doing that, we need to make more explicit the notion of optimality that is relevant here.

## K.2 MINIMAX FORMULATION

We want formalize the idea that we do not know the conditional distribution of $y$ given $\boldsymbol{x}$, but we have some information about it coming from the surrogate model $\mathsf{P}_{\mathrm{su}}$. With this in mind, we introduce a set of probability kernels

$$\mathscr{K}_d \subseteq \mathscr{K}_d^0 := \big\{\mathsf{P} : \mathcal{B}_{\mathbb{R}} \times \mathbb{R}^d \to [0, 1]\ \text{probability kernel}\big\}. \tag{K.7}$$

Informally, $\mathscr{K}_d$ is a neighborhood of the surrogate model $\mathsf{P}_{\mathrm{su}}$, and captures our uncertainty about the actual conditional distribution: we know that $\mathbb{P}_{y|\boldsymbol{x}} \in \mathscr{K}_d$. For instance, we could consider, for some $r \in [0, 1)$,

$$\mathscr{K}_d(\mathsf{P}_{\mathrm{su}}; r) := \big\{\mathsf{P} :\ \mathbb{E}_{\boldsymbol{x}}\|\mathsf{P}(\,\cdot\,|\boldsymbol{x}) - \mathsf{P}_{\mathrm{su}}(\,\cdot\,|\boldsymbol{x})\|_{\mathrm{TV}} \le r\big\}. \tag{K.8}$$

We will assume $\mathscr{K}_d$ (and its variant $\mathscr{K}_{N,d}$ introduced below) to be convex. Namely, for all $\lambda \in [0, 1]$,

$$\mathsf{P}_0, \mathsf{P}_1 \in \mathscr{K}_d \ \Rightarrow\ \mathsf{P}_\lambda(\mathrm{d}y|\boldsymbol{x}) = (1 - \lambda)\mathsf{P}_0(\mathrm{d}y|\boldsymbol{x}) + \lambda\mathsf{P}_1(\mathrm{d}y|\boldsymbol{x}) \in \mathscr{K}_d. \tag{K.9}$$

We are interested in a data selection scheme that works well uniformly over the uncertainty encoded in $\mathscr{K}_d$.

Given a probability kernel $\mathsf{P}(\mathrm{d}y|\boldsymbol{x})$ (i.e. $\mathsf{P} : \mathcal{B}_{\mathbb{R}} \times \mathbb{R}^d \to [0, 1]$), we write $\mathbb{P}(\mathsf{P})$ for the data distribution (on $\mathbb{R}^d \times \mathbb{R}$) induced by $\mathsf{P}$. Namely $\mathbb{E}_{\mathbb{P}(\mathsf{P})} F(y, \boldsymbol{x}) := \mathbb{E}_{\boldsymbol{x}}\{\int F(y, \boldsymbol{x}) \mathsf{P}(\mathrm{d}y|\boldsymbol{x})\}$.

Let $R_{\text{test}}(\boldsymbol{\theta}) = \mathbb{E}L_{\text{test}}(\boldsymbol{\theta}; y, \boldsymbol{x})$ be the test error with respect to a certain target distribution $\mathbb{P}$. Given an estimator $\hat{\boldsymbol{\theta}}$, we let $\hat{\boldsymbol{\theta}}_S(\boldsymbol{y}, \boldsymbol{X})$ denote its output when applied to data $\boldsymbol{y}, \boldsymbol{X}$ in conjunction with data selection scheme $S$. For clarity of notation, we define

$$R_{\#}(S; \boldsymbol{y}, \boldsymbol{X}) := R_{\text{test}}(\hat{\boldsymbol{\theta}}_S(\boldsymbol{y}, \boldsymbol{X})). \tag{K.10}$$

We define the minimax risk $R_{\text{MM}}(\mathscr{K}_d)$ by (here we recall that $\mathscr{A}$ is the set of all data selection methods)

$$R_*(S; \mathscr{K}_d) := \sup_{\mathsf{P} \in \mathscr{K}_d} \mathbb{E}_{\boldsymbol{y}, \boldsymbol{X} \sim \mathbb{P}(\mathsf{P})} R_{\#}(S; \boldsymbol{y}, \boldsymbol{X}), \tag{K.11}$$

$$R_{\text{MM}}(\mathscr{K}_d) := \inf_{S \in \mathscr{A}} R_*(S; \mathscr{K}_d). \tag{K.12}$$

We seek near optimal data selection schemes $S$, namely schemes such that $R_*(S; \mathscr{K}_d) \approx R_{\text{MM}}(\mathscr{K}_d)$. We note that the expectation in Eq. (K.11) includes expectation over the randomness in $S$.

**Remark K.1.** The set $\mathscr{K}_d$ provides information about $\boldsymbol{\theta}_*$. This information can be exploited in other ways than via data selection. For instance, we could restrict the empirical risk minimization of Eq. (1.2) to a set that is "compatible" with $\mathscr{K}_d$. However, we are only interested in procedures that follow the general data selection framework defined in the previous sections and hence are not necessarily optimal against this broader set.

## K.3 DUALITY AND ITS CONSEQUENCES

We can apply Sion's minimax theorem to a relaxation of $R_{\text{MM}}(\mathscr{K}_d)$. Namely,

- We replace $\mathscr{K}_d$ by a set of probability kernels $\mathbb{R}^{N \times d}$ to $\mathbb{R}^N$:

$$\mathscr{K}_{N,d} \subseteq \mathscr{K}_{N,d}^0 := \{\mathsf{P} : \mathcal{B}_{\mathbb{R}^N} \times \mathbb{R}^{N \times d} \to [0, 1] \text{ probability kernel}\}. \tag{K.13}$$

  such that each marginal of $\mathsf{P} \in \mathscr{K}_{N,s}$ is in $\mathscr{K}_d$. In other words, we allow for entries of $\boldsymbol{y}$ to be conditionally dependent, given $\boldsymbol{X}$. Generalizing the notations above, $\mathbb{P}(\mathsf{P}_N)$ denotes the induced distribution on $\boldsymbol{y}, \boldsymbol{X}$.

- We replace the space of data selection schemes $\mathscr{A}$ by a the set $\overline{\mathscr{A}}$ of probability kernels $\mathsf{Q}$ such that, for any $A \subseteq [N]$, and any $\boldsymbol{X} \in \mathbb{R}^{N \times d}$, $\mathsf{Q}(A|\boldsymbol{X})$ is the conditional probability of selecting data in the set $A$ given covariate vectors $\boldsymbol{X}$. In other words, we consider more general data-selection schemes in which the selected set is allowed to depend on all the data points.

The following result is an application of the standard minimax theorem in statistical decision theory, see e.g. Liese & Miescke (2008, Section 3.7).

**Theorem 7.** *Assume that any $\mathsf{P}_N \in \mathscr{K}_{d,N}$ is supported on $\|\boldsymbol{y}\| \leq M$, and that $(\boldsymbol{y}, \boldsymbol{X}) \mapsto R(\hat{\boldsymbol{\theta}}_A(\boldsymbol{y}, \boldsymbol{X}))$ is continuous for any $A$. Define*

$$\overline{R}_{\text{MM}}(\mathscr{K}_d) := \inf_{S \in \overline{\mathscr{A}}} \overline{R}_*(S; \mathscr{K}_d) := \inf_{S \in \overline{\mathscr{A}}} \sup_{\mathsf{P}_N \in \mathscr{K}_{d,N}} \mathbb{E}_{\boldsymbol{y}, \boldsymbol{X} \sim \mathbb{P}(\mathsf{P}_N)} R_{\#}(S; \boldsymbol{y}, \boldsymbol{X}). \tag{K.14}$$

*Then we have*

$$\overline{R}_{\text{MM}}(\mathscr{K}_d) = \sup_{\mathsf{P}_N \in \mathscr{K}_{d,N}} \inf_{S \in \overline{\mathscr{A}}} \mathbb{E}_{\boldsymbol{y}, \boldsymbol{X} \sim \mathbb{P}(\mathsf{P}_N)} R_{\#}(S; \boldsymbol{y}, \boldsymbol{X}). \tag{K.15}$$

*Further, assume $\mathsf{P}_{\text{MM}}$ achieves the supremum over $\mathscr{K}_d$ above. Then any*

$$S_{\text{MM}} \in \arg \min_{S \in \overline{\mathscr{A}}} \mathbb{E}_{\boldsymbol{y}, \boldsymbol{X} \sim \mathbb{P}(\mathsf{P}_{\text{MM}})} R_{\#}(S; \boldsymbol{y}, \boldsymbol{X}) \tag{K.16}$$

*achieves the minimax error.*

As is common in estimation theory, the minimax theorem can be difficult to apply since computing the supremum over $\mathsf{P} \in \mathscr{K}_d$ is in general very difficult. Nevertheless, the theorem implies the following important insight. We should not perform data selection by plugging in the surrogate conditional model for $y$ given $x$ for the actual one. Instead, we should optimize data selection as if labels were distributed according to the 'worst' conditional model in a neighborhood of the surrogate.

Below we work out a toy case to illustrate this insight. Instead of studying the minimax problem for the finite-sample risk $R$, we will consider its asymptotics defined in Proposition B.1. We define the asymptotic minimax coefficients $\rho_*(S; \mathscr{K}_d)$ and $\rho_{\mathrm{MM}}(\mathscr{K}_d)$ in analogy with Eqs. (K.11) and (K.12).

**Example K.2** (Discrete covariates). Consider $x$ taking values in $[k] = \{1, \ldots, k\}$, with probabilities $p_\ell = \mathbb{P}(x = \ell)$, and $y$ taking values in $\{0, 1\}$, with $\mathbb{P}(y = 1|x) = \theta_x^*$, whereby $\boldsymbol{\theta}^* = (\theta_1^*, \ldots, \theta_k^*)$ is a vector of unknown parameters. We estimate $\boldsymbol{\theta}^*$ using empirical risk minimization with respect to log-loss

$$L(\boldsymbol{\theta}; y, x) = -y \log \theta_x - (1 - y) \log(1 - \theta_x). \tag{K.17}$$

We are interested in the quadratic estimation error $\|\hat{\boldsymbol{\theta}} - \boldsymbol{\theta}^*\|_{\boldsymbol{Q}}$ with $\boldsymbol{Q} = \mathrm{diag}(q_1, q_2, \ldots, q_k)$. We consider a non-reweighting subsampling scheme whereby a sample with covariate $x$ is retained independently with probability $\pi_x$. Either applying Proposition B.1, or by a straightforward calculation, we obtain:

$$\mathbb{E}\{\|\hat{\boldsymbol{\theta}}^S - \boldsymbol{\theta}^*\|_{\boldsymbol{Q}}^2\} = \frac{1}{N} \rho(\pi; \boldsymbol{\theta}^*, \boldsymbol{Q}) + o(1/N), \tag{K.18}$$

$$\rho(\pi; \boldsymbol{\theta}^*, \boldsymbol{Q}) = \sum_{x=1}^k \frac{\theta_x^*(1 - \theta_x^*)}{\pi_x p_x} q_x. \tag{K.19}$$

(Here we made explicit the dependence on $\boldsymbol{\theta}^*$.) For $\mathscr{K} \subseteq \mathbb{R}^k$ a convex set, we then define[5]

$$\rho_*(\pi; \mathscr{K}, \boldsymbol{Q}) := \sup_{\boldsymbol{\theta}^* \in \mathscr{K}} \rho(\pi; \boldsymbol{\theta}^*, \boldsymbol{Q}), \qquad \rho_{\mathrm{MM}}(\mathscr{K}, \boldsymbol{Q}) := \inf_\pi \rho_*(\pi; \mathscr{K}, \boldsymbol{Q}). \tag{K.20}$$

We will assume that $\mathscr{K}$ is closed (hence compact) and convex. We can apply the minimax theorem to Eq. (K.20):

$$\rho_{\mathrm{MM}}(\mathscr{K}, \boldsymbol{Q}) := \max_{\boldsymbol{\theta}^* \in \mathscr{K}} \min_\pi \rho(\pi; \boldsymbol{\theta}^*, \boldsymbol{Q}). \tag{K.21}$$

Hence, the minimax optimal data selection strategy is obtained by selecting the optimum $\pi$ for the worst case $\boldsymbol{\theta}$, to be denoted by $\boldsymbol{\theta}^{\mathrm{MM}}$. A simple calculation yields

$$\pi_x^{\mathrm{MM}} = \min\left(\frac{c(\gamma)}{p_x}\sqrt{q_x \theta_x^{\mathrm{MM}}(1 - \theta_x^{\mathrm{MM}})}; 1\right), \tag{K.22}$$

$$\boldsymbol{\theta}_s^{\mathrm{MM}} = \arg\max_{\boldsymbol{\theta} \in \mathscr{K}} \sum_{x=1}^k \max\left(\frac{1}{c(\gamma)}\sqrt{q_x \theta_x(1 - \theta_x)}; \frac{q_x}{p_x}\theta_x(1 - \theta_x)\right), \tag{K.23}$$

where $c(\gamma)$ is obtained by solving

$$\sum_{x=1}^k \min\left(c(\gamma)\sqrt{q_x \theta_x^{\mathrm{MM}}(1 - \theta_x^{\mathrm{MM}})}; p_x\right) = \gamma. \tag{K.24}$$

The above formulas have a clear interpretation (for simplicity we neglect the factor $q_x$). Note that $\theta_x^{\mathrm{MM}}(1 - \theta_x^{\mathrm{MM}})$ can be interpreted as measure of uncertainty in predicting $y_{\mathrm{new}}$ at a test point $x_{\mathrm{new}} = x$ under the minimax model $\boldsymbol{\theta}^{\mathrm{MM}}$. We select data so that the fraction of samples with $x_i = x$ is proportional to the square root of this uncertainty. Assuming that the inner maximum in Eq. (K.23) is achieved on the first argument, the minimax model itself is the one that maximize total uncertainty within the set $\mathscr{K}$.

---

[5]Notice that we are not taking the infimum over all randomized data selection schemes as in Theorem 7. For simplicity, we are restricting the minimization to non-reweighting schemes. The substance of Theorem 7 does not change since this set is convex.

For instance, if $\mathscr{K} = [\theta_{\text{su},1} - \varepsilon, \theta_{\text{su},1} + \varepsilon] \times \cdots \times [\theta_{\text{su},k} - \varepsilon, \theta_{\text{su},k} + \varepsilon] \subseteq [0, 1/2]^k$, then $\theta_x^{\text{MM}} = \theta_{\text{su},x} + \varepsilon$ for all $x$. We then obtain

$$\pi_x^{\text{MM}} = \min\Big(\frac{c(\gamma)}{p_x}\sqrt{q_x(\theta_{\text{su},x} + \varepsilon)(1 - \theta_{\text{su},x} - \varepsilon)}; 1\Big) \tag{K.25}$$

In other words, we should not use the uncertainties computed within the surrogate model, but within the most uncertain model in its neighborhood.

## L  HIGH-DIMENSIONAL ASYMPTOTICS: GENERAL VERSION OF THEOREM 4

In this section, we present the complete version of the Theorem 4 followed by the proof.

**Theorem 8.** *Assume $u \mapsto L(u, y)$ is convex, continuous, with at most quadratic growth, and $\lambda > 0$. Further denote by $\boldsymbol{\alpha}^*, \mu^*$ the solution of the following minimax problem ($\boldsymbol{\alpha}^*$ is uniquely defined by this condition)*

$$\min_{\boldsymbol{\alpha}} \max_{\mu \geq 0} \mathscr{L}(\boldsymbol{\alpha}, \mu). \tag{L.1}$$

*Then the following hold in the limit $N, p \to \infty$, with $N/p \to \delta_0$: If $(u, z) \mapsto \int L_{\text{test}}(u; y)\mathsf{P}(\mathrm{d}y|z)$ is a continuous function with at most quadratic growth, we have*

$$\operatorname*{p\text{-}lim}_{N,p\to\infty} R_{\text{test}}(\hat{\boldsymbol{\theta}}_\lambda) = \mathbb{E}L_{\text{test}}\Big(\alpha_0^* G_0 + \sqrt{(\alpha_s^*)^2 + (\alpha_\perp^*)^2}G; Y\Big), \tag{L.2}$$

*where expectation is taken with respect to the joint distribution of Eq.* (5.3).

*(b) If $\boldsymbol{\theta}_0 \in \arg\min R_{\text{test}}(\boldsymbol{\theta})$, then the excess risk is given by*

$$\operatorname*{p\text{-}lim}_{N,p\to\infty} R_{\text{exc}}(\hat{\boldsymbol{\theta}}_\lambda) = \mathbb{E}L_{\text{test}}\Big(\alpha_0^* G_0 + \sqrt{(\alpha_s^*)^2 + (\alpha_\perp^*)^2}G; Y\Big) - \mathbb{E}L_{\text{test}}\Big(c_* G_0; Y\Big). \tag{L.3}$$

*(See Footnote 3 for a definition of $c_*$.)*

*(c) Letting $\boldsymbol{P}_0^\perp$ be the projector orthogonal to $\boldsymbol{\theta}_0$ and $\boldsymbol{P}^\perp$ the projector orthogonal to $\operatorname{span}(\boldsymbol{\theta}_0, \hat{\boldsymbol{\theta}}^{\text{su}})$, we have*

$$\operatorname*{p\text{-}lim}_{N,p\to\infty} \frac{\langle\hat{\boldsymbol{\theta}}_\lambda, \boldsymbol{\theta}_0\rangle}{\|\boldsymbol{\theta}_0\|} = \alpha_0^*, \qquad \operatorname*{p\text{-}lim}_{N,p\to\infty} \frac{\langle\hat{\boldsymbol{\theta}}_\lambda, \boldsymbol{P}_0^\perp\hat{\boldsymbol{\theta}}^{\text{su}}\rangle}{\|\boldsymbol{P}_0^\perp\hat{\boldsymbol{\theta}}^{\text{su}}\|} = \alpha_s^*, \qquad \operatorname*{p\text{-}lim}_{N,p\to\infty} \|\boldsymbol{P}^\perp\hat{\boldsymbol{\theta}}_\lambda\| = \alpha_\perp^*. \tag{L.4}$$

*(d) Further, the asymptotic subsampling fraction is given by*

$$\frac{n}{N} \to \gamma = \mathbb{P}(S(\beta_0 G_0 + \beta_s G_s) > 0). \tag{L.5}$$

We can further specialize the above formulas to the two classes of data selection schemes studied before: unbiased and non-reweighting schemes.

**Unbiased data selection.** In this case $S_i(\langle\hat{\boldsymbol{\theta}}^{\text{su}}, \boldsymbol{x}_i\rangle) = 1/\pi(\langle\hat{\boldsymbol{\theta}}^{\text{su}}, \boldsymbol{x}_i\rangle)$ with probability $\pi(\langle\hat{\boldsymbol{\theta}}^{\text{su}}, \boldsymbol{x}_i\rangle)$ and $S_i(\langle\hat{\boldsymbol{\theta}}^{\text{su}}, \boldsymbol{x}_i\rangle) = 0$ otherwise. The Lagrangian reduces to

$$\mathscr{L}(\boldsymbol{\alpha}, \mu) := \frac{\lambda}{2}\|\boldsymbol{\alpha}\|^2 - \frac{1}{2\delta_0}\mu\alpha_\perp^2 + \mathbb{E}\Big\{\min_{u\in\mathbb{R}}\Big[L(\alpha_0 G_0 + \alpha_s G_s + u; Y) + \frac{1}{2}\mu\,\pi(\langle\boldsymbol{\beta}, \boldsymbol{g}\rangle)\big(u - \alpha_\perp G_\perp\big)^2\Big]\Big\}.$$

**Non-reweighting data selection.** In this case $S_i(\langle\hat{\boldsymbol{\theta}}^{\text{su}}, \boldsymbol{x}_i\rangle) = 1$ with probability $\pi(\langle\hat{\boldsymbol{\theta}}^{\text{su}}, \boldsymbol{x}_i\rangle)$ and $S_i(\langle\hat{\boldsymbol{\theta}}^{\text{su}}, \boldsymbol{x}_i\rangle) = 0$ otherwise. The Lagrangian reduces to

$$\mathscr{L}(\boldsymbol{\alpha}, \mu) := \frac{\lambda}{2}\|\boldsymbol{\alpha}\|^2 - \frac{1}{2\delta_0}\mu\alpha_\perp^2 + \mathbb{E}\Big\{\pi(\langle\boldsymbol{\beta}, \boldsymbol{g}\rangle)\min_{u\in\mathbb{R}}\Big[L(\alpha_0 G_0 + \alpha_s G_s + u; Y) + \frac{1}{2}\mu\big(u - \alpha_\perp G_\perp\big)^2\Big]\Big\}. \tag{L.6}$$

We further provide simple expressions in the case of misspecified linear regression, already studied in Section 4.3, in Appendix M.

**Proof of Theorem 8**  This proof is based on Gordon Gaussian comparison inequality, following a well established technique, see Thrampoulidis et al. (2015; 2018); Miolane & Montanari (2021). Our presentation will be succinct, emphasizing novelties with respect to earlier derivations of this type.

Recall that we consider test error of the form

$$R_{\text{test}}(\hat{\boldsymbol{\theta}}_\lambda) = \mathbb{E}L_{\text{test}}(\langle \hat{\boldsymbol{\theta}}_\lambda, \boldsymbol{x}_{\text{new}}\rangle, y_{\text{new}}), \tag{L.7}$$

where $\hat{\boldsymbol{\theta}}_\lambda := \arg\min_{\boldsymbol{\theta}} \hat{R}_N(\boldsymbol{\theta})$. We will also consider the excess error $R_{\text{exc}}(\hat{\boldsymbol{\theta}}_\lambda) := R_{\text{test}}(\hat{\boldsymbol{\theta}}_\lambda) - \inf_{\boldsymbol{\theta}} R_{\text{test}}(\boldsymbol{\theta})$.

Define

$$G_{0,i} := \frac{\langle \boldsymbol{x}_i, \boldsymbol{\theta}_0 \rangle}{\|\boldsymbol{\theta}_0\|_2}, \qquad G_{s,i} := \frac{\langle \boldsymbol{x}_i, \boldsymbol{P}_0^\perp \hat{\boldsymbol{\theta}}^{\text{su}} \rangle}{\|\boldsymbol{P}_0^\perp \hat{\boldsymbol{\theta}}^{\text{su}}\|_2}, \qquad \boldsymbol{g}_i := \boldsymbol{P}^\perp \boldsymbol{x}_i, \tag{L.8}$$

where we recall that $\boldsymbol{P}_0^\perp$ is the projector orthogonal to $\boldsymbol{\theta}_0$ and $\boldsymbol{P}^\perp$ is the projector orthogonal to $\text{span}(\boldsymbol{\theta}_0, \hat{\boldsymbol{\theta}}^{\text{su}})$. Further define

$$\alpha_0 := \frac{\langle \boldsymbol{\theta}, \boldsymbol{\theta}_0 \rangle}{\|\boldsymbol{\theta}_0\|_2}, \qquad \alpha_s := \frac{\langle \boldsymbol{\theta}, \boldsymbol{P}_0^\perp \hat{\boldsymbol{\theta}}^{\text{su}} \rangle}{\|\boldsymbol{P}_0^\perp \hat{\boldsymbol{\theta}}^{\text{su}}\|_2}, \qquad \overline{\boldsymbol{\alpha}}_\perp := \boldsymbol{P}_{0,s}^\perp \boldsymbol{\theta}, \tag{L.9}$$

With a slight abuse of notation, we can then identify the empirical risk

$$\hat{R}_N(\alpha_0, \alpha_s, \overline{\boldsymbol{\alpha}}_\perp) = \frac{1}{N}\sum_{i=1}^{N} s(\beta_0 G_{0,i} + \beta_s G_{s,i}, U_i)\, L(\alpha_0 G_{0,i} + \alpha_s G_{s,i} + \langle \overline{\boldsymbol{\alpha}}_\perp, \boldsymbol{g}_i \rangle, y_i) +$$

$$\frac{\lambda}{2}(\alpha_0^2 + \alpha_s^2 + \|\overline{\boldsymbol{\alpha}}_\perp\|^2). \tag{L.10}$$

We can identify $\overline{\boldsymbol{\alpha}}_\perp$ and $\boldsymbol{g}_i$ with $(p-2)$-dimensional vectors. For any closed set $\Omega \subseteq \mathbb{R}^p$, let

$$\hat{R}_N(\Omega) := \min\left\{ \hat{R}_N(\alpha_0, \alpha_s, \overline{\boldsymbol{\alpha}}_\perp) : (\alpha_0, \alpha_s, \overline{\boldsymbol{\alpha}}_\perp) \in \Omega \right\}. \tag{L.11}$$

Further define

$$\hat{R}_N^G(\Omega) = \min_{(\alpha_0, \alpha_s, \overline{\boldsymbol{\alpha}}_\perp) \in \Omega}\ \min_{\boldsymbol{v}\in\mathbb{R}^n}\ \max_{\boldsymbol{\xi}\in\mathbb{R}^N}\ \widehat{\mathscr{L}}_N^{(0)}(\alpha_0, \alpha_s, \overline{\boldsymbol{\alpha}}_\perp; \boldsymbol{v}; \boldsymbol{\xi}) \tag{L.12}$$

$$= \min_{(\alpha_0, \alpha_s, \overline{\boldsymbol{\alpha}}_\perp) \in \Omega}\ \max_{\boldsymbol{\xi}\in\mathbb{R}^N}\ \min_{\boldsymbol{v}\in\mathbb{R}^n}\ \widehat{\mathscr{L}}_N^{(0)}(\alpha_0, \alpha_s, \overline{\boldsymbol{\alpha}}_\perp; \boldsymbol{v}; \boldsymbol{\xi}), \tag{L.13}$$

where

$$\widehat{\mathscr{L}}_N^{(0)}(\alpha_0, \alpha_s, \overline{\boldsymbol{\alpha}}_\perp; \boldsymbol{v}; \boldsymbol{\xi}) := \|\overline{\boldsymbol{\alpha}}_\perp\|\langle \boldsymbol{g}_\perp, \boldsymbol{\xi}\rangle + \|\boldsymbol{\xi}\|\langle \boldsymbol{h}, \overline{\boldsymbol{\alpha}}_\perp\rangle - \langle \boldsymbol{v}, \boldsymbol{\xi}\rangle + \frac{\lambda}{2}(\alpha_0^2 + \alpha_s^2 + \|\overline{\boldsymbol{\alpha}}_\perp\|^2) \tag{L.14}$$

$$+ \frac{1}{N}\sum_{i=1}^{N} s(\beta_0 G_{0,i} + \beta_s G_{s,i}, U_i)\, L(\alpha_0 G_{0,i} + \alpha_s G_{s,i} + v_i, y_i),$$

and the identity of the two lines (L.12), (L.13) holds by the minimax theorem. By an application of Gordon's inequality and using Gaussian concentration (Boucheron et al., 2013), we obtain the following.

**Lemma L.1.** *There exist subgaussian error terms, $\text{err}_1(N, \Omega), \text{err}_2(N, \Omega)$ (with $\|\text{err}_2(N, \Omega)\|_{\psi_2} \leq CN^{-1/2}$, such that:*

1. *For any closed set $\Omega$:*

$$\hat{R}_N(\Omega) \geq \hat{R}_N^G(\Omega) + \text{err}_1(N).$$

2. *For any closed convex set $\Omega$:*

$$\hat{R}_N(\Omega) = \hat{R}_N^G(\Omega) + \text{err}_2(N).$$

*Proof.* This follows from an application of Gordon's inequality and using Gaussian concentration (Thrampoulidis et al., 2015; 2018; Miolane & Montanari, 2021). In applying Gordon's inequality, we need to check that the minimizer $\overline{\boldsymbol{\alpha}}_\perp$ lie with high probability in a compact set $B_N$. This is immediate for $\lambda > 0$ by strong convexity of $\hat{R}_N$. $\square$

Note that, subject to the constraints $\|\overline{\boldsymbol{\alpha}}_\perp\| = \alpha_\perp$, $\|\boldsymbol{\xi}\| = \mu/\sqrt{N}$, the minimization over $\overline{\boldsymbol{\alpha}}_\perp$ and maximization over $\boldsymbol{\xi}$ can be performed before the other optimizations. This yields the reduced Lagrangian, with argument $\boldsymbol{\alpha} := (\alpha_0, \alpha_s, \alpha_\perp)$:

$$\widehat{\mathscr{L}}_N^{(1)}(\boldsymbol{\alpha}; \mu, \boldsymbol{v}) := -\frac{\|\boldsymbol{h}\|}{\sqrt{N}}\alpha_\perp \mu + \frac{\mu}{\sqrt{N}}\|\alpha_\perp \boldsymbol{g}_\perp - \boldsymbol{v}\| + \frac{\lambda}{2}\|\boldsymbol{\alpha}\|^2 \tag{L.15}$$
$$+ \frac{1}{N}\sum_{i=1}^N s(\beta_0 G_{0,i} + \beta_s G_{s,i}, U_i) L(\alpha_0 G_{0,i} + \alpha_s G_{s,i} + v_i, y_i).$$

For $A \in \mathbb{R}^2 \times \mathbb{R}_{\geq 0}$, let $\Omega_A := \{(\alpha_0, \alpha_s, \overline{\boldsymbol{\alpha}}_\perp) \in \mathbb{R}^p : (\alpha_0, \alpha_s, \|\overline{\boldsymbol{\alpha}}_\perp\|) \in A\}$, and write

$$\hat{R}_{\#,N}(A) := \hat{R}_N(\Omega_A) = \min\left\{\hat{R}_N(\alpha_0, \alpha_s, \overline{\boldsymbol{\alpha}}_\perp) : (\alpha_0, \alpha_s, \|\overline{\boldsymbol{\alpha}}_\perp\|) \in A\right\}, \tag{L.16}$$
$$\hat{R}_{\#,N}^G(A) := \hat{R}_N^G(\Omega_A). \tag{L.17}$$

We then have

$$\hat{R}_{\#,N}^G(A) = \min_{(\alpha_0, \alpha_s, \alpha_\perp) \in A} \max_{\mu \in \mathbb{R}_{\geq 0}} \min_{\boldsymbol{v} \in \mathbb{R}^n} \widehat{\mathscr{L}}_N^{(1)}(\alpha_0, \alpha_s, \alpha_\perp; \mu, \boldsymbol{v}), \tag{L.18}$$

Finally, we can take the limit $N, p \to \infty$. In this limit, the minimization over $\boldsymbol{v}$ is replaced by minimization over a random variable $V$. Namely, let $(\mathcal{S}, \mathcal{F}, \mathsf{P})$ be a probability space on which the random variables $(G_0, G_s, G_\perp, U, Y)$ are defined with the same joint law of $(G_{0,1}, G_{s,1}, g_{\perp,1}, U_1, y_1)$. Namely $(G_0, G_s, G_\perp, U) \sim \mathsf{N}(0,1)^{\otimes 3} \otimes \mathsf{Unif}([0,1])$, and $Y|G_0, G_s, G_\perp, U \sim \mathsf{P}(\cdot \,|\, \|\boldsymbol{\theta}_0\| G_0)$. For $V$ another random variable in the same space, taking values in the extended real line $\overline{\mathbb{R}}$, and letting $\boldsymbol{\alpha} := (\alpha_0, \alpha_s, \alpha_\perp)$, define

$$\widehat{\mathscr{L}}(\boldsymbol{\alpha}; \mu, V) := -\frac{1}{\sqrt{\delta_0}}\alpha_\perp \mu + \mathbb{E}\{(\alpha_\perp G_\perp - V)^2\}^{1/2}\mu \tag{L.19}$$
$$+ \frac{\lambda}{2}\|\boldsymbol{\alpha}\|^2 + \mathbb{E}\left\{s(\beta_0 G_0 + \beta_s G_s, U) L(\alpha_0 G_0 + \alpha_s G_s + V, Y)\right\}.$$

**Theorem 9.** *With the definitions given above, and under the assumptions of Theorem 4, the following hold:*

1. *For any compact set $A \subseteq \mathbb{R}^2 \times \mathbb{R}_{\geq 0}$*

$$\lim_{N,p\to\infty} \hat{R}_{\#,N}^G(A) = \hat{R}_\#^G(A) := \min_{\boldsymbol{\alpha} \in A} \max_{\mu \in \mathbb{R}_{\geq 0}} \min_{V \in m\mathcal{F}} \widehat{\mathscr{L}}(\boldsymbol{\alpha}; \mu, V). \tag{L.20}$$

2. *For any closed set $A$,*

$$\lim_{N\to\infty}\inf \hat{R}_{\#,N}(A) \geq \hat{R}_\#^G(A). \tag{L.21}$$

3. *For any closed convex set $A$,*

$$\lim_{N\to\infty}\inf \hat{R}_{\#,N}(A) = \hat{R}_\#^G(A). \tag{L.22}$$

4. *Further, denoting by $\boldsymbol{\alpha}^* := (\alpha_0^*, \alpha_s^*, \alpha_\perp^*)$ the minimizer of*

$$\hat{R}_\#^G(\boldsymbol{\alpha}) := \max_{\mu \in \mathbb{R}_{\geq 0}} \min_{V \in m\mathcal{F}} \widehat{\mathscr{L}}(\boldsymbol{\alpha}; \mu, V)$$

*Conclusions $(a)$ to $(d)$ of Theorem 4 hold.*

*Proof.* Note that we can rewrite $\hat{R}_{\#,N}(A) = \min\{\hat{R}_{\#,N}(\boldsymbol{\alpha}) : \boldsymbol{\alpha} \in A\}$, where

$$
\hat{R}_{\#,N}(\boldsymbol{\alpha}) = \begin{cases} \min & \frac{\lambda}{2}\|\boldsymbol{\alpha}\|^2 + \frac{1}{N}\sum_{i=1}^{N} s(\beta_0 G_{0,i} + \beta_s G_{s,i}, U_i)\, L(\alpha_0 G_{0,i} + \alpha_s G_{s,i} + v_i, y_i) \\ \text{subj. to} & \big\|\alpha_\perp \boldsymbol{g}_\perp - \boldsymbol{v}\big\| \le \|\boldsymbol{h}\|\alpha_\perp\,. \end{cases}
$$
(L.23)

Further, this can be written as as a function of the joint empirical distribution of $\{(G_{0,i}, G_{s,i}, g_{\perp,i}, U_1, y_i, v_i)\}$. Namely, defining

$$
\hat{p}_N := \frac{1}{N}\sum_{i=1}^{N}\delta_{(G_{0,i},G_{s,i},g_{\perp,i},U_1,y_i,v_i)}\,,
$$
(L.24)

we have (with $\mathbb{E}_{\hat{p}_N}$ denoting expectation with respect to $\hat{p}_N$)

$$
\hat{R}_{\#,N}(\boldsymbol{\alpha}) = \begin{cases} \min & \frac{\lambda}{2}\|\boldsymbol{\alpha}\|^2 + \mathbb{E}_{\hat{p}_N}\big\{s(\beta_0 G_0 + \beta_s G_s, U)\, L(\alpha_0 G_0 + \alpha_s G_s + V, y_i)\big\} \\ \text{subj. to} & \mathbb{E}_{\hat{p}_N}\big\{[\alpha_\perp G_\perp - V]^2\big\} \le \frac{\|\boldsymbol{h}\|^2}{N}\alpha_\perp^2\,. \end{cases}
$$
(L.25)

Let $\hat{R}_\#(\boldsymbol{\alpha})$ be the same quantity, in which minimization over $\boldsymbol{v}$ is replaced by minimization over random variables $(G_0, G_s, G_\perp, U, Y, V)$ with $(G_0, G_s, G_\perp, U, Y)$ having the prescribed $N = \infty$ distribution, and $\|\boldsymbol{h}\|^2/N$ is replaced by $1/\delta_0$. In fact, se define a slight generalization

$$
\hat{R}_\#(\boldsymbol{\alpha};\varepsilon) = \begin{cases} \min & \frac{\lambda}{2}\|\boldsymbol{\alpha}\|^2 + \mathbb{E}\big\{s(\beta_0 G_0 + \beta_s G_s, U)\, L(\alpha_0 G_0 + \alpha_s G_s + V, y_i)\big\} \\ \text{subj. to} & \mathbb{E}\big\{[\alpha_\perp G_\perp - V]^2\big\} \le (1-\varepsilon)\alpha_\perp^2\,. \end{cases}
$$
(L.26)

Let $\hat{p}_N^*$ be the joint distribution of that achieves the above minimum at finite $N$. By tightness, $\hat{p}_N^* \Rightarrow \hat{p}^*$ along subsequences, and further the limit satisfies $\mathbb{E}\big\{[\alpha_\perp G_\perp - V]^2\big\} \le \alpha_\perp^2/\delta_0$ by Fatou's. Therefore

$$
\liminf_{N\to\infty} \hat{R}_{\#,N}(\boldsymbol{\alpha}) \ge \hat{R}_\#(\boldsymbol{\alpha};0)\,,
$$
(L.27)

almost surely. On the other hand, if $(G_0, G_s, G_\perp, U, Y, V)$ achieves the minimum to define $\hat{R}_\#(\boldsymbol{\alpha};\varepsilon)$, $\varepsilon > 0$, let $(G_{0,i}, G_{s,i}, G_{\perp,i}, U_i, y_i, v_i)$, $i \le N$ be i.i.d.'s vectors from this distribution. Of course this $\boldsymbol{v}$ is almost surely feasible for problem $\hat{R}_{\#,N}(\boldsymbol{\alpha})$ for all $N$ large enough. Therefore

$$
\limsup_{N\to\infty} \hat{R}_{\#,N}(\boldsymbol{\alpha}) \le \hat{R}_\#(\boldsymbol{\alpha};\varepsilon)\,,
$$
(L.28)

Finally, we claim that $\lim_{\varepsilon\to 0+} \hat{R}_\#(\boldsymbol{\alpha};\varepsilon) = \hat{R}_\#(\boldsymbol{\alpha};0)$, thus yielding

$$
\lim_{N\to\infty} \hat{R}_{\#,N}(\boldsymbol{\alpha}) \le \hat{R}_\#(\boldsymbol{\alpha};0) =: \hat{R}_\#(\boldsymbol{\alpha})\,,
$$
(L.29)

To prove the claim notice that, if $V$ satisfies $\mathbb{E}\big\{[\alpha_\perp G_\perp - V]^2\big\} \le \alpha_\perp^2$, then $V_\delta = (1-\delta)V + \alpha_\perp \delta G_\perp$ satisfies $\mathbb{E}\big\{[\alpha_\perp G_\perp - V_\delta]^2\big\} \le (1-\varepsilon)\alpha_\perp^2$, with $\varepsilon = 2\delta - \delta^2$. Further, the objective is continuous as $\delta \to 0$ by continuity of $L$, whence the claim follows.

Next we claim that $\boldsymbol{\alpha} \mapsto \hat{R}_{\#,N}(\boldsymbol{\alpha})$ is Lipschitz continuous for $\boldsymbol{a} \in A$, where $A$ is a compact set, on the high probability event

$$
\mathcal{G} := \Big\{ \sum_{i=1}^{N}(G_{0,i}^2 + G_{s,i}^2 + G_{\perp,i}^2) \le 10N, \quad \frac{N}{2} \le \|\boldsymbol{h}\|^2 \le 2N \Big\}\,.
$$
(L.30)

To prove this claim, note that on this event, define $s_i = s(\beta_0 G_{0,i} + \beta_s G_{s,i}, U_i)$, $a_i(\boldsymbol{\alpha}) = \alpha_0 G_{0,i} + \alpha_s G_{s,i}$, $L_i(u) = L_i(u; y_i)$, $r = \|\boldsymbol{h}\|/\sqrt{N}$. such that $\|\boldsymbol{s}\|, \|\boldsymbol{a}\| \le C\sqrt{N}$, $1/2 \le r \le 2$, it is

sufficient to prove that $\boldsymbol{\alpha} \mapsto F(\boldsymbol{\alpha})$ is Lipschitz on $A$ where

$$F(\boldsymbol{\alpha}) := \min \left\{ H(\boldsymbol{\alpha}, \boldsymbol{v}) \ : \ \boldsymbol{v} \in S(\boldsymbol{\alpha}) \right\}, \tag{L.31}$$

$$H(\boldsymbol{\alpha}, \boldsymbol{v}) := \frac{1}{N} \sum_{i=1}^{N} s_i \, L_i(a_i(\boldsymbol{\alpha}) + v_i), \quad S(\boldsymbol{\alpha}) := \{ \boldsymbol{v} : \ \|\boldsymbol{v} - \alpha_\perp \boldsymbol{g}_\perp\| \le \alpha_\perp \sqrt{N} \}. \tag{L.32}$$

On the event $\mathcal{G}$ above, $|H(\boldsymbol{\alpha}_1, \boldsymbol{v}_1) - H(\boldsymbol{\alpha}_2, \boldsymbol{v}_2)| \le C\|\boldsymbol{\alpha}_1 - \boldsymbol{\alpha}_2\| + C\|\boldsymbol{v}_1 - \boldsymbol{v}_2\|/\sqrt{N}$ and $\mathrm{dist}(S(\boldsymbol{\alpha}_1), S(\boldsymbol{\alpha}_2)) \le C\sqrt{N}\|\boldsymbol{\alpha}_1 - \boldsymbol{\alpha}_2\|$[6]. The claim follows from these bounds.

Since $\hat{R}_{\#,N}(\boldsymbol{\alpha}) \to \hat{R}_\#(\boldsymbol{\alpha})$ for each $\alpha \in A$, $\boldsymbol{\alpha} \mapsto \hat{R}_{\#,N}(\boldsymbol{\alpha})$ is almost surely Lipschitz continuous with Lipshitz constant $C$ independent of $N$, for all $N$ large enough, we have $\sup_{\boldsymbol{\alpha} \in A} |\hat{R}_{\#,N}(\boldsymbol{\alpha}) - \hat{R}_\#(\boldsymbol{\alpha})| \to 0$, and therefore, for any compact set $\boldsymbol{A}$,

$$\lim_{N \to \infty} \hat{R}_{\#,N}(A) = \hat{R}_\#(A). \tag{L.33}$$

For $\lambda > 0$, both $\hat{R}_{\#,N}$ and $\hat{R}_{\#,N}$ are uniformly strongly convex and therefore the last claim extend to any closed set $A$. (Because eventually almost surely $\arg\min \hat{R}_{\#,N}(\boldsymbol{\alpha}) \in B$ for some compact $B$.)

Finally, by introducing a Lagrange multiplier for the constraint

$$\hat{R}_\#(\boldsymbol{\alpha}) = \min_{V \in m\mathcal{F}} \max_{\mu \in \mathbb{R}_{\ge 0}} \widehat{\mathscr{L}}(\boldsymbol{\alpha}; \mu, V)$$

$$= \max_{\mu \in \mathbb{R}_{\ge 0}} \min_{V \in m\mathcal{F}} \widehat{\mathscr{L}}(\boldsymbol{\alpha}; \mu, V).$$

This concludes the proof of point 1 (Eq. (L.20)).

Points 2 and 3 follow from the previous one by applying Lemma L.1.

Finally, the proof of point 4 is also straightforward. Indeed by taking $A = \{\boldsymbol{\alpha} \in \mathbb{R}^2 \times \mathbb{R}_{\ge 0} : \|\boldsymbol{\alpha} - \boldsymbol{\alpha}^*\| \ge \varepsilon\}$ and $A = \mathbb{R}^2 \times \mathbb{R}_{\ge 0}$ and applying points 2 and 3, for arbitrary $\varepsilon > 0$, implies Eq. (L.4), thus establishing claim $(c)$ of Theorem 4.

Claims $(a)$, $(b)$, to $(d)$ of Theorem 4 follow from the previous one by noting that $\hat{\boldsymbol{\theta}}_\lambda$ is uniformly random, conditional to $\langle \boldsymbol{\theta}_0, \hat{\boldsymbol{\theta}}_\lambda \rangle, \langle \hat{\boldsymbol{\theta}}^{\mathrm{su}}, \hat{\boldsymbol{\theta}}_\lambda \rangle, \|\boldsymbol{P}^\perp \hat{\boldsymbol{\theta}}_\lambda\|$. $\qquad \square$

The proof of Theorem 4 is completed by showing the following.

**Lemma L.2.** *Under the assumptions of Theorem 4, we have the following equivalent characterizations of $\hat{R}_\#(\boldsymbol{\alpha})$ (where we use the shorthand $S(b) = s(b, U)$ for $U \sim \mathsf{Unif}([0, 1])$):*

$$\hat{R}_\#^G(\boldsymbol{\alpha}) = \max_{\mu \in \mathbb{R}_{\ge 0}} \min_{V \in m\mathcal{F}} \widehat{\mathscr{L}}(\boldsymbol{\alpha}; \mu, V), \tag{L.34}$$

$$\hat{R}_\#(\boldsymbol{\alpha}) = \begin{cases} \min & \frac{\lambda}{2}\|\boldsymbol{\alpha}\|^2 + \mathbb{E}\{S(\beta_0 G_0 + \beta_s G_s) \, L(\alpha_0 G_0 + \alpha_s G_s + V, y_i)\} \\ \text{subj. to} & \mathbb{E}\{[\alpha_\perp G_\perp - V]^2\} \le \alpha_\perp^2, \end{cases} \tag{L.35}$$

$$\hat{R}_\#^G(\boldsymbol{\alpha}) = \max_{\mu \ge 0} \mathscr{L}(\boldsymbol{\alpha}, \mu). \tag{L.36}$$

*Proof.* The equivalence of Eq. (L.34) and (L.35) was already established in the proof of Theorem 9. As for Eq. (L.36), the equivalence with (L.35) follows by introducing a Lagrange multiplier for the inequality $\mathbb{E}\{[\alpha_\perp G_\perp - V]^2\} \le \alpha_\perp^2$, whence $\hat{R}_\#^G(\boldsymbol{\alpha}) = \max_{\mu \ge 0} \mathscr{L}(\boldsymbol{\alpha}, \mu)$

$$\mathscr{L}(\boldsymbol{\alpha}, \mu) = \frac{\lambda}{2}\|\boldsymbol{\alpha}\|^2 - \frac{1}{2\delta_0}\mu\alpha_\perp^2$$

$$+ \min_{V \in m\mathcal{F}} \left\{ \frac{1}{2}\mu \cdot \mathbb{E}\{(\alpha_\perp G_\perp - V)^2\} + \mathbb{E}\{S(\langle \boldsymbol{\beta}, \boldsymbol{G} \rangle) L(\alpha_0 G_0 + \alpha_s G_s + V, Y)\} \right\}$$

$$= \frac{\lambda}{2}\|\boldsymbol{\alpha}\|^2 - \frac{1}{2\delta_0}\mu\alpha_\perp^2 + \mathbb{E}\left\{ \min_{u \in \mathbb{R}} \left[ S(\langle \boldsymbol{\beta}, \boldsymbol{G} \rangle) L(\alpha_0 G_0 + \alpha_s G_s + u, Y) + \frac{1}{2}\mu(\alpha_\perp G_\perp - u)^2 \right] \right\},$$

---

[6]For $S_1, S_2 \subseteq \mathbb{R}^N$, we let $\mathrm{dist}(S_1, S_2) := \sup_{\boldsymbol{x}_1 \in S_1} \sup_{\boldsymbol{x}_2 \in S_2} \|\boldsymbol{x}_1 - \boldsymbol{x}_2\|$.

which yields the desired claim. □

## M    HIGH-DIMENSIONAL ASYMPTOTICS: THE CASE OF MISSPECIFIED LINEAR REGRESSION

We revisit the case of misspecified linear regression, already studied in Section 4.3. We assume a non-reweighting data-selection scheme, whose asymptotic behavior is characterized by the Lagrangian (L.6).

In the case of square loss, the inner minimization over $u$ is easily solved and we can then perform the maximization over $\mu$ in Theorem 4 analytically. This calculation yields the following Lagrangian

$$\mathscr{L}_{\text{ls}}(\boldsymbol{\alpha}) := \frac{1}{2} \left( \sqrt{\mathbb{E}\left\{ \pi(\langle \boldsymbol{\beta}, \boldsymbol{g} \rangle) \left[ Y - \langle \boldsymbol{\alpha}, \boldsymbol{g} \rangle \right]^2 \right\}} - \frac{\alpha_\perp}{\sqrt{\delta_0}} \right)_+^2 + \frac{\lambda}{2} \|\boldsymbol{\alpha}\|_2^2. \tag{M.1}$$

We then have the following consequence of Theorem 4.

**Corollary M.1.** *Assume the misspecified generalized linear model of Section 5.1, and further consider the case of square loss $L(\boldsymbol{\theta}; y, \boldsymbol{x}) = (y - \langle \boldsymbol{\theta}, \boldsymbol{x} \rangle)^2/2$. Let $\boldsymbol{\alpha}^* = (\alpha_0^*, \alpha_s^*, \alpha_\perp^*)$ be the unique minimizer of the Lagrangian $\mathscr{L}_0$. Then claims $(a)$ to $(d)$ of Theorem 4 hold.*

The next statement gives a particularly simple expression in the case of perfect surrogate, and ridgeless limit $\lambda \to 0$. For $n > p$ this is standard least squares, while for $n \le p$, this is minimum $\ell_2$ norm interpolation. Its proof is deferred to Appendix M.1.

**Proposition M.2.** *Assume the misspecified generalized linear model of Section 5.1, and further consider the case of square loss $L(\boldsymbol{\theta}; y, \boldsymbol{x}) = (y - \langle \boldsymbol{\theta}, \boldsymbol{x} \rangle)^2/2$, and further consider the case of a perfect surrogate which, without loss of generality, we assume normalized: $\hat{\boldsymbol{\theta}}^{\text{su}} = \boldsymbol{\theta}_0 / \|\boldsymbol{\theta}_0\|$. Define the quantities*

$$A_\pi := \frac{1}{\gamma} \mathbb{E}[G^2 \pi(G)], \quad B_\pi := \frac{1}{\gamma} \mathbb{E}[GY \pi(G)], \quad C_\pi := \frac{1}{\gamma} \mathbb{E}[Y^2 \pi(G)], \tag{M.2}$$

*where expectation is with respect to $G \sim \mathsf{N}(0,1)$ and $Y \sim \mathsf{P}(\cdot | G)$. In particular, we let $A_1$, $B_1$, $C_1$ be the quantities defined above with $\pi(t) = 1$ identically.*

*Then the asymptotic excess risk of ridgeless regression is as follows*

*(here $\delta := \delta_0 \gamma = \lim_{n,p \to \infty} (n/p)$):*

1. *For $\delta > 1$:*

$$\lim_{\lambda \to 0} \operatorname*{p\text{-}lim}_{N,p \to \infty} R_{\text{exc}}(\hat{\boldsymbol{\theta}}_\lambda) = \left( \frac{B_1}{A_1} - \frac{B_\pi}{A_\pi} \right)^2 + \frac{1}{\delta - 1} \cdot \left( C_\pi - \frac{B_\pi^2}{A_\pi} \right). \tag{M.3}$$

2. *For $\delta < 1$:*

$$\lim_{\lambda \to 0} \operatorname*{p\text{-}lim}_{N,p \to \infty} R_{\text{exc}}(\hat{\boldsymbol{\theta}}_\lambda) = \left( \frac{B_1}{A_1} - \frac{B_\pi \delta}{1 - \delta + A_\pi \delta} \right)^2 + \frac{B_\pi^2}{A_\pi} \cdot \frac{\delta(1 - \delta)}{(1 - \delta + A_\pi \delta)^2} \tag{M.4}$$
$$+ \frac{\delta}{1 - \delta} \cdot \left( C_\pi - \frac{B_\pi^2}{A_\pi} \right).$$

**Remark M.1** (Random data selection). In the case of no data selection, $\pi(\boldsymbol{x}) = \gamma$, we have $A_\pi = A_1 = 1$, $B_\pi = B_1$, $C_\pi = C_1$, and we recover the result of ordinary ridgeless regression (Advani et al., 2020; Hastie et al., 2022)

$$\delta > 1: \quad \lim_{\lambda \to 0} \operatorname*{p\text{-}lim}_{N,p \to \infty} R_{\text{exc}}(\hat{\boldsymbol{\theta}}_\lambda) = \frac{1}{\delta - 1} \cdot \left( C_1 - \frac{B_1^2}{A_1} \right), \tag{M.5}$$

$$\delta < 1: \quad \lim_{\lambda \to 0} \operatorname*{p\text{-}lim}_{N,p \to \infty} R_{\text{exc}}(\hat{\boldsymbol{\theta}}_\lambda) = B_1^2(1 - \delta) + \frac{\delta}{\delta - 1} \cdot \left( C_1 - \frac{B_1^2}{A_1} \right). \tag{M.6}$$

## M.1 Proof of Proposition M.2

Since we are assuming a perfect surrogate, $\alpha_s = 0$, by applying Theorem 4, we get

$$\underset{N,p\to\infty}{\text{p-lim}} R(\hat{\boldsymbol{\theta}}_\lambda) = \mathbb{E}\big\{(Y - \alpha_0^* G_0 - \alpha_\perp^* G_\perp)^2\big\}$$

$$= C_1 - 2B_1\alpha_0^* + A_1(\alpha_0^*)^2 + (\alpha_\perp^*)^2, \tag{M.7}$$

whence the excess error is

$$\underset{N,p\to\infty}{\text{p-lim}} \min_{\boldsymbol{\theta}} R_{\text{exc}}(\hat{\boldsymbol{\theta}}_\lambda) = \left(\frac{B_1}{A_1} - \alpha_0^*\right)^2 + (\alpha_\perp^*)^2. \tag{M.8}$$

Simplifying the Lagrangian (M.1) in the case of perfect surrogate, we get (recalling that $\delta = \delta_0\gamma$)

$$\frac{1}{\gamma}\mathscr{L}_{\text{ls}}(\alpha_0, \alpha_\perp) = \frac{1}{2}\left(\sqrt{C_\pi - 2B_\pi\alpha_0 + A_\pi\alpha_0^2 + \alpha_\perp^2} - \frac{\alpha_\perp}{\sqrt{\delta}}\right)_+^2 + \frac{\lambda}{2\gamma}\|\boldsymbol{\alpha}\|_2^2$$

$$=: \frac{1}{2}G(\boldsymbol{\alpha})^2 + \frac{\lambda}{2\gamma}\|\boldsymbol{\alpha}\|_2^2.$$

In the limit $\lambda \to 0$, $\boldsymbol{\alpha}^* = (\alpha_0^*, \alpha_\perp^*)$ is given by

$$\boldsymbol{\alpha}^* = \arg\min\Big\{\|\boldsymbol{\alpha}\|^2 : \boldsymbol{\alpha} \in \arg\min_{\boldsymbol{a}\in\mathbb{R}\times\mathbb{R}_{\geq 0}} G(\boldsymbol{a})^2\Big\}. \tag{M.9}$$

Depending on the value of $\delta$, the solution of this problem is achieved in different domains of the plane:

- For $\delta > 1$, $\arg\min_{\boldsymbol{a}\in\mathbb{R}\times\mathbb{R}_{\geq 0}} G(\boldsymbol{a})^2$ is uniquely achieved when $G(\boldsymbol{\alpha}) > 0$, and hence satisfies $\nabla G(\boldsymbol{\alpha}) = 0$, $G(\boldsymbol{\alpha}) > 0$. Simple calculus yields

$$\alpha_0^* = \frac{B_\pi}{A_\pi}, \tag{M.10}$$

$$\alpha_\perp^* = \sqrt{\frac{1}{\delta - 1}\left(C_\pi - \frac{B_\pi^2}{A_\pi}\right)}. \tag{M.11}$$

- For $\delta < 1$, $\arg\min_{\boldsymbol{a}\in\mathbb{R}\times\mathbb{R}_{\geq 0}} G(\boldsymbol{a})^2$ is $S_0 := \{\boldsymbol{\alpha} : G(\boldsymbol{\alpha}) = 0\}$, and it is easy to see that (for $r_0 := C_\pi - B_\pi^2/A_\pi$):

$$S_0 = \left\{(\alpha_0, \alpha_\perp)\mathbb{R} \times \mathbb{R}_{\geq 0} : Q(\boldsymbol{\alpha}) := \left(\frac{1}{\delta} - 1\right)\alpha_\perp^2 - A_\pi\left(\alpha_0 - \frac{B_\pi}{A_\pi}\right)^2 - r_0 \geq 0\right\}. \tag{M.12}$$

Hence we $\boldsymbol{\alpha}_*$ solves (for a Lagrange multiplier $\xi$)

$$\nabla Q(\boldsymbol{\alpha}) = \xi\boldsymbol{\alpha}, \quad Q(\boldsymbol{\alpha}) = 0, \tag{M.13}$$

which are easily solved.

The proof is completed by substituting this solution in Eq. (M.8).

## N Synthetic data numerical experiments

It is instructive to compare the scheme described in Eq. (6.1) with influence function-based data selection, cf. Example 4.2. Within the present data model, the population Hessian takes the form $\boldsymbol{H} = a_+\boldsymbol{I} + b_+\boldsymbol{\theta}_0\boldsymbol{\theta}_0^\mathsf{T}/\|\boldsymbol{\theta}_0\|^2$ where $a_+ = \mathbb{E}\{\phi''(\|\boldsymbol{\theta}_0\|G)\}$, $b_+ = \mathbb{E}\{\phi''(\|\boldsymbol{\theta}_0\|G)(G^2 - 1)\}$. The score of Example 4.2 (cf. Eq. (4.7)) reads (an overall factor $da_-$ is immaterial and introduced for convenience)

$$Z(\boldsymbol{x}_i) = \phi''(\langle\boldsymbol{\theta}_0, \boldsymbol{x}_i\rangle)\left\{\frac{\|\boldsymbol{x}_i\|^2}{d} + b_-\frac{\langle\boldsymbol{\theta}_0, \boldsymbol{x}_i\rangle^2}{da_-\|\boldsymbol{\theta}_0\|^2}\right\}, \tag{N.1}$$

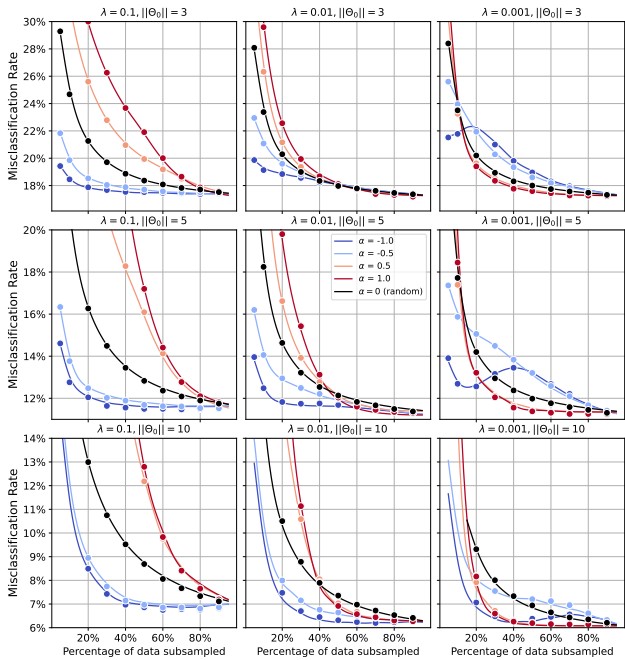

Figure 5: Misclassification error for logistic regression after non-reweighted subsampling under the scheme of Eq. (6.1). Here $N = 34345$, $p = 932$. Circles are results of numerical simulations, and continuous lines are theoretical predictions. Each panel corresponds to a different choice of $\|\boldsymbol{\theta}_0\|$, $\lambda$.

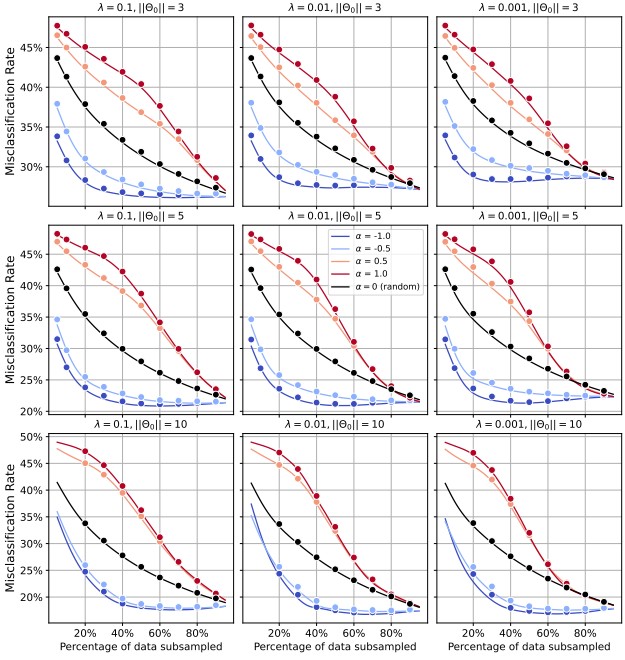

Figure 6: Same as Figure 5, with $N = 6870$ and $p = 3000$.

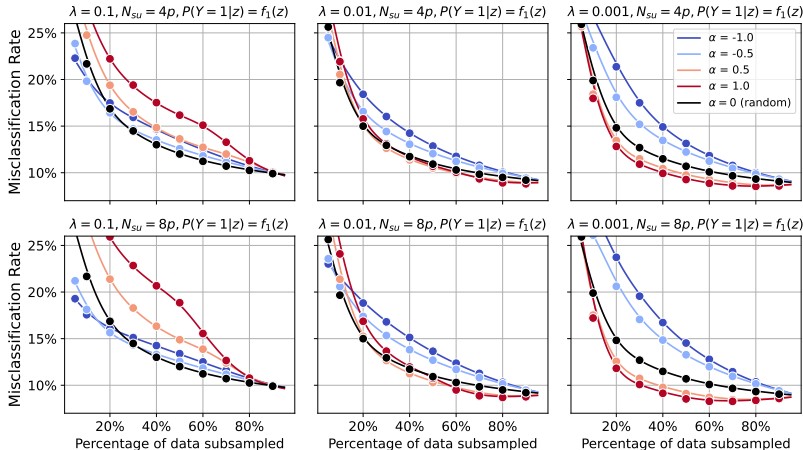

Figure 7: Misclassification error for logistic regression after non-reweighted subsampling with data generated according to the same misspecified model as in Figure 2. Circles: simulations. Continuous lines: theory. Here $N = 34345$, $p = 932$. Unlike in Figure 2, we use an imperfect surrogate $\hat{\boldsymbol{\theta}}^{\text{su}}$ that is fit on $N_{\text{su}}$ samples from the same distribution. Top row: $N_{\text{su}} = 4p$. Bottom row: $N_{\text{su}} = 8p$. The values of $\lambda$ indicated in the plot titles are used when learning on the selected subsample.

where $a_- = 1/a_+$, $b_- = (a_+ + b_+)^{-1} - 1/a_+$. For large dimension $d$, $\|\boldsymbol{x}_i\|^2/d = 1 + O_P(1/\sqrt{d})$, and $\langle \boldsymbol{\theta}_0, \boldsymbol{x}_i \rangle = O_P(1)$. We therefore get

$$Z(\boldsymbol{x}_i) = \phi''(\langle \boldsymbol{\theta}_0, \boldsymbol{x}_i \rangle) \cdot \left(1 + O_P(1/\sqrt{d})\right). \tag{N.2}$$

Therefore influence-function based subsampling essentially corresponds to the case $\alpha = 1/2$ of Eq. (6.1).

Figures 5 and 6 report results of simulations (circles) and theoretical predictions (continuous lines), respectively in a lower-dimensional setting ($N = 34345$, $p = 932$) and in a higher-dimensional setting ($N = 6870$, $p = 3000$). Simulation results are medians over 10 realizations. Theoretical predictions are computed by evaluating the saddle point formula of Theorem 4. Finally, while all previous figures use a perfect surrogate, Figure 7 explores the impact of an imperfect surrogate. Namely, we estimate $\hat{\boldsymbol{\theta}}^{\text{su}}$ by using $N_{\text{su}}$ independent samples from the same distribution as the training samples. We train $\hat{\boldsymbol{\theta}}^{\text{su}}$ using ridge-regularized logistic regression, with an oracle choice of the regularization parameter $\lambda$. This setting gives an intuitive understanding of the 'cost' of the surrogate model. We choose $N_{\text{su}} = 4p$ (top row) or $N_{\text{su}} = 8p$ (bottom row), corresponding to $N_{\text{su}}/N \approx 10.9\%$ or $N_{\text{su}}/N \approx 21.7\%$, respectively.

The agreement between theoretical predictions and simulation results is again excellent. Also, we observe behaviors that are qualitatively new with respect to the previous setting that assumes well-specified data and a perfect surrogate. Most notably:

1. Learning after data selection often outperforms learning on the full sample.

2. Upsampling 'hard' datapoints (i.e. using $\alpha > 0$) is often the optimal strategy. This appears to be more common than in the well-specified case.

3. As shown in Figure 7, the performance of data selection-based learning degrades gracefully with the quality of the surrogate.

4. In particular, we observe once more the striking phenomenon of Figure 1, cf. bottom row, rightmost plot of Figure 7. At subsampling fraction $n/N = 60\%$, learning on selected data outperforms learning on the full data, even if the surrogate model only used additional $N_{\text{su}}/N \approx 21.7\%$ samples. As shown in next section, this effect is even stronger with real data.

## O   REAL DATA NUMERICAL EXPERIMENTS

### O.1   DATASET

We use a subset of images obtained from the KITTI-360 train set (Liao et al., 2022). The KITTI-360 train set comprises $1408 \times 376$ dimensional 8-bit stereo images, with 2D semantic and instance labels. These images are sourced from 9 distinct continuous driving trajectories. We only consider the left stereo image for our dataset. To adapt this dataset for a binary classification task, we initially center-crop the images to dimensions of $224 \times 224$ pixels. Subsequently, we assign binary labels, by setting $y_i = +1$ if the count of pixels containing the semantic label in a certain class surpasses a predefined threshold. We choose 'car' as the label and the pixel cutoff threshold is set at $50$, resulting in a chance accuracy of approximately $0.69$.

We then extract SwAV embeddings of the images to serve as feature vectors (Caron et al., 2021). We use `torch.hub.load('facebookresearch/swav:main', 'resnet50')` as the base model and we use the 2048 dimensional outputs from the penultimate layer (head) as the SwAV features. Following common practice, we normalize the images before computing feature vectors, using mean and standard deviations of the images calculated on ImageNet. This results in a dataset with a total of $N = 61,280$ images with $p = 2048$ dimensional features with binary labels indicating the presence or absence of a car.

### O.2   EXPERIMENT SETUP

We randomly partition this dataset into four disjoint sets: $N_{\text{train}} = 34,345$ images to perform subsampling and train models, $N_{\text{surr}} = 14,720$ images to train surrogate models, $N_{\text{val}} = 3665$ images for validation and $N_{\text{test}} = 8550$ images for reporting the final experiment results. Prior to model training, we center and normalize each of the features using mean and standard deviation calculated from $N_{\text{train}}$ and $N_{\text{surr}}$.

We proceeded by training a ridge-regularized logistic regression model without intercept. The training utilized the L-BFGS optimization algorithm with a cap of 10,000 iterations implemented using the scikit-learn library (Pedregosa et al., 2011). Surrogate models are trained on a fraction $k$ of the surrogate set ($N_{\text{surr}}$) where $k \in \{10\%, 50\%, 100\%\}$ using 5-fold cross validation to choose the regularization parameter $\lambda$. Note that, the sample sizes used for these surrogate models correspond to $\{4.2\%, 21.4\%, 42.8\%\}$ of $N_{\text{train}}$ (but we use a disjoint set of data).

We use the data selection procedure introduced in the previous section, cf. Eqs. (6.1), (6.1). We show empirical results for $\alpha \in \{-2, -1, -0.5, 0, 0.5, 1, 2\}$, where $\alpha = 0$ corresponds to random subsampling and positive (negative) values of $\alpha$ correspond to upsampling hard (easy) examples respectively. To ensure numerical stability for negative values of $\alpha$ modify the definition of $\pi$ in Eqs. (6.1), by limiting[7] $\langle \hat{\boldsymbol{\theta}}^{\text{su}}, \boldsymbol{x}_i \rangle$ to be in $[-10, 10]$.

We also perform experiments in which we select the $n$ samples with the largest (or smallest) values of $p(\langle \hat{\boldsymbol{\theta}}^{\text{su}}, \boldsymbol{x}_i \rangle)(1 - p(\langle \hat{\boldsymbol{\theta}}^{\text{su}}, \boldsymbol{x}_i \rangle))$. This corresponds to the limit cases $\alpha \to \infty$ and $\alpha \to -\infty$ respectively. We will refer to these limit cases as 'Hard topK' and 'Easy topK', respectively.

### O.3   ADDITIONAL RESULTS

All experiments are repeated across five random subsamplings when the data selection scheme is probabilistic and across three different surrogate models when the surrogate model is trained an a strict subset of the $N_{\text{surr}}$ samples reserved for this.

We vary four different parameters in our experiments:

- The ridge regularization parameter $\lambda$. This is either fixed or selected optimally by taking $\lambda^* = \arg\min_{\lambda \in \Lambda} R_{\text{val}}(\hat{\boldsymbol{\theta}}_\lambda)$, where $R_{\text{val}}$ is the risk on the validation set and $\Lambda := \{0.001, 0.01, 0.03, 0.06, 0.1, 1, 10\}$.

---

[7] Formally, we replace $\langle \hat{\boldsymbol{\theta}}^{\text{su}}, \boldsymbol{x}_i \rangle$ by $T(\langle \hat{\boldsymbol{\theta}}^{\text{su}}, \boldsymbol{x}_i \rangle)$, where $T(x) = \min(\max(x, -10), 10)$.

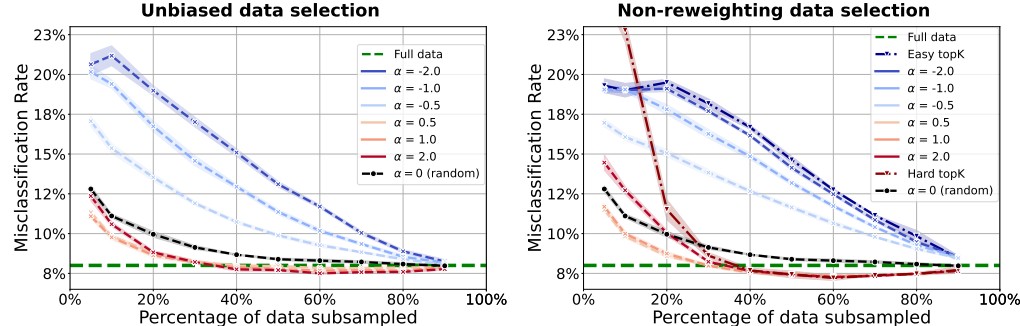

Figure 8: Test error on image classification task, for a model trained after data subsampling. Effect of changing $\alpha$ in the subsampling probabilities, cf. Eq. (6.1). Here we use both unbiased (left) and non-reweighting (right) subsampling schemes with $n_{\text{surr}}/N_{\text{train}} = 4.2\%$.

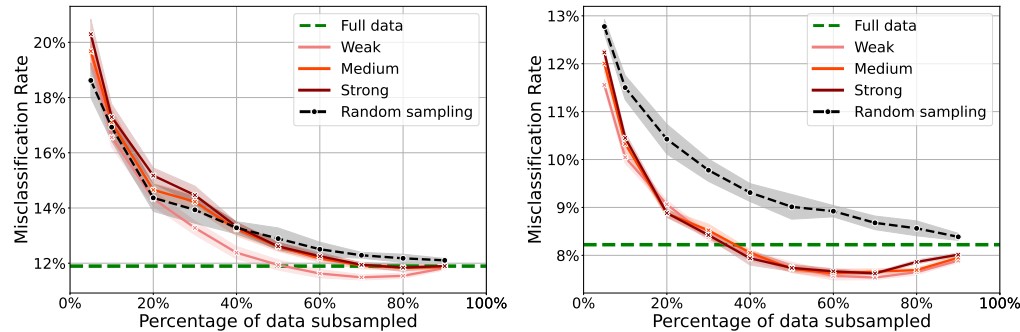

Figure 9: Test error on image classification task. Here $\lambda = 0.001$, $\alpha = 0.5$, and non-reweighting subsampling. Left plot: $N = 3434$, $p = 2048$; right plot: $N = 34345$, $p = 2048$. Various curves correspond to different surrogate models.

- The exponent $\alpha$ that parameterizes the subsampling probabilities.
- The 'strength' of the surrogate model, surr, namely, the sample size $n_{\text{surr}} \in [0, N_{\text{surr}}]$ used to learn $\hat{\boldsymbol{\theta}}^{\text{su}}$. We will report the ratio $n_{\text{surr}}/N_{\text{train}}$, as this provides a direct measurement of how much information is required to train the surrogate model. In particular, we will qualitatively refer to surrogate models in results and figures as 'weak', 'medium' and 'strong' for the cases where: $n_{\text{surr}}/N_{\text{train}} = 4.2\%$, $n_{\text{surr}}/N_{\text{train}} = 21.4\%$ and $n_{\text{surr}}/N_{\text{train}} = 42.8\%$ respectively.
- A binary parameter bias indicating whether we are using unbiased or non-reweighted subsampling (referred to as biased sampling in Fig. 1).

Figure 8 reports the misclassification rate on the test set as a function of the subsampling fraction $\gamma$ for various values of the exponent $\alpha$. We consider both unbiased and non-reweighting subsampling and use 'weak' surrogate models. In this case, we select $\lambda$ optimally, as described above. We observe that subsampling with $\alpha > 0$ outperforms training on the full sample down to subsampling ratios $\gamma \gtrsim 0.4$. This is most significant with non-reweighing subsampling, as anticipated by the asymptotic theory of Section 4.1. Further, above this value of $\gamma$, the test error is fairly insensitive to the choice of $\alpha > 0$. The situation changes dramatically at smaller subsampling fractions. In particular, for non-reweighting subsampling and for $\gamma < 0.3$, soft subsampling $\alpha = 0.5$ outperforms substantially $\alpha = 2$ and $\alpha = \infty$. Negative $\alpha$ (upsampling easy examples) always underperforms with respect to random in this case.

Figure 9 investigates the effect of the strength of the surrogate model. In both subplots, we fix $\lambda = 0.001$, $\alpha = 0.5$, and subsampling with no-reweighting. The two subplots correspond to different regimes of the number of samples-to-parameters ratio. The left subplot uses 10% of the total

available training set (i.e. $N = 0.1 \times N_{\text{train}} = 3434$) as $100\%$ train data, while the right subplot uses the entire training set (i.e. $N = N_{\text{train}} = 34,345$). Each subplot shows subselection performance for three different surrogate models – 'weak', 'medium' and 'strong' described above.

We observe that at larger sample size $N$ (right subplot), the test error of the model learnt after subsampling is insensitive to the accuracy of the surrogate model. We recover the results of Figure 8 irrespective of the strength of the surrogate. This is encouraging because it indicates that weak supervision is sufficient for effective data selection. Even more surprising is the behavior at smaller sample size (left plot). In this case the weak surrogate outperforms medium and strong surrogates. A similar phenomenon was derived in a minimax setting in Section 4.5.

## P    SOME USEFUL FORMULAS FOR BINARY CLASSIFICATION

In line with the model introduced in the main text, we consider $y_i \in \{+1, -1\}$ and

$$\mathbb{P}(y_i = +1 | \boldsymbol{x}_i) = f(\langle \boldsymbol{\theta}_0, \boldsymbol{x}_i \rangle). \tag{P.1}$$

(In other words, $\mathsf{P}(+1|z) = 1 - \mathsf{P}(-1|z) = f(z)$. We use logistic loss

$$L(y; z) = -yz + \log(e^{-z} + e^z). \tag{P.2}$$

Formulas below are obtained by specializing the results of Section 5.

**Unbiased subsampling.**    The Lagrangian takes the form

$$\mathscr{L}(\boldsymbol{\alpha}, \mu) := \frac{\lambda}{2} \|\boldsymbol{\alpha}\|^2 - \frac{1}{2\delta_0} \mu \alpha_\perp^2 \tag{P.3}$$

$$+ \mathbb{E} \Big\{ f(\|\boldsymbol{\theta}_0\| G_0) \min_{u \in \mathbb{R}} \big[ L(\alpha_0 G_0 + \alpha_s G_s + u; +1) + \frac{1}{2} \mu \, \pi(\langle \boldsymbol{\beta}, \boldsymbol{g} \rangle) \big( u - \alpha_\perp G_\perp \big)^2 \big]$$

$$+ (1 - f(\|\boldsymbol{\theta}_0\| G_0)) \min_{u \in \mathbb{R}} \big[ L(\alpha_0 G_0 + \alpha_s G_s + u; -1) + \frac{1}{2} \mu \, \pi(\langle \boldsymbol{\beta}, \boldsymbol{g} \rangle) \big( u - \alpha_\perp G_\perp \big)^2 \big] \Big\}.$$

**Non-reweighted data selection.**    In this case Lagrangian reduces to

$$\mathscr{L}(\boldsymbol{\alpha}, \mu) := \frac{\lambda}{2} \|\boldsymbol{\alpha}\|^2 - \frac{1}{2\delta_0} \mu \alpha_\perp^2 \tag{P.4}$$

$$+ \mathbb{E} \Big\{ f(\|\boldsymbol{\theta}_0\| G_0) \pi(\langle \boldsymbol{\beta}, \boldsymbol{g} \rangle) \min_{u \in \mathbb{R}} \big[ L(\alpha_0 G_0 + \alpha_s G_s + u; +1) + \frac{1}{2} \mu \big( u - \alpha_\perp G_\perp \big)^2 \big]$$

$$+ (1 - f(\|\boldsymbol{\theta}_0\| G_0)) \pi(\langle \boldsymbol{\beta}, \boldsymbol{g} \rangle) \min_{u \in \mathbb{R}} \big[ L(\alpha_0 G_0 + \alpha_s G_s + u; -1) + \frac{1}{2} \mu \big( u - \alpha_\perp G_\perp \big)^2 \big] \Big\}.$$

**Misclassification error.**    For $L_{\text{test}}(y; z) = \mathbf{1}(yz < 0)$:

$$R(\lambda; \boldsymbol{\theta}) = \frac{1}{2} - \frac{1}{2} \mathbb{E} \Big\{ \big( 2f(\|\boldsymbol{\theta}_0\|_2 G) - 1 \big) \big( 2\Phi(q\,G) - 1 \big) \Big\}, \quad q := \frac{\alpha_0}{\sqrt{\alpha_s^2 + \alpha_\perp^2}}. \tag{P.5}$$

