# TOWARDS A STATISTICAL THEORY OF DATA SELECTION UNDER WEAK SUPERVISION: RESPONSE TO REVIEWERS

## 1   RESPONSE TO REVIEWER VO45

In regards to the comment made by Reviewer Vo45, we reproduced the base algorithm from Munteanu et al. (2018). Below we reproduce the Figure 1 (left) subplot with added results from the paper. We modified their method minimally, to allow for varying subsampling rates.

We emphasize that these are **preliminary results** and we need to carry out a more careful study before including this comparison in the paper. In our simulations, the method of Munteanu et al. (2018) does not behave better than random subsampling.

Notice that the simulations of that paper are all carried out in a very low-dimensional (underparametrized) regime in which the sample size $n$ is much larger than the number of parameters $p$: in all of their examples $n > 2000 \times p$. In contrast, we work in a higher dimensional setting, in which the model is either moderately underparametrized, or is overparametrized. We also point out that, the coreset approach presents a fundamental limitation. Indeed, it aims at approximating as well as possible the full sample empirical risk minimization problem. As a consequence, it will never outperform the full sample ERM, while our approach does.

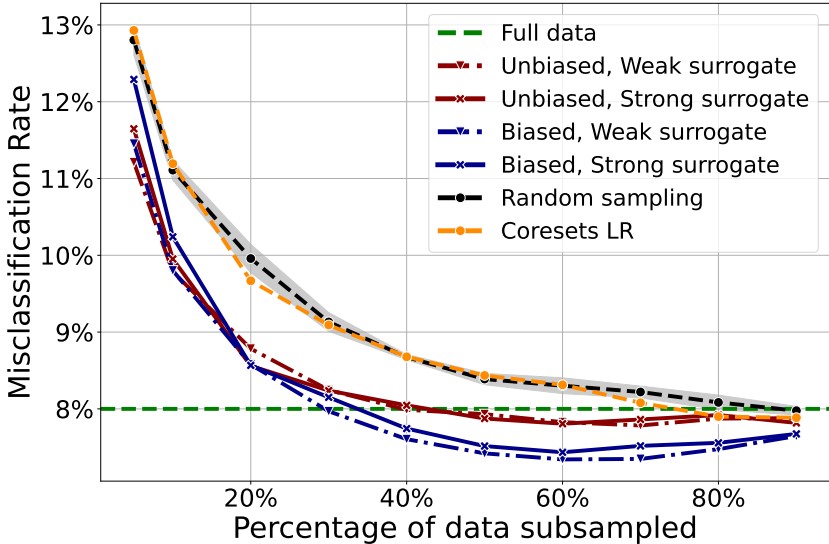

Figure 1: Misclassification error in an image classification problem after data selection (logistic regression on SwAV embeddings). $N = 34345$ samples, $p = 2048$ dimensions. We use surrogate models trained on a small separated fraction of the data ('strong': $N_{\text{su}} = 14720$, 'weak': $N_{\text{su}} = 1472$).

## REFERENCES

Alexander Munteanu, Chris Schwiegelshohn, Christian Sohler, and David Woodruff. On coresets for logistic regression. *Advances in Neural Information Processing Systems*, 31, 2018.