# OpenReview forum: "Towards a statistical theory of data selection under weak supervision"
_ICLR.cc/2024/Conference — ICLR 2024 oral_

### Official Review · Reviewer_NQ88 · 2023-10-27

**Soundness:** 4 excellent
**Presentation:** 4 excellent
**Contribution:** 3 good
**Rating:** 8
**Confidence:** 4

**Summary:**

This paper investigates subsampling in supervised learning. It begins by assuming a collection of labeled data points, denoted as $\left(x_i, y_i\right)_{i \leq N}$, which are drawn as independent and identically distributed (i.i.d.) samples from a given distribution, denoted as $P$. Additionally, they introduce a surrogate model, denoted as $\hat{P}(y \vert x)$, which is capable of predicting labels more effectively than random guessing. The central objective, in the absence of access to the true labels of the data points, is to leverage the surrogate model to perform subsampling on the training data, resulting in a reduced dataset of size $n = \gamma N$, where $\gamma$ is a parameter in the range of $(0,1)$. This reduced dataset is then used to train the final model.

The experiments conducted by the authors have shown some intriguing outcomes. Firstly, the choice of subsampling strategy seems to have a significant impact, offering more than mere randomness in the process of reducing the samples. Secondly, in specific cases, subsampling can actually yield lower estimation errors during testing. It's worth noting that this study is supported by a solid framework of theoretical guarantees.

While the authors have made notable strides in their experimental validations, I believe some theoretical aspects remain unaddressed through analytical tools. Several propositions and theorems presented inside the paper are scattered and might be considered as ancillary results. Nevertheless, this paper is still a solid theoretical work with robust mathematical underpinnings. The theoretical formulation of the problem and the experimental observations collectively present a nice contribution to the field, which makes this work a nice addition to ICLR. My vote is in favor of acceptance, and I am open to revising my evaluation should the authors give convincing responses to the questions raised in the Questions and Weaknesses sections.

**Strengths:**

- The problem setup is straightforward and easily comprehensible, yet it leads to intriguing and intricate implications from both theoretical and experimental standpoints.

- Notably, the experimental findings presented in Figure 1, particularly when subsampling effectively reduces estimation errors compared to utilizing all data points, are highly intriguing.

- The authors have explored a wide range of subsampling schemes, including both biased and unbiased methods. Additionally, the asymptotic analysis in this work covers scenarios in both low-dimensional and high-dimensional regimes.

- The paper provides a substantial foundation of solid mathematical guarantees. I did not find any notable mathematical errors, and the theoretical results exhibit a commendable level of mathematical rigor. However, these results don't always align seamlessly with the compelling experimental findings, appearing as somewhat scattered attempts to tackle a very challenging problem.

- The authors have asserted that they've uncovered intriguing connections between an almost universally unbiased subsampling scheme and a method based on "influence functions" from prior research, as mentioned in Remark 4.1.

- The paper is well-written and is easy to read.

**Weaknesses:**

- The primary limitation of this paper, to the best of my understanding, is that all mathematical analyses beyond Section 4 assume that the surrogate model is equivalent to the optimal Bayes conditional distribution. In other words, the sample selection process somewhat presumes knowledge of the true label distributions. Consequently, the authors have not been able to provide a sound theoretical justification for the "magic" effects claimed in Figure 1. This drawback significantly affects the significance of the work, in my opinion.

- Theorem 1 asserts that $\rho_{\mathrm{unb}}/\rho_{\mathrm{nr}}$ can grow arbitrarily large by selecting the feature vectors' distribution in an adversarial manner. However, can the same be said for $\rho_{\mathrm{nr}}/\rho_{\mathrm{unb}}$? What happens when the feature vectors' distribution is more generic, such as Gaussian? Without additional guarantees in these respects, the theorem may lack substantial significance.

- All the mathematical analyses in this work are based on asymptotic conditions ($n,N\rightarrow\infty$ with $n/N\rightarrow\gamma$), which remain intriguing but could be expanded to encompass a broader scope. For instance, non-asymptotic guarantees and cases where $n/N\rightarrow 0$ could be explored.

- The section related to high-dimensional analysis exclusively considers a linear model with Gaussian feature vectors. Additionally, it assumes that $N/p$ converges to a known constant. These assumptions offer room for extension and relaxation.

- This paper is densely packed with intricate mathematical statements, often presented in a dense manner. Many results, sometimes unrelated, may require more context and explanation than a "9-page limit" can accommodate. In this regard, the authors might consider submitting their work to a journal to allow for a more comprehensive presentation.

- Paper has no conclusions section.

**Minor comments**:
- In Theorem 2: "an non-empty" -> "a non-empty".

**Questions:**

- **Why $\Omega:\mathbb{R}^p\rightarrow \mathbb{R}$**: In other words, why the dimensionality of parameter $\theta$ has been assumed to be the same as that of the input vector $\boldsymbol{x}$. Obviously, this assumption makes sense when using a linear model. However, does it alter the generality of the presented framework in any shape or form?

- **In Figure 1, how do you justify the reduction in estimation error after subsampling?** The way I have understood the experiment: "Full data" curve uses all the 34345 data samples with **true** labels. On the other hand, the proposed scheme (in the case of a weak surrogate model)
- - 1) First, uses only 1472 samples (with true labels) from another fraction of the dataset in order to train a surrogate model.
- - 2) You use the above-mentioned surrogate model to select, say, 50% of the dataset (without seeing their true labels, right?), and then
- - 3) The training procedure considers the true labels, only for the above-mentioned selected samples, and uses them for training the main model.

At the end, the main model outperforms the "Full data" curve. Is it because the surrogate model has been trained too good? can you please also report the estimation error of the weak and strong surrogate models? Also, it should be noted that Theorem 2 only shows that there "exists" cases for which $\rho_{\mathrm{nr}}$ is not monotonic. But it does not prove error can become lower than that of a full data ERM.

- **Assumption A.1 (in Proposition B.1)**: I am a little confused here... In Proposition 4.1, do we have to assume that A.1 (of Proposition B.1) holds? Or the subsampling strategy that is guaranteed to minimize $\rho_{\mathrm{unb}}$ automatically satisfies this condition? (i.e., forcing the minimizers of $R_S$ and $R$ to coincide with each other)

- **What (or should I say: Where) is $b(x)$ in Proposition 4.3**?

---

> ### Author Response · Authors · 2023-11-17
>
> We thank the reviewer for their thorough and insightful feedback and questions. Below we summarize the responses to weaknesses and questions.
>
> **Imperfect surrogate.** It is not true that our analysis beyond Section 4 is limited to perfect surrogates.
> - The analysis summarized in Section 5 (and Appendix L) covers the general case in which the surrogate model is misaligned with the true model. The strength of the surrogate model is represented by the parameter $\beta_s$ in Eq 5.2, where a perfect surrogate is a special case with $\beta_0 =  || \theta_0 || $.
> - We use this theory to study the effect of surrogate model misspecification in Figure 2 and estimation error in the surrogate model in Figure 7 (Appendix L).
> - Notice in particular that the panels in Figure 7 capture the qualitative behavior of Figure 1 (the “magic” effect) with comparable choices for sample size and dimension. We conclude that this theory provides a good starting point for understanding these effects.
> - We also point out that the analysis of imperfect surrogates summarized in Section 5 occupies the whole appendix K.
>
> **Theorem 1.** *This theorem is not about adversarial examples*, but random i.i.d. ones (“there exists a distribution”). Further, by inspection of the proof (Example G.2), the distribution is very simple and natural (as ‘natural’ as the Gaussian distribution). We simply take feature vectors $x_i$ that are generated as $x_i=r \cdot u_i$, where $u_i$ is uniform on the sphere, and $r$ is a scalar that is independent of $u_i$ and has a power law tail. Gaussian with identity covariance does not work for this example because (informally) 'all Gaussian vectors look the same'. In a sense, this is what makes Gaussian data an unrealistic model in many applications: they have way less variability than real data.
>
> **Dimensionality of the parameter.** The theory of Section 4 accommodates for a number of parameters (indicated by $p$) different from the number of input dimensions ($d$). The theory of Section 5 assumes a generalized linear model, and therefore $p=d$.
>
> **Explaining Figure 1.** The experiment settings of the Figure 1 that you described are correct. The test accuracy of the weak (strong) surrogate model is 87% (91%). Weaker models can perform better than stronger ones, depending on $N/p$, see Figure 9 Appendix O.
> We agree that these results are surprising. Indeed, we were motivated to work out the theory in order to demystify these results. In summary:
>
> - The high-dimensional theory of Section 5 and Appendix L captures this phenomenon. See in particular Figures 2 and 7, where (in some of the settings) the same phenomenon is observed both under misspecification and imperfect surrogate estimation.
> - Theorem 2 also proves the same phenomenon in a low-dimensional setting. (In this case the surrogate is assumed to be perfect, but it is elementary to see that using parameters estimated by a number of samples much smaller than $d$ yields the same result).
> - The reason for this is that the intuition “every sample is useful” is only correct if the data distribution is perfectly matched to the loss function (e.g. logistic and logistic). As soon as this does not hold (which is the case in most real data applications), not all datapoints improve the estimation error.
>
> **Theorem 2.** The theorem states that the error is strictly increasing for gamma in the interval $(\gamma_0,1)$. In particular, this means that the error at $\gamma=\gamma_0$ is strictly smaller than the error at $\gamma=1$. Recalling the definition of $\gamma$, the theorem proves that the (test) error achieved by training on subsampled data is strictly smaller than the error on the full sample ERM. We also note that:
>
> - For brevity, we state the theorem in the form “there exists.” However, the proof is constructive and provides data distributions for which data selection improves over full sample ERM.
> - As mentioned above, the same claims can be shown to hold if the surrogate model is trained on a vanishing fraction of the n samples for which labels are provided.
>
> We also fixed the typographical error pointed by the reviewer in the updated version. Thank you for catching it.
>
> **Proposition 4.1.** First of all, we apologize for the somewhat awkward formulation here. We were led to this by the attempt to defer to appendices material that is pedagogically useful, but is not novel.
>
> Second, if the assumption is not satisfied, then the subsampling scheme is (roughly speaking) ‘much worse’ than (say) random subsampling. (Technically, the procedure is not consistent.) In other words, we could write a longer form of this proposition without this assumption, and establish that schemes that do not satisfy it are ‘very bad.’
>
> Finally, under many models, satisfying these assumptions is fairly easy.
>
> **Proposition 4.3.** Thanks, this is a typographical error, please see the updated version.

---

> > ### Comment · Reviewer_NQ88 · 2023-11-22
> >
> > I would like to thank the authors for providing clarifications. At this point, I have no additional questions.
> > The authors assert that the analysis in this study encompasses imperfect surrogates. I recommend revisiting certain sections of the paper to underscore these findings.
> >
> > I stand by my decision to vote for acceptance

---

### Official Review · Reviewer_fDUv · 2023-10-29

**Soundness:** 3 good
**Presentation:** 2 fair
**Contribution:** 2 fair
**Rating:** 5
**Confidence:** 2

**Summary:**

The paper studies the following problem.
Suppose we are given $N$ unlabeled samples where each of them has an underlying label and a surrogate model that predicts labels better than random guesses.
We would like to select a small subset of these $N$ samples of size $n$ and use the surrogate model to obtain their corresponding labels.
Then, we train a model based on these $n$ selected samples and their corresponding labels.
The question is: How do we select this subset?
The authors showed that if this subset is selected "correctly" then training on it can beat training on the full dataset in some cases.
Also, the authors showed that some popular choices of data selection can be suboptimal.

**Strengths:**

- The problem seems well-motivated.

**Weaknesses:**

- The presentation is quite technical.
Readers who are not experts in this area may find this paper hard to follow.

**Questions:**

Note:

- Page 1 second paragraph "close to $n$": $n$ is not defined at this point. It is a bit weird to say close to $n$.

- Paragraph below (1.2): What is cst?

- In (1.3): Is $\ell_{\text{test}}$ a new loss function? Or should $\ell_{\text{test}}$ be $\ell$?

- Section 2: What is $dy$? Is it the label predicted by the surrogate model?

---

> ### Author Response · Authors · 2023-11-17
>
> **Close to $n$.** Here $n$ is the target  subsample size. However, since we study probabilistic data selection methods as well, we allow for the subsampled size to have small fluctuations around $n$. (We could of course eliminate these fluctuations, at the cost of making the whole paper clumsier.)
>
> **cst.**  Cst is the standard abbreviation for constant, which means we re-weight the samples inversely proportional to the sampling probabilities.
>
> **$\ell_{test}$.**  This is the loss used for testing. This needs not to be the same as for training (eg cross-entropy for training and classification for testing).
>
>  **$dy$.** Y is the label, $dy$ is an infinitesimal variation of y, $P(dy|x)$ is the probability distribution of the labels conditioned on feature $x$. This is standard calculus notation.

---

> > ### Comment · Reviewer_fDUv · 2023-11-22
> >
> > Thanks for the clarification.

---

### Official Review · Reviewer_NBpo · 2023-10-29

**Soundness:** 4 excellent
**Presentation:** 3 good
**Contribution:** 4 excellent
**Rating:** 8
**Confidence:** 2

**Summary:**

This work studies the problem of data selection: given a large unlabeled dataset of size $N$, we would like to select a subsample of smaller size $n$ to be used for statistical analysis; e.g., one could collect only $n$ instead of $N$ labeled examples and perform statistical estimation only with this subset of data.

The authors assume access to $N$ unlabeled examples $\\{x_i\\}$ and to a "surrogate model", which is a weak-learner for the actual labeling problem (labels data better than random guessing) and hence can be used to predict the label $y_i$ of $x_i$.  We then select a subset $G \subset \\{x_i\\}$ of size $n < N$ of the examples, we label the data of $G$ using the "surrogate model" and finally train a parameterized model with that data via regularized ERM. The question is how to select $G$ so that the trained model is actually "good".

The paper presents both theoretical results and practical evidence of interesting phenomena for data selection mechanisms (in both low- and high-dimensional settings). For instance, data selection can beat training on the full sample in some cases and unbiased data selection can be highly sub-optimal compared to biased mechanisms.

**Strengths:**

The paper provides various results that I find interesting:

(i) While a standard method for data selection is unbiased sub-sampling, Theorem 1 shows that the error coefficient of unbiased schemes can be arbitrarily larger than that of biased ones. Hence, in many cases, unbiased subsampling is sub-optimal (e.g., Figure 1).

(ii) Figure 1 and Theorem 2 provide a setting where ERM using a selected subset of the data can lead to a better model than ERM on the full dataset.

(iii) The surrogate model is an important component in data selection. The authors give an example where better surrogate models do not lead to better selection.

I think that the above results (among others appearing in the paper) paint an interesting picture for data selection and open nice research directions. While most of the results are based on toy examples motivating the underlying phenomena, I find this paper a good fit for ICLR and I vote for acceptance.

**Weaknesses:**

I think that some parts of the paper are hard to follow and could be more clearly written (for instance, Sections 4-5). I understand that due to space constraints, presentation could be more challenging.

I do not find some other significant weakness.

**Questions:**

(1) How does the provided results in semi-supervised learning (where one has a small labeled dataset and many unlabeled examples but uses both for training) compare with the setting of the paper?

(2) The improvement in the ERM generalization using a well-selected subsample holds even for imperfect surrogate models (from Figure 1). Is there a result that compares the weakness of the surrogate model with the improvement (for some specific examples)?

---

> ### Author Response · Authors · 2023-11-17
>
> We thank the reviewer for their thoughtful feedback and questions. Below we summarize the responses to weaknesses and questions.
>
> **Presentation.** We agree that some of the sections are somewhat terse, largely as a byproduct of fitting all the results within the space constraints. Any advice is welcome.
>
> **Semi-supervised learning.** Semi-supervised learning addresses the same problem as the one in our paper: learning under a limited ‘labeling budget.’ However a formal comparison is at the moment difficult because of the differences between the two settings. In semi-supervised learning, the subset of labeled examples is given (not chosen algorithmically) and the learner tries to make the best use of unlabeled examples.
>
> In our setting, we are allowed to choose which examples to label, but once this is done, learning is carried out uniquely on the labeled examples. The two models are both interesting, but motivated by different use cases.
>
> On the other hand, we believe it should be possible (and interesting) to define a unified setting that captures both scenarios. We expect that in this setting, a mixture of the two strategies will be required.
>
> **Imperfect surrogate models.** Empirically, Figure 9 in Appendix O carries out a comparison of surrogate models of different strength.
> Mathematically, formalizing what 'optimality' means in the context of imperfect surrogates is far from obvious. This requires formalizing the notion of uncertainty about the true model.
> We provides two types of rigorous results for imperfect surrogates:
> - *Low-dimensional.* This is mainly contained in Appendix K and summarized in Section 4.5. In particular, we prove that using a weaker model can yield better data selection (in minimax sense) than using the perfect surrogate (See Theorem 7 and Example K.2).
> - *High-dimensional.* This is mainly contained in Appendix L and summarized in Section 5. In this case we fix an imperfect surrogate and associated data selection procedure, and characterize the resulting generalization error of data learned on data selected by that procedure (See Theorems 4 and 8). We study the effect of surrogate model misspecification (Figure 2) and estimation error in the surrogate model (Figure 7).

---

> > ### Comment · Reviewer_NBpo · 2023-11-22
> > **Thanks for the response**
> >
> > I would like to thank the authors for the response. I suggest adding these two comments in the paper along with the other reviewers’ comments. Given that, I would like to keep my score the same.

---

### Official Review · Reviewer_Vo45 · 2023-11-01

**Soundness:** 3 good
**Presentation:** 3 good
**Contribution:** 4 excellent
**Rating:** 10
**Confidence:** 3

**Summary:**

The paper considers a problem of selecting a subset of the data that results in a best performance under the empirical risk minimization setting. Further, labels are considered to be unknown, with an available surrogate model that can be used for estimating the labels. The task is accomplished by finding selection probabilities and a corresponding reweighting scheme that depends on the input data without labels. There are multiple results, that showcase properties of different selection schemas, their applicability and general recommendations for designing subsampling procedures.

**Strengths:**

A thorough study is performed about the general properties, that are beneficial for all subsampling schemes. Importantly, model generalization was well studied, in addition to the task of just solving an optimization problem. Biased to unbiased sampling comparison was very insightful, as there are many works where only unbiased sampling is considered, which appeared to be suboptimal under the presented setting.

Additional note after the Reviewer-Authors discussion (review score raised):

The paper reveals properties of data selection schemes, unusual in the well-established fields of data selection, such as the field of coresets. This is in part accomplished by formulating a different goal (such as the equation 4.3 in section 4) of data selection compared to one of the common goals of replicating the ERM loss on the full labeled data. It is expected that the paper will have a broader impact on the field of data selection under a variety of practical settings.

**Weaknesses:**

My main concern is the applicability in general setting and the assumptions in the paper:
- There is a concern in that (to my understanding) only the behavior exactly at the the optimum was considered (or at least in a small neighbourhood of the optimum), for example, refering to the equation B.3 (definition of the error based only on optimal values of the parameters); and the assumption B.1.A1. (lack of multiple optimal values). In most non-trivial non-linear models an iterative optimization procedure must be considered, which results in parameters passing through a range of values in addition to the final minima (global or local). In this case, even having a low error at the optimal paramter values will not help the optimization procedure.
- There appears to be a dependency of some calculated values on the value of optimal parameters that are to be estimated (\theta^*) starting from the equation 4.2. It was unclear for me whether we can use estimates of such parameters and how correct would be the final results when the assumptions are violated or some values are replaced with estimates (in case if closed form solutions are unavailable).
- Since the paper is aimed at establishing a new branch of the data selection theory, would be nice to state applicability limits (to my understanding, only linear models were considered in the examples, including linear models with simple single non-linearity, such as generalized linear models)

**Questions:**

- To clarify the assumptions, how applicable is the method to models with multiple local minimas? (non-convex losses and models)
- Related to the previous question, how applicable is the method to various non-linear models? For example, any model that contains non-trivial non-linearity, such as a two-layer network?
- Is it possible to use gradient-based and similar approximate procedures that require low subsampling error over the whole parameters space to converge (difference between Loss under subsampled data and full data to be small not only at the optimal values of parameters)?
- There is an extensive theory of data selection in the case when labels are known (in that case there are methods that guarantee low multiplicative or additive error over whole parameters space, far from optimum, when comparing full and subsampled datasets); would it be possible to evaluate the proposed model against such prior art, for example, considering a "perfect oracle" that exactly predicts the labels in the proposed setting?

---

> ### Author Response · Authors · 2023-11-17
>
> We thank the reviewer for their thoughtful feedback and questions. Below we summarize the responses to weaknesses and questions.
>
> **Non-convexity and related questions.** We present two sets of results:
> - *Low-dimensional asymptotics*: These are not limited to convex methods, but they do assume that the global optimizer is used for estimation.
> - *High-dimensional asymptotics*: these assume a (generalized) linear model, and convex loss.
>
> The referee raises an interesting question, which we rephrase as:
>
> *"What is the impact of data selection on the global optimization landscape in non-convex settings?"*
>
> This is a very difficult question and indeed more basic questions are widely open in non-convex statistical settings. A few remarks:
> - If subsampling is unbiased then we expect that uniform convergence results for the landscape [e.g. Mei, Bai, Montanari Ann. Stat. 2018] can be applied to show that the whole landscape under subsampling is 'as good as' the original one.
> - For biased subsampling, the same uniform convergence results will imply that the landscape can be characterized in terms of a certain 'modified population landscape'. The latter can be 'worse' or 'better' than the original landscape, and which one will depend on a case-by-case analysis.
> - For models outside the uniform convergence regime (e.g. highly overparameterized models) the above arguments will not apply, but uniform convergence of the landscape is often too strong of a requirement, and convergence in a neighborhood of the optimization trajectory is sufficient.
>
> Overall, we think there is some promise for extending the present results to capture the effects of data selection on non-convex optimization.
>
> **Dependency on estimated parameters.** Indeed, some of our estimates depend on unobserved parameters (as pointed out in Remark C.1). However, empirical versions of these quantities can be plugged in. Under the stated assumptions, corrections due to this preplacement are negligible. We refrain from making this explicit at each passage uniquely to avoid further notational burden.
>
> **Known labels.** Our experience is that our weakly supervised approach compares well with data selection methods that make use of the actual labels. If the reviewer has in mind a specific algorithm of this type that would be meaningful to use as a benchmark, we would be interested in carrying out a comparison.

---

> > ### Comment · Reviewer_Vo45 · 2023-11-18
> >
> > Thanks for a detailed answer to my questions.
> >
> > Regarding the question about "Known labels" case (which also covers the case of uniform convergence over the whole optimization landscape) - I may suggest comparing to any of the known "coreset" methods for regression or classification problems (such methods usually provide uniform guarantees about the whole optimization landscape, and in that case are sometimes called "strong coresets", however they require labels to be known)
> >
> > Particularly, easiest to implement known methods are based on an unbiased importance sampling, which involves computing "sensitivity" of the input points, and state of the art formulas for computing sensitivity are known for various simple machine learning models.
> >
> > As an example of a non-trivial construction for a simple machine learning model, I may suggest "On Coresets for Logistic Regression", Munteanu, et. al. (or any alternative paper for models other than logistic regression, outlined in the "Related Work" section of that paper).  General review of such methods can be found in "Practical Coreset Constructions
> > for Machine Learning", Bachem, et. al.
> >
> > If the newly suggested data selection approach is expected to improve the existing bounds in the "known labels" case as well, this would result in an additional significant advancement of the field. So it would be very interesting so see such a comparison.  However, it would be also understandable if this is left for a future work.

---

> > > ### Author Response · Authors · 2023-11-22
> > >
> > > Thanks for the concrete suggestions. We agree that this is an interesting comparison.
> > >
> > > We carried out **preliminary simulations** in the setting of Figure 1 using the base algorithm of [Munteanu et al. 2018]. (We modified their method minimally, to allow for varying subsampling rates.)
> > >
> > > The results are reported in the Figure 1 in the newly submitted supplementary material file (titled: "Towards a Statistical Theory of Data Selection Under Weak Supervision: Response to Reviewers"): we emphasize that these are preliminary results and we need to carry out a more careful study before including this comparison in the paper.
> > >
> > > Below we reproduce the text from the response:
> > > In our simulations, the method of [Munteanu et al. 2018] does not behave better than random subsampling. Notice that the simulations of that paper are all carried out in a very low-dimensional (underparametrized) regime in which the sample size $n$ is much larger than the number of parameters $p$: in all of their examples $n>2000\times p$. In contrast, we work in a higher dimensional setting, in which the model is either moderately underparametrized, or is overparametrized.
> > > We also point out that, the coreset approach presents a fundamental limitation. Indeed, it aims at approximating as well as possible the full sample empirical risk minimization problem. As a consequence, it will never outperform the full sample ERM, while our approach does.

---

> > > > ### Comment · Reviewer_Vo45 · 2023-11-22
> > > >
> > > > Thanks for a prompt response and the initial comparison, the results are at the same time surprising and reasonable in the given settings (it is possible that coresets will not perform better than random in moderately underparameterized settings as suspected in your response).
> > > >
> > > > I also agree with the note about optimization objective of the data selection procedure itself, which is different between proposed work and most prior works in the field of coresets. Making a comparison to such prior works (upon re-confirmation of the preliminary claims) would allow faster dissemination of the novel knowledge to nearby fields; I am happy to raise my score due to the potential impact of this paper.

---

### Meta-Review · Area_Chair_ugsL · 2023-12-05

**Metareview:**

The paper studies the problem of selecting a subsample of the data that can be used in an ERM framework. One of the aims is here to reduce the amount of labelling that is required for inference. The reviewers are generally very positive and appreciate the contributions of the paper and have only very minor criticisms.

**Justification For Why Not Higher Score:**

n/a

**Justification For Why Not Lower Score:**

There are a variety of contributions in the paper and the reviewers rate the paper highly (beside one reviewer). I believe it is worth highlighting the contributions in this paper.

---

### Decision · Program_Chairs · 2024-01-16

Accept (oral)